# p300 nucleocytoplasmic shuttling underlies mTORC1 hyperactivation in Hutchinson–Gilford progeria syndrome

Sung Min Son [1,2], So Jung Park[1,2], Sophia Y. Breusegem[1], Delphine Larrieu [1,3] & David C. Rubinsztein [1,2] ✉

The mechanistic target of rapamycin complex 1 (mTORC1) is a master regulator of cell growth, metabolism and autophagy. Multiple pathways modulate mTORC1 in response to nutrients. Here we describe that nucleus–cytoplasmic shuttling of p300/EP300 regulates mTORC1 activity in response to amino acid or glucose levels. Depletion of these nutrients causes cytoplasm-to-nucleus relocalization of p300 that decreases acetylation of the mTORC1 component raptor, thereby reducing mTORC1 activity and activating autophagy. This is mediated by AMP-activated protein kinase-dependent phosphorylation of p300 at serine 89. Nutrient addition to starved cells results in protein phosphatase 2A-dependent dephosphorylation of nuclear p300, enabling its CRM1-dependent export to the cytoplasm to mediate mTORC1 reactivation. p300 shuttling regulates mTORC1 in most cell types and occurs in response to altered nutrients in diverse mouse tissues. Interestingly, p300 cytoplasm–nucleus shuttling is altered in cells from patients with Hutchinson–Gilford progeria syndrome. p300 mislocalization by the disease-causing protein, progerin, activates mTORC1 and inhibits autophagy, phenotypes that are normalized by modulating p300 shuttling. These results reveal how nutrients regulate mTORC1, a cytoplasmic complex, by shuttling its positive regulator p300 in and out of the nucleus, and how this pathway is misregulated in Hutchinson–Gilford progeria syndrome, causing mTORC1 hyperactivation and defective autophagy.

The mechanistic target of rapamycin complex 1 (mTORC1) complex comprises the serine/threonine-protein kinase mTOR, regulatory-associated protein of mTOR (commonly known as raptor), mammalian lethal with SEC13 protein 8 (MLST8), PRAS40 and DEPTOR[1]. It is a central regulator of core cellular functions, including growth, translation and autophagy, and its activity is positively regulated by nutrients, including amino acids (AAs) and glucose (Glc)[2,3].

Mammalian mTORC1 activity is stimulated by growth factors and intracellular energy signals via inhibition of tuberous sclerosis complex 1 and 2, which are negative mTORC1 regulators[4]. AAs signal to mTORC1 through Rag GTPases[2,5]. mTORC1 is active when tethered to lysosomes via interactions between raptor and Rag proteins on the lysosomal surface[2,6]. This allows mTOR to bind to its activator RHEB. Multiple pathways have been proposed to regulate mTORC1 activity,

[1]Cambridge Institute for Medical Research, University of Cambridge, Cambridge, UK. [2]UK Dementia Research Institute, Cambridge Institute for Medical Research, University of Cambridge, Cambridge, UK. [3]Department of Pharmacology, University of Cambridge, Cambridge, UK. ✉e-mail: dcr1000@cam.ac.uk

but these have mainly derived from studies in HEK293 cells[7]. Recently, we described that leucine (Leu) regulates mTORC1 via its metabolite acetyl-CoA (AcCoA) in most primary and cancer cell types we analysed (with a few exceptions, such as HEK293 cells), and this was supported by correlative data in vivo[8]. Elevated AcCoA levels stimulate mTORC1 activity and inhibit autophagy via raptor acetylation, which is necessary for its binding to Rag proteins and mTORC1 lysosomal localization, a requirement and driver for mTORC1 activation in response to nutrient replenishment. p300, rather than other lysine acetyltransferases (KATs), is a critical mediator of this process by regulating raptor acetylation[8,9]. mTORC1 activity inhibits macroautophagy (henceforth autophagy) via inhibitory phosphorylation of key regulators such as ULK1 (ref. [10]).

Disregulated mTOR signalling is implicated in many diseases, including neurodegeneration, as well as ageing[6]. Hutchinson–Gilford progeria syndrome (HGPS) is a rare sporadic autosomal dominant accelerated ageing disorder. At 9–24 months, affected individuals show striking growth delay and develop a characteristic appearance. The condition manifests with numerous ageing features, including decreased skin elasticity, hair loss and coronary artery disease, which typically causes death in the early teens. HGPS is caused by a silent point mutation in *LMNA* (1824C>T; G608G) that encodes the inner nuclear membrane proteins lamins A and C. Lamin A/C is critical for maintaining nuclear structural integrity and chromatin organization. The HGPS *LMNA* G608G mutation activates a cryptic splice site[11,12], generating the expression of progerin (50 AA-deleted lamin A). This shorter form of lamin A remains permanently farnesylated, resulting in its constitutive association with the inner nuclear membrane. Progerin accumulation induces diverse cellular defects, including aberrant nuclear morphology and defective DNA repair[13,14]. In addition, progerin aggregation causes mitochondrial dysfunction and oxidative stress[15] and impaired epigenetic regulation[16], altogether causing premature cellular senescence. Removal of progerin has been investigated as a therapeutic strategy for HGPS. For example, treatment of human HGPS fibroblasts with the mTORC1 inhibitor, rapamycin, and its analogue everolimus, which enhance autophagy-dependent progerin clearance[17], ameliorate nuclear blebbing. mTORC1 is hyperactivated in HGPS, although the mechanism has not been described[17,18].

In this Article, we show how p300 localization regulates mTORC1 activity in response to nutrient status. In most cell types and diverse mouse tissues, p300 nucleus–cytoplasmic transport is essential for mTORC1 regulation by starvation/restimulation, and p300 trafficking in and out of the nucleus is regulated by AMP-activated protein kinase (AMPK) and protein phosphatase 2A (PP2A), respectively. This biology is perturbed in HGPS, resulting in increased cytoplasmic p300 and mTORC1 hyperactivation.

## Results

### p300 is critical for mTORC1 regulation

To confirm the importance of p300 for mTORC1 activity regulation after AA starvation, we established p300 knockout (KO) HeLa cells using clustered regularly interspaced short palindromic repeats–associated protein 9 (CRISPR–Cas9) editing (Extended Data Fig. 1a,b). p300 KO cells showed no altered expression of other KATs, except for increased CBP expression (Extended Data Fig. 1c), a likely compensatory effect, since p300 and CBP share many targets[19,20]. p300 KO cells had decreased acetylated H4 at K12, H3 at K56 and, especially, H2B acetylation at K16, consistent with previous data (Extended Data Fig. 1d)[21]. H3 at K9 is acetylated by KAT2A or KAT2B, and not p300 (ref. [21]), and this site was unchanged in p300 KO cells (Extended Data Fig. 1d, middle). p300 KO cells had decreased lysosomal mTOR localization (Extended Data Fig. 1e), reduced raptor acetylation and consequent mTORC1 inhibition, as judged by phosphorylation of its substrate S6K1 (Extended Data Fig. 1f). AA replenishment to starved cells did not restore mTORC1 activity in p300 KO HeLa cells (Extended Data Fig. 1g). p300 KO SH-SY5Y neuroblastoma cells behaved like HeLa cells (Extended Data Fig. 1h). Leu depletion inhibits mTORC1 activity by multiple pathways, including the Leu–MCCA–p300 axis[8,22]. Leu replenishment to Leu-starved cells did not restore mTORC1 activity in p300 KO cells (Extended Data Fig. 1i). As we observed with AAs, Glc deprivation also inhibited mTORC1 activity, which was rescued by Glc replenishment, but this was abrogated by p300 KO (Extended Data Fig. 1j).

mTORC1 negatively regulates autophagy[23] and p300 KO cell clones (#2, #4, #9; Extended Data Fig. 1a) showed increased LC3-II (which correlates with autophagosome numbers) in the absence and presence of bafilomycin A1 (BafA1), an inhibitor of autophagosome–lysosome fusion (Extended Data Fig. 2a), suggesting autophagy activation by p300 depletion. As BafA1 blocks autophagosome/LC3-II degradation, LC3-II levels in the presence of this inhibitor reflect autophagosome biogenesis[24]. Also, p300 KO clones showed inhibited mTORC1 activity (Extended Data Fig. 2b) and increased the numbers of endogenous LC3-positive structures (autophagosomes) (Extended Data Fig. 2c), consistent with previous reports that p300 inhibition increases autophagosome numbers and autophagic activity[9,25,26]. To further confirm the impact of p300 depletion on autophagic flux, we utilized the signal-retaining autophagy indicator (SRAI)–LC3B assay. This ratiometric assay, recently validated in our previous study[27], generates a positive signal when autophagic substrates are degraded within lysosomes. The SRAI reporter, a tandem construct comprising TOLLES (a blue fluorescent protein resistant to acid proteolysis) and YPet (a yellow fluorescent protein that undergoes proteolysis in lysosomes)[28] was fused to the N-terminus of LC3B, a well-established autophagy marker, to adapt this tool for studying autophagy. Using this reporter, we observed an increased TOLLES:YPet ratio in p300 KO cells

**Fig. 1 | p300 nucleus–cytoplasm shuttling depends on nutrient status, and regulates mTORC1. a**, The effect of p300 acetyltransferase activity on mTORC1 signalling (phosphorylated S6K1 (p-S6K1 at Thr389)/S6K1 and p-4E-BP1 (Thr37/46)/4E-BP1) in response to AA deprivation with or without restimulation in p300 KO HeLa cells reconstituted with His-tagged p300 WT or p300 DN constructs. HeLa cells were incubated in AA-depleted medium for 1 h, followed by re-addition of AAs to the medium for 0.5 h. The blots are representative of three biologically independent experiments (N = 3). One-way ANOVA with post hoc Tukey test. **b**, p300 nucleocytoplasmic shuttling in AA-depleted medium for 1 h, and then after re-addition of AAs to the medium for 0.5 h (N = 3, about 50 cells scored per condition per experiment). Nu and Cyto indicate nuclear and cytoplasmic fractions, respectively. One-way ANOVA with post hoc Tukey test. Scale bar, 5 μm. **c**, Cytoplasmic p300 activity in AA- or Glc-starved cells with or without the re-addition of AAs or Glc, respectively. N = 5. One-way ANOVA with post hoc Tukey test. **d**, Less binding of p300 with raptor after starvation (N = 3, about 40 cells scored per condition per experiment). Scale bars, 5 μm and 1 μm (enlarged images). The arrowhead indicates the interaction of p300 with raptor. One sample *t*-test. **e**, The effect of incubation time in AA-depleted (AAs dep.) medium on mTORC1 activity (p-S6K1 at Thr389 and p-S6 at Ser235/236), whole-cell p300 activity (as acetylated p300 at Lys1499) and localization of p300 (as fractionation) in HeLa cells (N = 3). S.E. and L.E. are short and long exposures of the same blot. **f**, Reduced raptor acetylation levels by AA starvation. HeLa cells were transfected with HA-tagged raptor and then incubated in AA-depleted medium for either 60 or 120 min (N = 3). Two-tailed paired *t*-test. **g**, Validation of p300 dNLS construct (N = 4, about 30 cells scored per condition per experiment). Scale bar, 3 μm. **h**, The remaining acetylated raptor levels and mTORC1 activity (phosphorylated S6K1, S6 and 4E-BP1) in His-tagged p300 dNLS-expressing p300 KO cells in AA-depleted medium for 1.5 h. The differential expression levels following transfection are depicted by the quantification of the His-tag (N = 4). One-way ANOVA with post hoc Tukey test. Data are presented as mean values ± s.d. unless otherwise specified. Source numerical data and unprocessed blots are available in the source data.

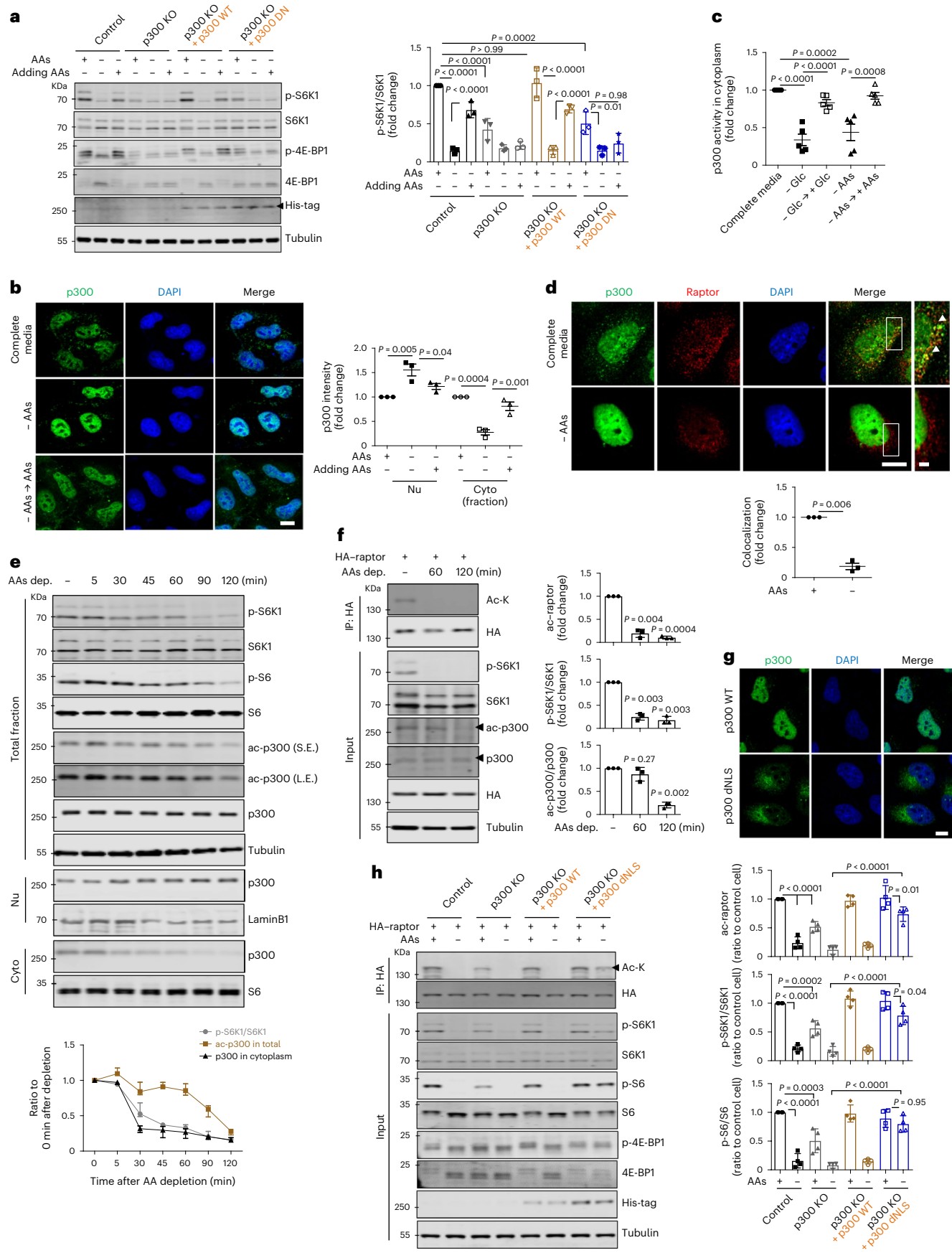

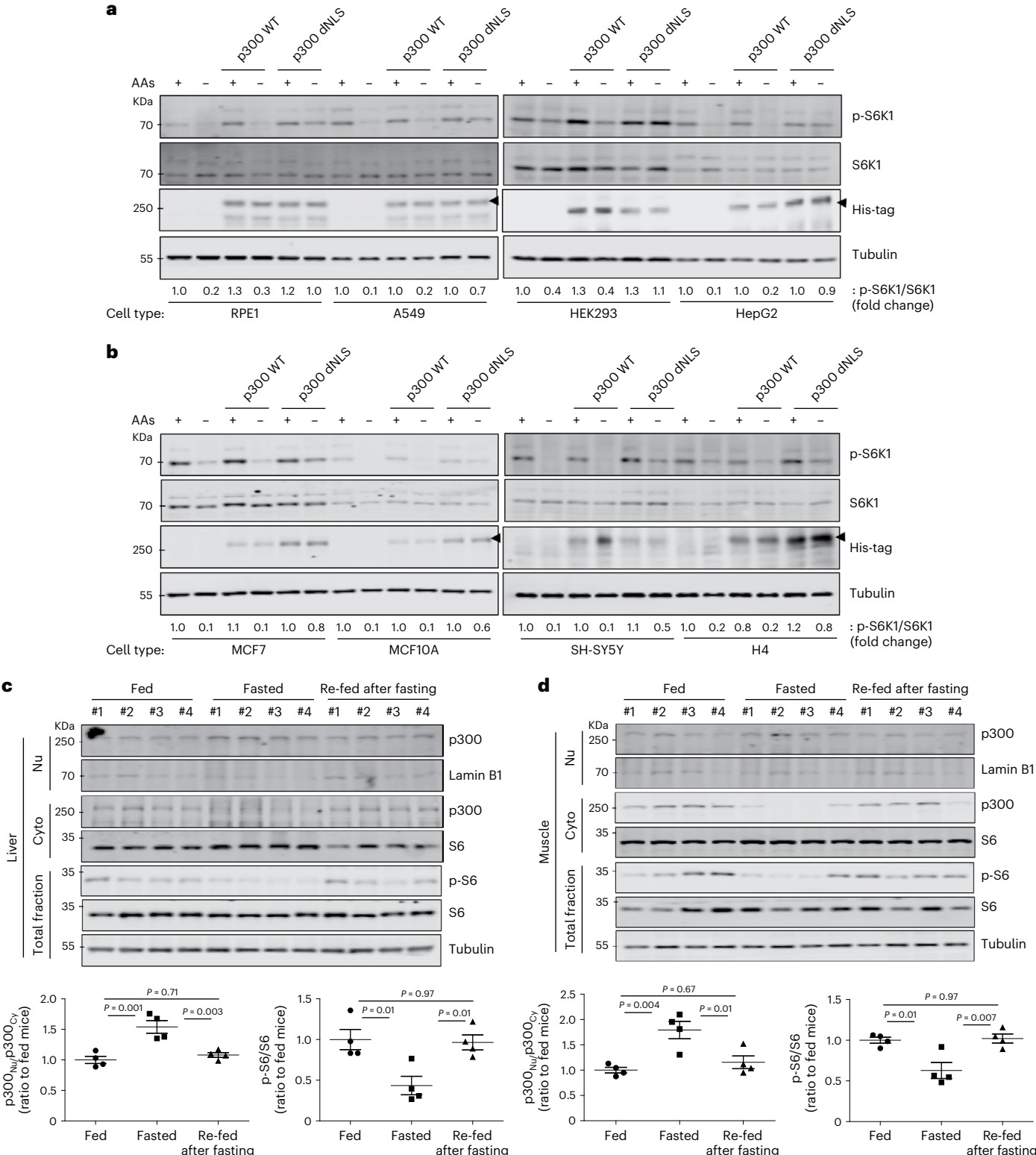

**Fig. 2 | p300 shuttling depending nutrient status in most cell types and in tissues from mice. a,b,** Remaining mTORC1 activity in His-tagged p300 dNLS-expressing cells (RPE1, A549, HEK293 and HepG2 (**a**), MCF7, MCF10A, SH-SY5Y and H4 (**b**)) in response to AA depletion for 1 h. The differential expression levels following transfection are depicted by the quantification of the His-tag. The blots are representative of three biologically independent experiments (*N* = 3). **c,d,** p300 nucleocytoplasmic shuttling in livers (**c**) and muscles (**d**) from mice fasted or re-fed after fasting. After 24 h starvation, mice were given free access to food for 2 h. The tissue samples from fed (*n* = 4) and fasted mice (*n* = 4) were analysed for mTORC1 and p300 levels in nuclear or cytoplasmic fractions. One-way ANOVA with post hoc Tukey test. Data are presented as mean values ± s.e.m. unless otherwise specified. Source numerical data and unprocessed blots are available in the source data.

compared with the control (Extended Data Fig. 2d). This shift suggests enhanced functional autophagy upon p300 depletion. Furthermore, either reducing p300 levels or inhibiting p300 activity with the specific inhibitor A485 decreased the percentage of cells containing mutant huntingtin aggregates (which correlates inversely with autophagy activity[9]) (Extended Data Fig. 2e).

We next determined whether other KATs impacted mTORC1 regulation by p300. CBP knockdown in p300 KO cells showed a slight additional effect on mTORC1 activity lowering, but this was not seen with knockdowns of other KATs (a small fraction of the proteins still remain, with approximately 20% of CBP, 10% of KAT2A and 30% of KAT2B after transfection with small interfering RNAs (siRNAs); Extended Data Fig. 2f). In CBP knockdown cells (with no p300 KOs), mTORC1 activity was decreased after AA depletion and was restored after adding back AAs to these cells, although baseline mTORC1 activity in nutrient-replete medium was decreased compared with control (Extended Data Fig. 2g), suggesting that the highly homologous acetyltransferases p300 and CBP are non-redundant with respect to mTORC1 regulation by nutrients, consistent with previous reports[29]. These data suggest that KAT3, especially p300, is crucial for nutrient regulation of mTORC1. The effects of p300 depend on its acetyltransferase activity, since the mTORC1 defects in p300 KO cells were rescued by p300 wild-type (WT) but not a dominant-negative (DN) p300 construct (Fig. 1a).

### p300 nucleocytoplasmic shuttling regulates mTORC1

Next, we investigated how nutrients impact p300 to regulate mTORC1. Previously, we assumed that AcCoA, the catabolic product of Leu (and other branched-chain AAs), directly stimulated p300 activity[8]. While p300 is predominantly localized in the nucleus[8,30,31], it can shuttle between the nucleus and cytoplasm[8,30]. As we observed previously, 1 h of AA starvation enriched nuclear p300, which was reversed by AA replenishment (Fig. 1b and Extended Data Fig. 3a). Similar results were observed with Glc starvation and replenishment (Extended Data Fig. 3a,b). Both AA and Glc depletion reduced cytoplasmic p300 activity (Fig. 1c). Consistent with our previous observation that p300-mediated raptor acetylation stimulates mTORC1 activity[20], we found less interaction of p300 with raptor after starvation (Fig. 1d), which correlated with the nuclear accumulation/cytoplasmic depletion of p300.

To try to uncouple possible direct effects of nutrients or their catabolic products, such as AcCoA, on p300 activity towards mTORC1 (ref. 8), versus effects via p300 cellular localization, we assessed whole-cell p300 activity (as acetylated p300) and localization (by fractionation) after different times of AA depletion. Short-term AA starvation (30–60 min) induced substantially more nuclear localization of p300, but did not alter its activity (Fig. 1e). AA depletion of >2 h induced altered both p300 localization and activity (Fig. 1e). Since mTORC1 activity started to decrease after 30 min of AA starvation (Fig. 1e), and acetylated raptor levels were also reduced after 60 min of AA depletion without altering p300 activity (Fig. 1f), we speculated that p300 cytoplasm-to-nuclear transport impacted mTORC1 activity after AA

removal. We tested this hypothesis using a constitutively cytoplasmic, nuclear localization signal (NLS)-deleted (dNLS) p300 construct (Fig. 1g)[30]. Unlike p300 KO cells reconstituted with p300 WT, which showed a large decrease in mTORC1 activity after AA or Glc depletion, the decrease in raptor acetylation and mTORC1 activity in cells expressing p300 dNLS was less pronounced (80% decrease in raptor acetylation in WT versus 46% of dNLS; 81 or 90% decrease of mTORC1 activity (S6K1 or S6, respectively) for WT versus 45 or 22% (S6K1 or S6, respectively) of dNLS in response to AA depletion) in these starvation conditions (Fig. 1h for AAs and Extended Data Fig. 3c for Glc).

Previously, we reported that p300 regulated mTORC1 and, subsequently, autophagy by acetylating raptor, rather than other substrates[8,9]. Consistent with these data, raptor knockdown increased LC3-II levels under both basal and BafA1-treated conditions (Extended Data Fig. 3d). This increase was reversed by reconstituting these raptor KO cells with raptor WT, but not with the acetylation-dead mutant (KR; K1097R[8]) (Extended Data Fig. 3d), consistent with our previous data showing that raptor acetylation increases mTORC1 activity and impedes autophagosome biogenesis[8,9]. We confirmed these data using a transgenically expressed tandem-tagged monomeric red fluorescent protein (mRFP)–green fluorescent protein (GFP)–LC3 autophagy flux reporter[32], where unacidified autophagosomes are green and red and appear yellow, while acidified autolysosomes are red only. While overexpression of WT raptor decreased autophagy flux (red-only LC3 vesicle numbers) in raptor-knockdown cells, this was not seen with the raptor KR mutant (Extended Data Fig. 3e). To investigate whether p300 shuttling affects autophagy via raptor acetylation, p300 KO cells expressing raptor KR showed no change in LC3-II (in the presence or absence of BafA1), compared with raptor WT. However, in p300 KO cells reconstituted with p300 WT or dNLS, we observed elevated LC3-II levels when raptor KR was transfected compared with raptor WT (Extended Data Fig. 3f). Thus, cytoplasmic p300 negatively regulates autophagosome biogenesis (LC3-II in BafA1) in a manner that is dependent on the raptor acetylation site K1097.

Previously, simultaneous loss of CBP and p300 in skeletal muscle and adipocytes abrogated insulin-stimulated cellular uptake of Glc[33]. However, p300 KO HeLa cells did not exhibit significant differences in Glc uptake levels compared with control cells (Extended Data Fig. 3g).

p300 nuclear transport after starvation occurred in all cell types we studied: A549, HepG2, RPE1, HEK293, SH-SY5Y, H4, MCF7, MCF10A and mouse embryonic fibroblast (MEF) cells (Extended Data Fig. 4a,b). Furthermore, when p300 dNLS was transfected in these different cell lines, we found residual mTORC1 activity even under starvation conditions (Fig. 2a,b). mTORC1 activity (phosphorylated S6) was decreased along with increased p300 nuclear localization in fasted mice, which were reversed in mice re-fed after fasting (Fig. 2c,d and Extended Data Fig. 4c). Likewise, the brains of starved mice, which have decreased raptor acetylation and mTORC1 activity[8], also have more nuclear p300 (Extended Data Fig. 4d). These data suggest that p300 shuttling-mediated mTORC1 regulation is a general response to nutrient status in diverse cell types in culture and in vivo.

---

**Fig. 3 | AMPK is crucial for p300 nuclear import. a**, AMPK-dependent nuclear translocation of p300 after AA depletion using several inhibitors: 10 µM U0126 (U), an ERK inhibitor; 0.5 µM Torin1 (T), an mTOR inhibitor; 5 µM Comp.C (C), an AMPK inhibitor; 10 µM LY294002 (L), an inhibitor of PI3K/Akt and 0.5 µM Go6983 (G), a PKC inhibitor. The graph shows data from three independent biological replicates (N = 3, about 50 cells analysed per condition per experiment). One-way ANOVA with post hoc Tukey test. Scale bar, 10 µm. **b**, Change in p300 localization in AA-depleted medium for 1 h with or without inhibitors using fractionation (N = 3). One-way ANOVA with post hoc Tukey test. **c**, Failure of p300 nuclear transport in AA-starved AMPK knockdown cells (N = 3). Con, control siRNA. One-way ANOVA with post hoc Tukey test. **d**, Failure of nuclear transport of the SA mutant after AA depletion. HeLa cells were transfected with His-tagged p300 WT or SA and then incubated in AA-depleted medium for 1 h (N = 3). One sample

t-test. **e**, Localization of His-tagged p300 WT and p300 S89A mutant in complete or AA-depleted medium (N = 3, about 40 cells analysed per condition per experiment). One-way ANOVA with post hoc Tukey test. Scale bar, 5 µm. **f**, Remaining mTORC1 activity in p300 SA-expressing p300 KO cells in AA-depleted medium for 1 h. The differential expression levels following transfection are depicted by the quantification of the His-tag (N = 3). One-way ANOVA with post hoc Tukey test. **g**, Compromised mTORC1 inhibition in His-tagged p300 SA-expressing p300 KO cells after depletion of single AAs (Arg, Leu, Ile and Val). The phosphorylated ACC at Ser79 was measured to assess AMPK activity. The expression levels following transfection are depicted by the quantification of the His-tag (N = 3). One-way ANOVA with post hoc Tukey test. Data are presented as mean values ± s.d. unless otherwise specified. Source numerical data and unprocessed blots are available in the source data.

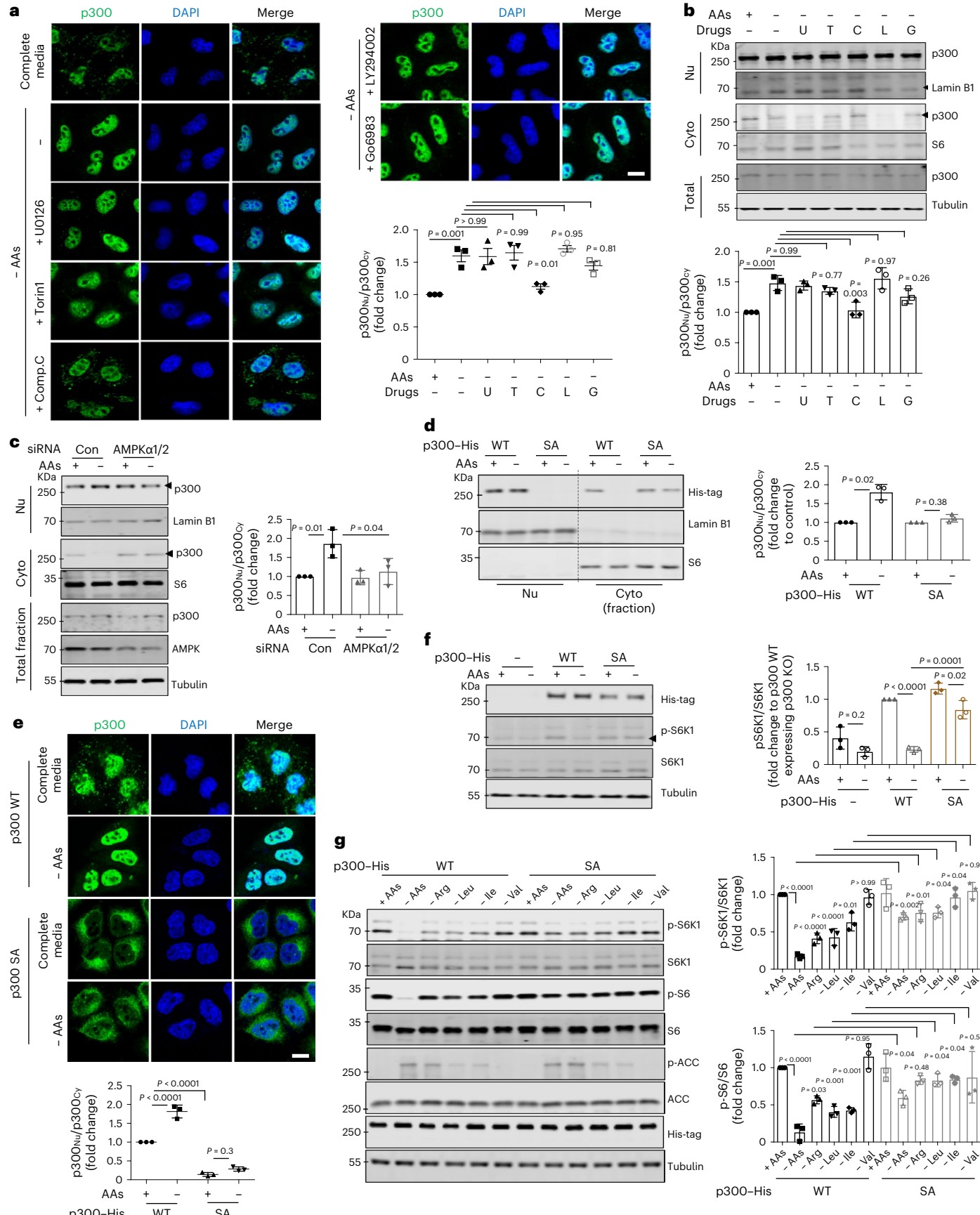

## AMPK is crucial for p300 nuclear import

Next, we investigated how p300 nucleocytoplasmic shuttling is regulated by nutrients. Previous studies suggested that p300 interactors, such as BAG6 or ALX1, regulate its cytoplasm–nuclear shuttling[34,35]. However, BAG6 or ALX1 knockdowns did not affect p300 nuclear/cytoplasmic distributions after AA starvation for 1 h (Extended Data Fig. 5a,b). Also, the p300 activator CTB (N-(4-chloro-3-trifluoromethyl-phenyl)-2-ethoxy-benzamide) and inhibitor c646 did not alter its localization (Extended Data Fig. 5c). Kinases, including extracellular signal-regulated kinase (ERK), AMPK, mTOR and protein kinase C (PKC) can phosphorylate p300 (refs. 31,36–39). Among specific inhibitors of these kinases, only AMPK inhibition (using compound C (Comp.C); also called dorsomorphin), inhibited nuclear transport of p300 after starvation (Fig. 3a,b) without altering its expression level or activity (Extended Data Fig. 6a). The AMPK activator 991 (also called EX229) induced nuclear accumulation of p300 (Extended Data Fig. 6b) and mTORC1 inhibition (Extended Data Fig. 6c). AMPK activation decreased raptor acetylation and mTORC1 activity in nutrient-replete cells (Extended Data Fig. 6d) and AMPK inhibition rescued the reduced raptor acetylation and mTORC1 activity in starved cells (Extended Data Fig. 6e). AMPK inhibition affected decreased acetylation of nuclear histones (Extended data Fig. 6f) and increased acetylation of p300 cytoplasmic substrates (raptor, ATG7 and hVps34) (Extended Data Fig. 6f). Likewise, AMPK activation in nutrient-replete cells mimicked the decreased cytoplasmic protein acetylation and increase in nuclear protein acetylation seen in starvation (Extended Data Fig. 6f).

Glc starvation inhibits mTORC1 and increases AMPK activity[40,41], so we tested whether AA depletion also activated AMPK. Previously, AA depletion was reported to induce $Ca^{2+}$/calmodulin-dependent protein kinase beta (CaMKKβ)-mediated AMPK activation[42]. Indeed, when STO-609, a selective inhibitor of CaMKK[43], was added to starved cells, we found significant attenuation of AMPK activation and blunted inhibition of S6K1 phosphorylation (Extended Data Fig. 6g). In addition, STO-609 restored p300 nuclear accumulation in starved cells (Extended Data Fig. 6h), indicating that CaMKKβ–AMPK induces p300 nuclear transport by starvation.

AMPK can phosphorylate p300 at serine 89 (ref. 39), and this phosphorylation was induced by AA starvation in an AMPK-dependent manner but did not affect total p300 levels (Extended Data Fig. 7a,b). Consistent with a role for AMPK in p300 shuttling, AMPK knockdown HeLa cells showed less p300 nuclear accumulation after AA starvation (Fig. 3c) without altering p300 activity (Extended Data Fig. 7c). Likewise, the starvation-induced increase in phospho-S89–p300 and p300 nuclear transport were impaired in AMPK double KO MEFs (Extended Data Fig. 7d–g), and were rescued by reconstituting these cells with α2 AMPK but not with DN kinase-dead AMPK α2 K45R (ref. 44) (Extended Data Fig. 7g). Phosphorylation of p300 at S89 is necessary for p300 nuclear transport by starvation, since p300 nuclear transport and lowering of mTORC1 activity after AA or Glc depletion

were compromised in cells expressing non-phosphorylatable p300 S89A (SA) (which has normal acetyltransferase activity (Extended Data Fig. 7h)) but not in cells expressing WT p300 (Fig. 3d–f and Extended Data Fig. 3c). The phosphomimic p300 S89D (SD) mutant showed the opposite effect on mTORC1 compared with p300 SA (Extended Data Fig. 7i). To investigate which AAs regulate mTORC1 through AMPK, we tested single AA-depleted medium (Arginine (Arg)−, Leu−, isoleucine (Ile)− or valine (Val)−) on cells expressing p300 WT or SA. Depletion of Arg, Leu or Ile but not Val activated AMPK (p-ACC at Ser79/total ACC) and inhibited mTORC1 in p300 WT expressing cells, but p300 SA expression blunted mTORC1 inhibition by single AA depletion (Fig. 3g). These data suggest that starvation results in AMPK-dependent p300 phosphorylation that leads to its nuclear translocation, leading to decreased raptor acetylation and mTORC1 activity.

The 14-3-3 proteins can mediate nucleus–cytoplasmic transport of various proteins, and p300 was identified as a 14-3-3 binding protein[29,45]. Phosphorylated p300 is predominantly located in the nucleus both in nutrient-replete and starvation media (Extended Data Fig. 8a). We found that p300 interacted with 14-3-3ζ, not 14-3-3ε, particularly on starvation (Extended Data Fig. 8b), and the interaction was inhibited by the AMPK inhibitor, Comp.C (Extended Data Fig. 8c,d), indicating that phosphorylation of p300 by AMPK is important for p300–14-3-3ζ binding. The p300 SA mutant did not interact with 14-3-3ζ (Extended Data Fig. 8e). Indeed, 14-3-3ζ knockdown relocated p300 to the cytoplasm and compromised mTORC1 inhibition after starvation (Extended Data Fig. 8f,g), suggesting that phosphorylated p300 was trapped by 14-3-3ζ proteins in the nucleus during AA starvation. Thus, 14-3-3ζ enables nuclear localization of AMPK-phosphorylated p300.

## PP2A is a p300 phosphatase enabling its nuclear export

As p300 phosphorylation regulates its nuclear import, we sought to identify the relevant phosphatase. Liquid chromatography–mass spectrometry analysis suggested that PP2A may interact with p300 (ref. 46), and when AAs were added to starved cells, we observed binding of p300 with PP2Ac (Fig. 4a). Furthermore, the specific PP2A inhibitor (PP2Ai; cantharidin[47]) prevented the dephosphorylation of p300–S89 after AA restimulation of starved cells (Fig. 4b). PP2A catalytic subunit (PP2Ac) activity (which correlates negatively with phosphorylated PP2Ac levels) was activated after AAs were added to starved cells (Fig. 4c,d). PP2Ac knockdown (Fig. 4e) or PP2A inhibition (Fig. 4f) prevented p300 from exiting the nucleus. After PP2A-inhibited starved cells were replenished with AAs, p300 was still in the nucleus, even in AAs restimulation. Consistent with these data, PP2Ac knockdown compromised the rescue of mTORC1 activity mediated by the addition of AAs to starved cells (Fig. 4g), which could be attributed to inhibition of the interaction of raptor with the Rag complex after PP2A inhibition (Fig. 4h), leading to less lysosomal localization of mTORC1 (its site of activation) (Fig. 4i).

**Fig. 4 | PP2A is a p300 phosphatase enabling its nuclear export. a**, The effect of incubation time after addition of AAs to starved cells on the interaction of myc-tagged p300 with PP2Ac. The graph shows data from three independent biological replicates (N = 3). One-way ANOVA with post hoc Tukey test. **b**, The level of phosphorylated p300 at S89 after treatment with PP2Ai (5 μM, cantharidin) (N = 3). One-way ANOVA with post hoc Tukey test. **c**, Decreased phosphorylation of PP2Ac at Y307 by the addition of AAs to starved cells (N = 3). One-way ANOVA with post hoc Tukey test. **d**, Measurement of PP2A activity using an in vitro PP2A activity assay kit. Okadaic acid (OA) was used as a negative control (N = 3). One-tailed paired t-test. **e**, Failure of cytoplasmic translocation of p300 after addition of AAs to starved PP2Ac knockdown cells. Note that PP1 knockdown cells showed the same to control cells (N = 3). One-way ANOVA with post hoc Tukey test. **f**, Failure of cytoplasmic translocation of p300 by PP2Ai (5 μM, cantharidin) when starved cells were replenished with AAs. Scale

bar, 5 μm. N = 3, about 30 cells scored per condition per experiment. One-way ANOVA with post hoc Tukey test. **g**, Failure of rescue of mTORC1 activity in PP2Ac knockdown cells when AAs were added after starvation (N = 3). One-way ANOVA with post hoc Tukey test. **h**, Inhibition of the interaction of raptor with Rag complex after treatment with PP2Ai. The expression levels following transfection were determined by using antibodies specific to the tagged protein (N = 3). One-way ANOVA with post hoc Tukey test. **i**, Control and PP2Ac-knockdown HeLa cells were immunostained with mTOR and LAMP1 antibodies as shown. Colocalization panels show an overlap between mTOR and LAMP1 signals. The fraction of mTOR-positive lysosomes was determined using Volocity software (N = 3, about 80 cells scored per condition). One-way ANOVA with post hoc Tukey test. Scale bars, 5 μm and 1 μm (enlarged images). Data are presented as mean values ± s.d. unless otherwise specified. Source numerical data and unprocessed blots are available in the source data.

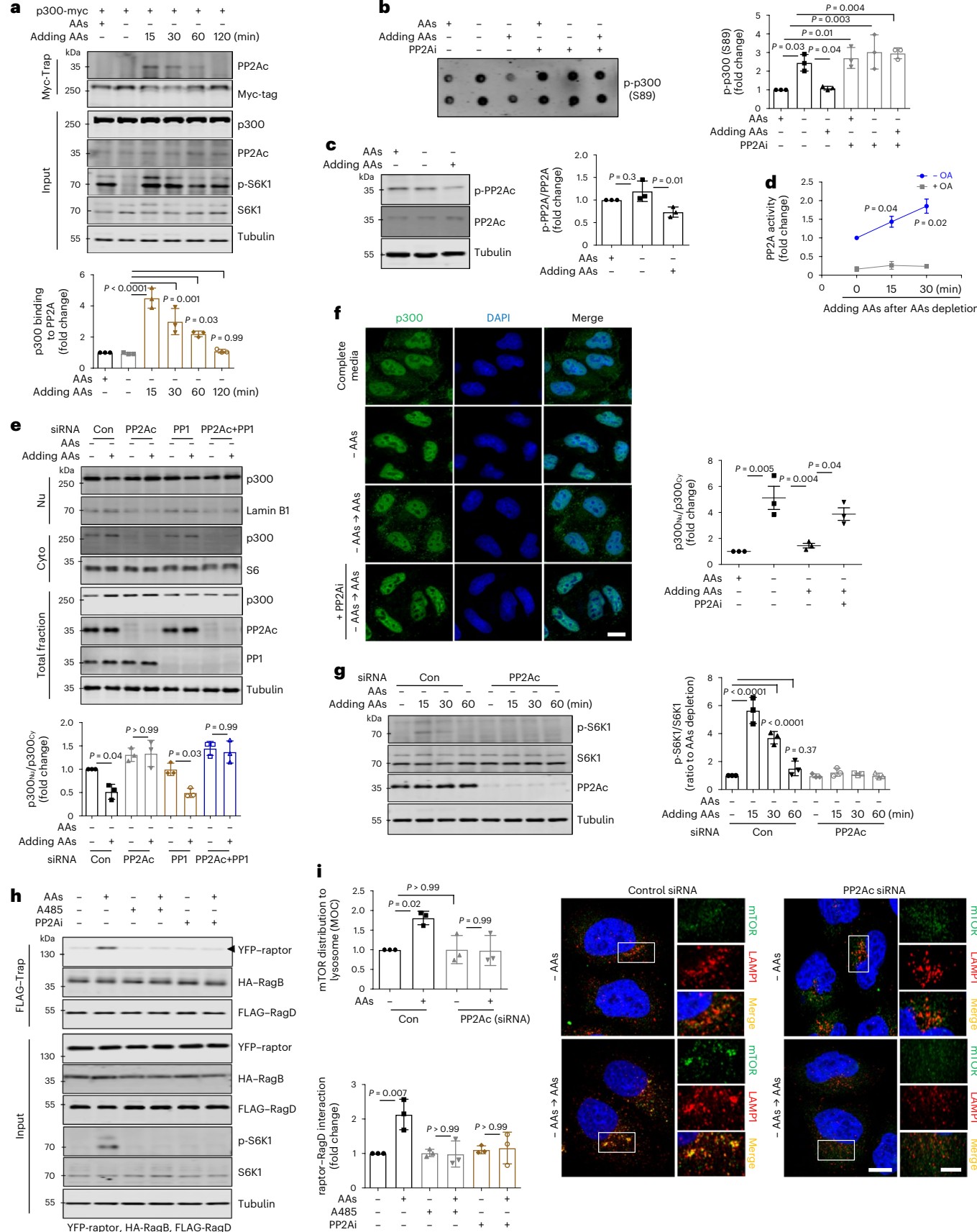

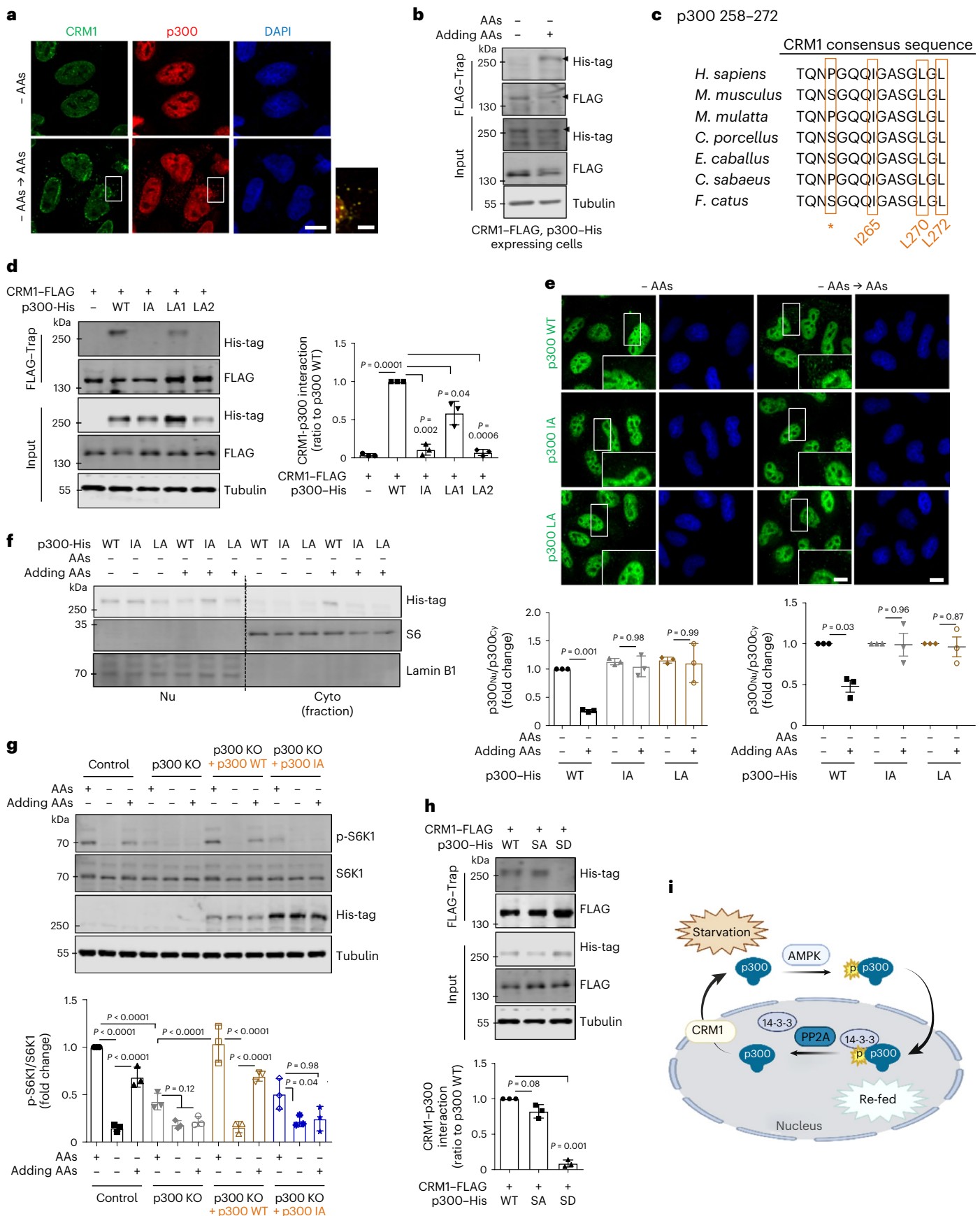

**Fig. 5 | p300 can be exported from nucleus in a CRM1-dependent manner.**
**a,b**, Interaction of p300 with CRM1 when AAs were added after starvation. For
immunoprecipitation, HeLa cells were cotransfected with His-tagged p300
and FLAG-tagged CRM1 (**b**). The expression levels following transfection were
determined by using antibodies specific to the tagged protein. The images (**a**)
and blots (**b**) are representative of three biologically independent experiments
($N = 3$). Scale bars, 5 μm and 1 μm (enlarged images). **c**, Cross-species sequence
alignment of a CRM1 consensus sequence located in the p300 protein. Full names
of species: *Homo sapiens*; *Mus musculus*; *Macaca Mulatta*; *Cavia porcellus*; *Equus
caballus*; *Chlorocebus sabaeus*; *Felis catus*. *hydrophobic residue. **d**, Interaction
of CRM1 with p300 mutant. For immunoprecipitation, HeLa cells were
cotransfected with His-tagged p300 and FLAG-tagged CRM1. The differential
expression levels following transfection are depicted by the quantification of the
His-tag ($N = 3$). Two-tailed paired *t*-test. **e**, p300 KO HeLa cells were transfected
with either His-tagged p300 WT or His-tagged NES-mutant p300 (IA and LA2)

subjected to AA starvation/re-feeding and analysed by confocal microscopy
for p300 localization. One sample *t*-test. Scale bars, 5 μm and 1 μm (enlarged
images). **f**, Failure of cytoplasmic translocation of p300 after addition of AAs
to NES-mutant p300-expressing cells ($N = 3$). One-way ANOVA with post hoc
Tukey test. **g**, Inhibition of mTORC1 activity in His-tagged NES-mutant p300-
expressing p300 KO cells when AAs were added after starvation. The differential
expression levels following transfection are depicted by the quantification of
the His-tag ($N = 3$). One-way ANOVA with post hoc Tukey test. **h**, Interaction of
CRM1 with p300 SA or SD mutant. For immunoprecipitation, HeLa cells were
cotransfected with His-tagged p300 and FLAG-tagged CRM1. The expression
levels following transfection are determined by using antibodies specific to
the tag protein ($N = 3$). Two-tailed paired *t*-test. **i**, A schematic diagram of the
nucleus–cytoplasmic transport of p300 in response to nutrients. Data are
presented as mean values ± s.d. unless otherwise specified. Source numerical
data and unprocessed blots are available in the source data.

## Dephosphorylated p300 is exported via a CRM1-dependent pathway

p300 is a large protein ~300 kDa, so its dephosphorylated form requires
exportin for nucleus-to-cytoplasm export. Liquid chromatography–
mass spectrometry data indicate that CRM1, a well-known exportin
protein, may interact with p300 (ref. 48). AA restimulation induced
more colocalization of p300 with CRM1 (Fig. 5a) and enhanced bind-
ing (Fig. 5b). As there is CRM1 consensus sequence in p300 (258–272)
in different species (Fig. 5c), we made putative CRM1–p300 binding
mutants (I265A (IA), L270A (LA1) and L272A (LA2)). IA and LA2 did not
bind CRM1 completely (Fig. 5d) and we predicted and confirmed that
they were deficient in nuclear–cytoplasm transport (Fig. 5e). Also,
after AA restimulation post-starvation, p300–IA and p300–LA2 were
not exported to the cytoplasm (Fig. 5e,f). When p300–WT or p300–IA
were transfected into p300 KO cells, p300–IA failed to rescue mTORC1
activity after AA restimulation, unlike p300–WT (Fig. 5g), suggesting
that CRM1-dependent p300 translocation from the nucleus to the
cytoplasm is important for mTORC1 activity after nutrient replenish-
ment. As expected, in nutrient-replete media, the SD did not inter-
act with CRM1, while the WT (which is largely dephosphorylated in
these conditions (Fig. 4b)) and the SA mutant interacted similarly
with CRM1 (Fig. 5h,i).

## p300 mislocalization and mTORC1 hyperactivity in HGPS

HGPS is associated with hyperactive mTORC1 and altered nucleus–
cytoplasmic transport[14,18,49]. Thus, we tested whether p300 cyto-
plasm–nuclear shuttling is an important mechanism underlying this
condition using doxycycline (Dox)-inducible progerin-expressing
cell lines and patient-derived fibroblasts (Supplementary Table 1).
Progerin expression induced abnormal nuclear morphology and
decreased lamin B1 protein levels, as previously reported (Extended
Data Fig. 9a)[17,50], compared with WT lamin A/C expressing cells. As
expected, nucleus–cytoplasmic transport was altered by progerin
when we used a nuclear-cytoplasm shuttling reporter or assessed RAN
nuclear localization[51,52] (Extended Data Fig. 9b,c). Progerin expres-
sion attenuated nuclear transport of p300 (Fig. 6a and Extended
Data Fig. 9d) and increased mTORC1 activity (Fig. 6b,c) in both
nutrient-replete and AA-depleted media. Also, progerin expression

decreased acetylation of nuclear histones regulated by p300 and
rather increased acetylation of p300 cytoplasmic substrates (rap-
tor and ATG7) (Extended Data Fig. 9e), suggesting that progerin
expression affects p300 shuttling to regulate differential acetyla-
tion between the nucleus and cytoplasm. Furthermore, p300 nuclear
transport and lowering of mTORC1 activity after AA depletion were
also compromised in primary dermal fibroblasts cells from HGPS
patients (Fig. 6d,e). These data suggest that mTORC1 dysregula-
tion in HGPS is associated with alteration in p300 localization
by progerin.

Previously, progerin was reported to upregulate CRM1 expression
and activity to mediate enhanced nuclear protein export, and induce
HGPS phenotypes[53]. In progerin-inducible RPE1 cells and HGPS fibro-
blasts, CRM1 expression was increased (Extended Data Fig. 9f,g) but
HGPS fibroblasts failed to activate AMPK (AMPK phosphorylation)
after starvation (Extended Data Fig. 9g), suggesting less p300 nuclear
import by AMPK and enhanced p300 export by upregulated CRM1 in
HGPS cells. The AMPK activator 991 or the CRM1 inhibitor selinexor
(SEL) decreased cytoplasmic p300 activity in progerin-expressing
cells (Extended Data Fig. 9h). These data suggest that upregulated
CRM1 and less AMPK activation in HGPS cells induce more cytoplas-
mic p300 localization, which would result in mTORC1 activation and
autophagy inhibition.

## p300–mTORC1 autophagy impacts HGPS cellular phenotypes

p300 inhibition (using A485) led to a decrease in progerin expression and
normalized the mTORC1 hyperactivation and autophagy inhibition seen
in HGPS fibroblasts compared with age- and gender-matched control
fibroblasts (Fig. 7a and Extended Data Fig. 9i). Progerin-mediated DNA
damage (assessed using an antibody against the DNA double-strand
break marker γ-H2AX) in HGPS fibroblasts was also reduced by treat-
ment with A485 (Fig. 7a,b and Extended Data Fig. 9i). Also, reduced
lamin B1 levels in HGPS fibroblasts (Extended Data Fig. 9j) was rescued
by treatment with A485, 991 (AMPK activator) or SEL (CRM1 inhibi-
tor) (Fig. 7c). Another well-known feature of HGPS cells is the loss of
heterochromatin marks. Accordingly, we found that tri-methylation of
lysine 9 on histone H3 (H3K9me3) was decreased in HGPS cells
(Fig. 7d,e) and these were similarly restored by A485, 991 or SEL (Fig. 7e).

**Fig. 6 | HGPS cells exhibit mislocalization of p300 and activation of mTORC1.**
**a**, Localization of p300 in Dox-inducible GFP–lamin A/C or progerin-expressing
RPE1 cells in complete or AA-depleted media for 1 h. Scale bar, 5 μm. $N = 3$,
about 50 cells scored per condition per experiment. One-way ANOVA with
post hoc Tukey test. **b**, Enhanced mTORC1 activity in Dox-inducible lamin A/C
or progerin-expressing RPE1 cells in complete or AA-depleted media for 1 h.
Scale bars, 5 μm and 1 μm (enlarged images). $N = 4$, about 40 cells scored per
condition per experiment. One-way ANOVA with post hoc Tukey test. **c**, Enhanced
mTORC1 activity in GFP-tagged progerin-expressing RPE1 cells ($N = 3$). One
sample *t*-test. **d**, Localization of p300 in control or HGPS fibroblasts in complete

or AA-depleted media for 2 h. Scale bars, 5 μm and 1 μm (enlarged images).
The arrowhead indicates cytoplasmic p300. $N = 4$, about 40 cells scored per
condition. One-way ANOVA with post hoc Tukey test. **e**, The remaining mTORC1
activity and failure of autophagy activation in HGPS fibroblasts in AA-depleted
medium. Vial #1 from the control was paired with HGPS #1 and control vial #2 was
paired with HGPS #2, as these pairs were tested on different days ($N = 3$). One-way
ANOVA with post hoc Tukey test. Data are presented as mean values ± s.d. unless
otherwise specified. Source numerical data and unprocessed blots are available
in the source data.

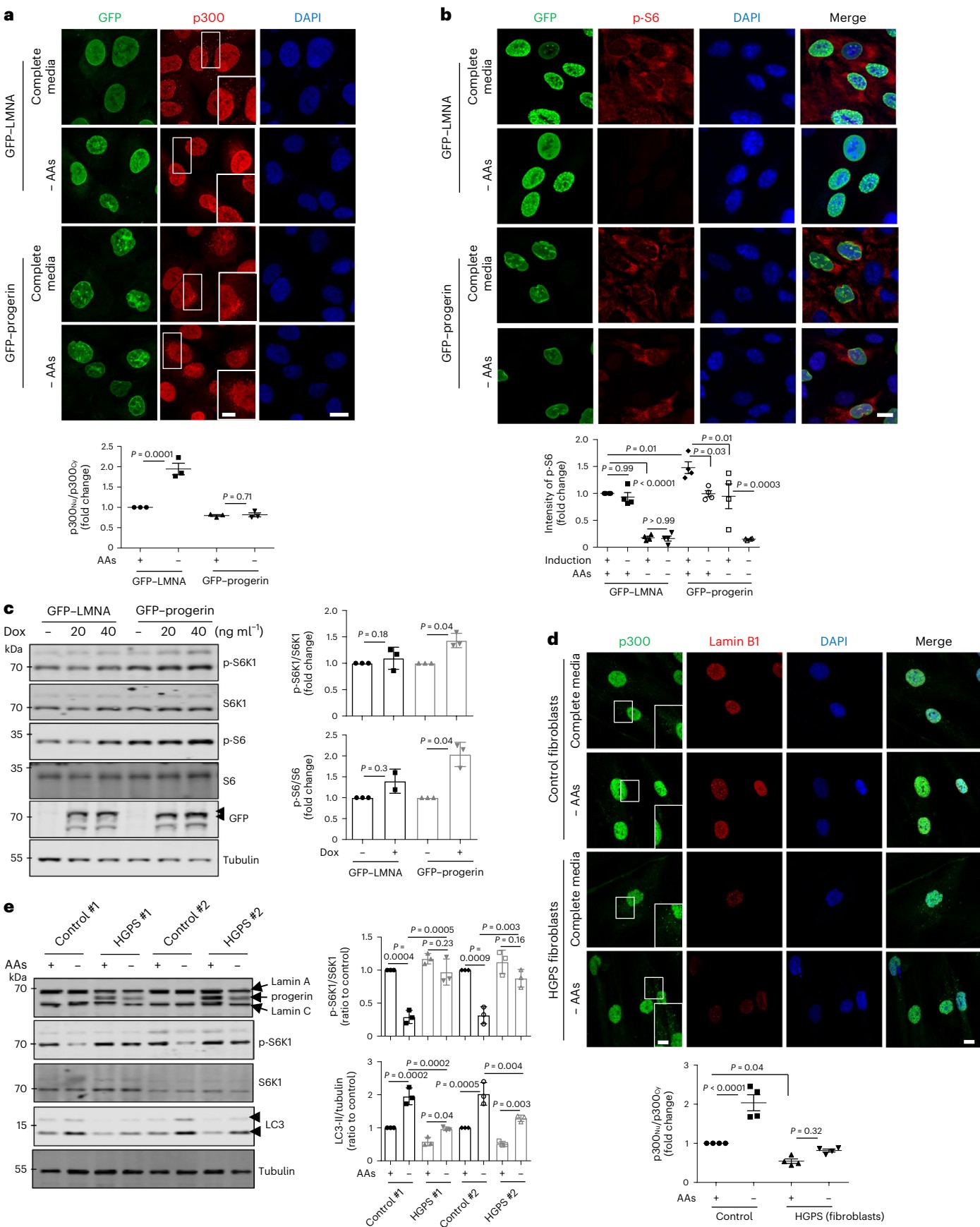

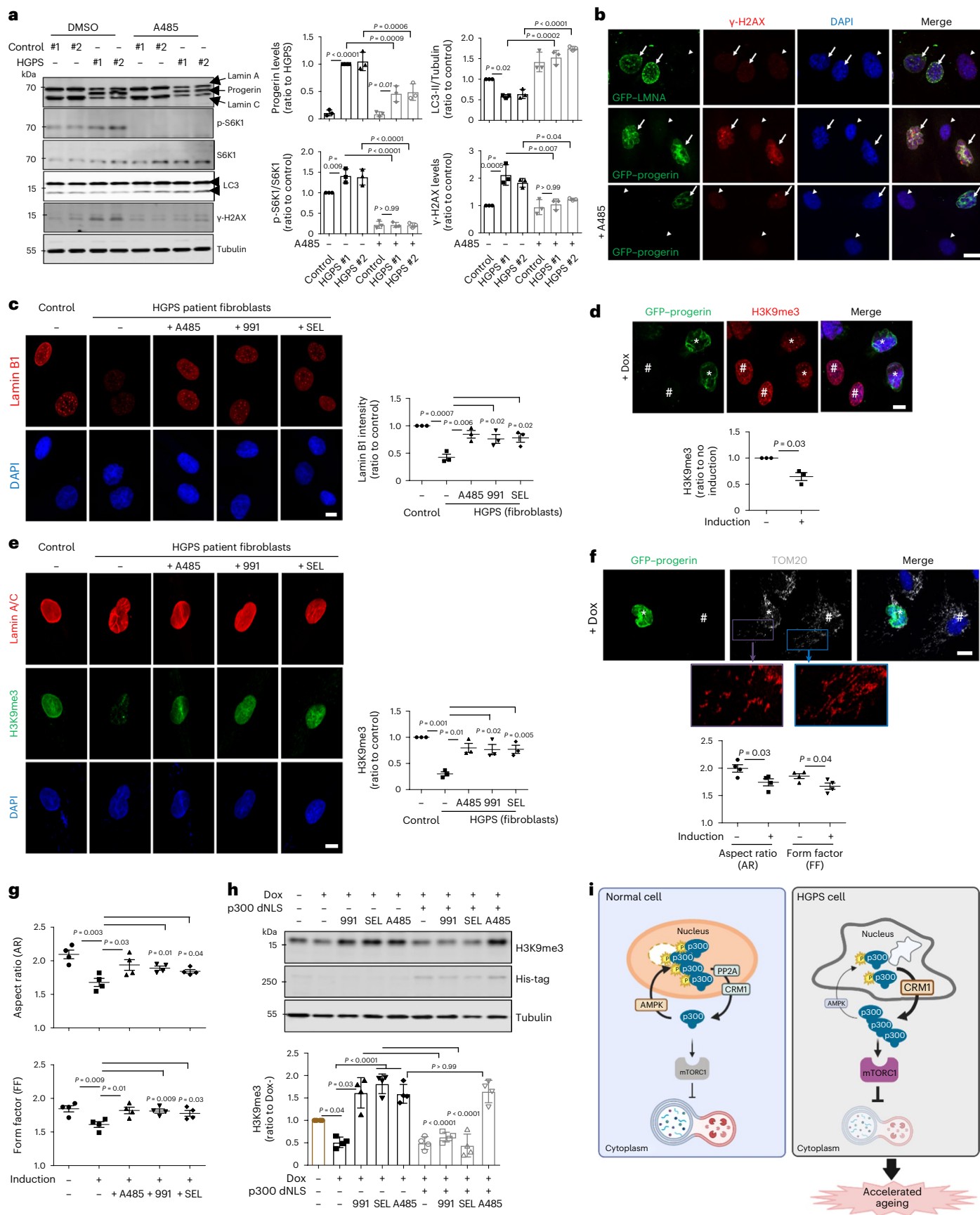

**Fig. 7 | Rescue of p300 mislocalization alleviates ageing features of HGPS cells. a**, Rescue of increased DNA damage (γ-H2AX) in HGPS fibroblasts by treatment with p300 inhibitor A485 for 24 h. The blots are representative of three biologically independent experiments ($N = 3$). One-way ANOVA with post hoc Tukey test. Data from control vials #1 and #2 were pooled as these are from the same individual (Supplementary Table 1). **b**, Rescue of DNA damage by progerin using A485. Arrow, induced cells and arrowhead, not induced cells. Scale bar, 5 μm. $N = 3$. **c**, Rescue of reduced lamin B1 levels in HGPS fibroblasts by treatment with A485, 991 or SEL. Scale bar, 5 μm. $N = 3$, about 50 cells scored per condition. One-way ANOVA with post hoc Tukey test. **d**, Decreased H3K9me3 expression level by progerin induction for 48 h. *GFP–progerin-induced cells, #not GFP–progerin-induced cells. Scale bar, 5 μm. $N = 3$, about 50 cells scored per condition. One sample *t*-test. **e**, Rescue of reduced H3K9me3 level in HGPS fibroblasts by treatment with A485, 991 or SEL. Scale bar, 5 μm. $N = 3$, about 40 cells scored per condition. One-way ANOVA with post hoc Tukey test. **f**, Altered mitochondrial morphology by progerin induction for 48 h. *GFP–progerin-induced cells, #not GFP–progerin-induced cells. Scale bars, 5 μm and 1 μm (enlarged images). $N = 4$, about 50 cells scored per condition. Two-tailed unpaired *t*-test. **g**, Rescue of defect in mitochondrial morphology in progerin-expressing cells by treatment with A485, 991 or SEL ($N = 4$, about 50 cells scored per condition). Two-tailed unpaired *t*-test. **h**, Rescue of reduced H3K9me3 level in HGPS cells by 991 or SEL is mediated by p300 localization using His-tagged p300 dNLS mutant. The expression levels following transfection are depicted by the quantification of the His-tag ($N = 4$). One-way ANOVA with post hoc Tukey test. **i**, A schematic diagram of this study. Depletion of nutrients causes cytoplasm-to-nucleus relocalization of p300, reducing mTORC1 activity and activating autophagy. This is mediated by AMPK-dependent phosphorylation of p300 at serine 89. Nutrient addition to starved cells results in PP2A-dependent dephosphorylation of nuclear p300, enabling its CRM1-dependent export to the cytoplasm to mediate mTORC1 reactivation. In HGPS cells, p300 cytoplasm–nucleus shuttling is altered, causing mTORC1 hyperactivation and autophagy inhibition. Modulating p300 shuttling normalizes HGPS phenotypes. Data are presented as mean values ± s.d. unless otherwise specified. Source numerical data and unprocessed blots are available in the source data.

Next, we found the altered mitochondrial morphology described previously in HGPS cells[15,54] (Fig. 7f) was improved by A485, 991 or SEL (Fig. 7g).

H3K9me3 levels were not restored by 991 or SEL in progerin-expressing cells when p300 dNLS was expressed (Fig. 7h), suggesting that altered p300 shuttling in HGPS cells underlies many of the critical phenotypes associated with mTORC1. Consistent with these data, A485, 991 or SEL accelerated progerin degradation, which was blocked by the autophagy–lysosome inhibitor BafA1 (Extended Data Fig. 10a,b). Also, LC3-positive dots colocalized more with progerin after treatment with p300 inhibitor A485 (Extended Data Fig. 10c). Thus, progerin is degraded by autophagy but progerin itself inhibits autophagy (and its own degradation). This raises the possibility that normalization of some defects in progerin-expressing cells after inhibition of cytoplasmic p300 activity may be, at least partially, driven by effects on progerin levels. To investigate whether p300 shuttling affects progerin degradation, we assessed the levels of remaining progerin in p300 KO cells expressing p300 WT, dNLS or IA. Progerin expression was restored in cells expressing p300 WT or dNLS, while p300 IA expression lowered progerin levels (Extended Data Fig. 10d). These results indicate that p300 shuttling can indeed modulate progerin levels through its influence on autophagic activity.

## Discussion

Our data suggest nucleocytoplasmic p300 shuttling regulates the mTORC1–autophagy response to AAs or Glc starvation and replenishment. These effects occur without any obvious inherent alteration of p300 activity, at least at early times after nutrient perturbations. While this shuttling may not be the sole determinant of mTORC1–autophagy responses in this context, it is clearly rate limiting and operates in diverse cell types in culture and mice.

Previous studies have shown that AAs signal to mTORC1 through Rag GTPases[2,5], which are regulated by several factors, including GATOR1, GATOR2 and Sestrin 2 (refs. [6,55]). Rag heterodimers are anchored to lysosomal membranes by associating with the lysosome-resident pentameric Ragulator complex[6]. After AA replenishment, Rag GTPases transition to their active nucleotide-bound state and active Rag GTPases then associate with raptor in mTORC1, facilitating the lysosomal recruitment of mTORC1. Multiple studies have underscored the importance of post-translational modifications of raptor, including acetylation by p300 at Lys1097, in regulating mTORC1. These modifications influence the recruitment of cytoplasmic raptor to Rag GTPases on the lysosomal membrane[8,56,57].

p300 activates mTORC1[8,20,25], and our data (Fig. 7i) suggest that p300 cytosolic localization is essential for AcCoA-mediated mTORC1 activation. Cytoplasm-to-nucleus p300 shuttling in response to AA or Glc deprivation is regulated by AMPK-dependent p300 phosphorylation, enabling 14-3-3ζ binding and nuclear entrapment of p300. Export from the nucleus after nutrient replenishment is mediated by PP2A-dependent p300 dephosphorylation (within short time frames of 15–30 min) and its interaction with the exportin, CRM1. This provides a central mechanism for mTORC1 regulation depending on nutrient status. In future, it would be intriguing to investigate whether hypoxia-[58] or DNA damage-[59]mediated mTORC1 inhibition, which are AMPK dependent, also involve nucleocytoplasmic transport of p300.

p300 cytoplasm–nuclear shuttling defects also appear to be involved in HGPS. Previous reports have shown altered mTORC1 activity in HGPS[18] and mTORC1 inhibition abolishes characteristic HGPS phenotypes, including nuclear defects, prolongs cellular life span and enhances progerin degradation by autophagy[17,18,60]. Our data show that p300 is more localized in the cytoplasm of HGPS cells (Fig. 6d and Extended Data Fig. 9d) where it induces acetylation of raptor (and other cytoplasmic proteins, such as ATG7) causing mTORC1 activation (Extended Data Fig. 9e). mTORC1 activation in HGPS cells compromises autophagy and inhibits progerin degradation[17].

Multiple reports have suggested that AMPK and dietary restriction promote longevity[61,62], and treatment with an AMPK activator alleviates HGPS phenotypes[63,64]. Our data show p300 shuttling into the nucleus is dependent on AMPK activation, which phosphorylates p300. AMPK activation induced more nuclear localization of p300 and reduced raptor acetylation to inhibit mTORC1 (Extended Data Fig. 6b–d). In addition, AMPK activation rescued HGPS phenotypes (Fig. 7c,e,g,h). However, in p300 dNLS mutant-expressing HGPS cells, AMPK activation could not restore reduced H3K9me3 levels (Fig. 7h), suggesting AMPK-mediated rescue of HGPS phenotypes might be associated with p300 localization.

Our data suggest that increased cytoplasmic and decreased nuclear p300 localization contribute to many cellular phenotypes characteristic of HGPS. Our exploration of possible effects has not been exhaustive, as key phenotypes such as mTORC1 hyperactivation seem to be compelling drivers of HGPS pathology. However, our data suggest that there will be abnormal histone acetylation in HGPS and that this may also contribute to its phenotypes. Previously, Zmpste24-deficient mice, used as models to study HGPS[65], showed histone H4 lysine 16 hypoacetylation, causing defective DNA repair and premature senescence[66], consistent with our model. This may be an area deserving of further investigation in future studies.

In conclusion, our study illustrates that the ability of p300 to shuttle in and out of the nucleus is a critical determinant of mTORC1–autophagy response on nutrients, and the regulation of p300 shuttling could be a useful therapeutic target to treat HGPS (Fig. 7i).

## Online content

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

## Methods

Our research complies with all relevant ethical regulations and guidelines. Mouse studies and procedures were performed in accordance with the United Kingdom Animals (Scientific Procedures) Act with appropriate Home Office project and personal animal licences, and with the approval of the University of Cambridge Animal welfare and Ethical Review Body.

### Cell culture

Human cervical epithelium HeLa (CCL-2; CVCL_0030; American Type Culture Collection (ATCC)), human neuroblastoma SH-SY5Y (94030304; European Collection of Authenticated Cell Cultures (ECACC)), human embryonic kidney cell line HEK293 (85120602; ECACC), human retinal pigment epithelium RPE1 (CRL-4000; ATCC), human breast cancer cell line MCF7 (HTB-22; ATCC), human lung carcinoma A549 cells (kindly provided by F. Buss (University of Cambridge)) and human neuroglioma H4 cells (HTB-148; ATCC) were cultured in Dulbecco's modified Eagle's medium (DMEM) (4.5 g l$^{-1}$ of glucose; catalogue no. D6546)) supplemented with 10% fetal bovine serum (FBS) (catalogue no. F7524), 2 mM $_L$-glutamine (catalogue no. G7513), 100 U ml$^{-1}$ penicillin and 100 mg ml$^{-1}$ streptomycin (catalogue no. P0781). Human hepatoma HepG2 cells (85011430; ECACC) were grown in Roswell Park Memorial Institute-1640 medium (catalogue no. R0883; Sigma-Aldrich). Human mammary epithelium MCF10A cells were purchased from Horizon (catalogue no. HD PAR-058) and cultured in DMEM–F12 supplemented with 5% horse serum (catalogue no. H1270), 20 ng ml$^{-1}$ human epidermal growth factor (catalogue no. E9644), 0.5 µg ml$^{-1}$ hydrocortisone (catalogue no. H0135), 100 ng ml$^{-1}$ cholera toxin (catalogue no. C8052), 10 µg ml$^{-1}$ insulin (catalogue no. I9278), 100 U ml$^{-1}$ penicillin and 100 mg ml$^{-1}$ streptomycin (all the components were obtained from Sigma-Aldrich), as described previously[8]. AMPK α1/α2 double KO (dKO) MEFs were a gift from B. Viollet (Universite Paris Descartes). Primary human dermal fibroblasts from patients with HGPS and healthy donors (Supplementary Table 1; all fibroblasts used in this study were obtained from Coriell Institute for Medical Research Biobank (https://www.coriell.org/) in compliance with their regulations) were cultured in DMEM (Invitrogen) supplemented with 20% FBS (Invitrogen) and antibiotics. Control #1 and #2 represent stocks from different early passages from the original dermal fibroblasts (identified as GM05565). For this study, we used LKB1 (catalogue no. 8590; Addgene)-expressing HeLa cells unless otherwise stated[40], and HeLa cells stably expressing mRFP–GFP–LC3 (kind gift from T. Yoshimori, Osaka University) were cultured in the presence of G418 at 500 µg ml. All the cell lines were maintained at 37 °C and 5% CO$_2$ and were regularly tested for mycoplasma contamination. For AA starvation, cells were washed twice with phosphate-buffered saline (PBS) and incubated for 1 h at 37 °C in AA-free medium (catalogue no. D9800-13; US Biological) containing 10% dialysed FBS (catalogue no. 26400-036; Invitrogen), unless otherwise indicated. For Glc starvation, cells were washed twice with PBS and cultured in DMEM free from Glc (catalogue no. A1443001; Gibco), 1 mM sodium pyruvate and 2 mM $_L$-glutamine for 1 h. For nutrient depletion of cells, we used Hank's balanced salt solution (catalogue no. 14025-092; Invitrogen) for 1 h. The cells were stimulated with 2× concentration (for AAs) or 1× (for Glc) compared with what is typically found in normal high-Glc DMEM (catalogue no. D6546; Invitrogen).

### Primary MEFs

Primary MEFs were isolated from WT C57BL/6 mouse embryos at E12, using a protocol modified from that described previously (www.thermofisher.com/). The sexes of mouse-derived primary cells were not determined, as this was not considered to impact the basic cell biology we are describing. All mouse experiments were performed with personal and project licences granted by the United Kingdom Home Office and with the approval of the University of Cambridge committee for animal studies.

### p300 KO cells

p300 KO HeLa and SH-SY5Y cells were generated with the CRISPR–Cas9 technology. Cells were transiently cotransfected using Trans IT-2020 reagent (catalogue no. MIR5400; Mirus) according to the manufacturer's instructions. The Lenti-PB (pKLV–PB–U6gRNA(BbsI)–PGKpuro-2ABFP) vector carrying two guide RNAs (gRNAs) was mixed with the LentiCas9blast vector in a molar ratio 1:3 (all constructs kind gifts from E. Metzakopian, UK Dementia Research Institute[67]). Transfected cells were treated with 3 µg ml$^{-1}$ puromycin (catalogue no. A11138-03; Gibco) to select for gRNA-expressing cells and then single sorted. Clones were selected based on p300 levels, compared with non-targeting gRNA-transfected cells.

Two gRNA sequence against p300 are below:
gRNA #1: sense 5′-CACCGTAGAGTTGATTAATTCATCGT-3′
antisense 5′-TAAAACGATGAATTAATCAACTCTAC-3′
gRNA #2: sense 5′-CACCGTGATTAATATCACCACCATGT-3′
antisense 5′-TAAAACATGGTGGTGATATTAATCAC-3′

### Generation of Dox-inducible cell lines

Inducible cell lines were generated as follows: lentiviral plasmids plenti_CMV_Hygro_R55M3 (RtTA3), plenti_Neo_CMV_TRE3G_GFP-Lamin A and plenti_Neo_CMV_TRE3G_GFP-Progerin (all generated by T. Misteli, National Institutes of Health (NIH) and obtained from Addgene). For lentiviral production, these plasmids were transfected into HEK293T cells using TransilT-2020 (Mirus Bio) together with the packaging plasmids pCMVti8.91 and pMD.G. RPE1 cells were then transduced with the two types of lentiviral particles (containing either the plasmid that encodes the constitutive tetracycline repressor A3 mutant or the GFP–lamin/progerin construct) and subjected to antibiotic selection (hygromycin at 200 units ml$^{-1}$ and G418 at 400 units ml$^{-1}$). Expression of the GFP constructs was induced with 1 µg ml$^{-1}$ Dox for 48 h.

### Animal studies

Mice were housed in individually ventilated cages with free access to standard animal food chow (catalogue no. R105; SAFE) and water, in a climate-controlled room with a 12 h light/dark cycle, except when subjected to starvation–re-feeding protocols specifically mentioned below. As previously described[8], we used 6–7-week-old C57BL/6 male or female mice for food deprivation for the 48 h fasting experiments with a 1:1 sex ratio. The number of mice used are indicated for each experiment in the figure legends (in general, $n = 6$). For 24 h fasting/re-fed experiments, C57BL/6 J male and female mice (age 10 weeks) were deprived of food for 24 h, followed by 2 h feeding period (for re-feeding group) or not (for fasting group). After 2 h re-feeding, all tissues were collected from the fed, fasting and re-fed after fasting groups. No inclusion or exclusion criteria were used.

### Antibodies and reagents

The following antibodies were used in this work: mouse anti-FLAG M2 (#F3165), rabbit anti-actin (#A2066), rabbit anti-HA tag (#AP1012A), mouse anti-progerin (#05-1231) and mouse anti-α-tubulin (#T9026) from Sigma-Aldrich; mouse anti-GAPDH clone 6C5 (#ab8245), rabbit anti-p300 (#ab10485), rabbit anti-phospho-p300 (Ser89; #ab135554), rabbit anti-CBP (#ab137334), rabbit anti-GFP tag (#ab6556), rabbit anti-H2B (acK16; #ab177427), rabbit anti-H4 (#ab7311), rabbit anti-LAMP1 (#ab24170), mouse anti-LAMP1 (#ab25630), rabbit anti-acetylated-lysine (Ac-K) (#ab21623), mouse anti-6X His-tag (#ab18184), rabbit anti-lamin B1 (#ab16048), rabbit anti-Atg7 (#ab133528), mouse anti-Myc tag (#ab32), rabbit anti-14-3-3ζ (#ab51129), rabbit anti-histone H3 (tri-methyl K9; #ab8898), mouse anti-BAG6 (#ab88292) and rabbit anti-ALX1 (#ab181101) from Abcam; rabbit anti-p300 (#sc-585, #sc-48343), goat anti-lamin B (#sc-6217), rabbit anti-p-PP2Ac (#sc-271903), mouse anti-PP1γ (#sc-515943), mouse anti-CRM1 (#sc-74454) and mouse anti-TOM20 (#sc-17764) from SantaCruz Biotechnology; mouse anti-GFP (#632375 and #632592;

Clontech), mouse anti-HA.11 clone 16B12 (#MMS-101P; Covance), mouse anti-p300 (#05-257; Millipore), mouse anti-acetylated-lysine (Ac-K) (#AAC01; Cytoskeleton Inc.), mouse anti-GFP tag (#66002-1-Ig), rabbit anti-Lamin A/C (#10298-1-AP), rabbit anti-Lamin B1 (#12987-1-AP) and mouse anti-mTOR (#66888-1-Ig) from Proteintech; rabbit anti-p300 (#86377), rabbit anti-CBP (#7389), rabbit anti-acetyl-CBP (Lys1535)/p300 (Lys1499) (#4771), rabbit anti-KAT2A (#3305), rabbit anti-KAT2B (#3378), rabbit anti-hVps34 (#4263), rabbit anti-histone H2B (#12364), rabbit anti-histone H3 (#9715), rabbit anti-H3 (acK9; #9649), rabbit anti-H3 (acK56; #4243), rabbit anti-H4 (acK12; #2591), rabbit anti-H4 (acK16; #13534), rabbit anti-LAMP1 (#9091), rabbit anti-mTOR (#2972, #2983), rabbit anti-raptor (#2280), rabbit anti-phospho-S6K1 (Thr389; #9234), anti-total S6K1 (#9202), rabbit anti-phospho-S6 ribosomal protein (p-S6) (Ser235/236; #4856), rabbit anti-S6 ribosomal protein (S6) (#2217), rabbit anti-phospho-4E-BP1 (Thr37/46; #9459), rabbit anti-4E-BP1 (#9452), rabbit anti-Ac-K (#9814, #9441), rabbit anti-PP2Ac (#2259, #2038), rabbit anti-phospho-ACC (Ser79; #11818), rabbit anti-ACC (#3676), rabbit anti-ACC2 (#8578), rabbit anti-phospho-AMPKα (Thr172; #2531), rabbit anti-AMPKα (#2532) and rabbit anti-phospho-histone H2A.X (Ser139; #2577, #9718) from Cell Signaling Technology; anti-mouse (#NA931V) and anti-rabbit (#NA934V) horseradish peroxidise-conjugated secondary antibodies (GE Healthcare); and anti-goat horseradish peroxidise-conjugated secondary antibody (#611620; Invitrogen/Life Technologies). All primary antibodies were used at a dilution between 1:500 and 1:1,000 (overnight incubation at 4 °C), and the secondary antibodies used at a dilution of 1:5,000 (1 h of incubation at room temperature).

Drug treatments include dimethylsulfoxide, 10 µM c646 (catalogue no. SML0002), 50 µM CTB (catalogue no. C6499), 10 mM nicotinamide (NAM) (catalogue no. N0636), 2 µM trichostatin A (TSA) (catalogue no. T8552), 10 mM sodium butyrate (catalogue no. B5887), Comp.C (catalogue no. P5499), 10 µM LY294002 (catalogue no. L9908), 5 µM cantharidin (catalogue no. C7632), 1 µM okadaic acid (OA) sodium salt (catalogue no. 459620), $_D$-(+)-Glc (catalogue no. 06-758) from Sigma-Aldrich; 0.2–1 µM Torin1 (catalogue no. 4247), 2 µM A485 (catalogue no. 6387), 10 µM U0126 (catalogue no. 1144), 0.5 µM Go6983 (catalogue no. 2285) from Tocris Bioscience; 5 µM Comp.C (catalogue no. 171260) from Merck; 10 µM 991 (catalogue no. AOB8150) from AOBIOUS Inc.; 500 µg ml$^{-1}$ Geneticin Selective Antibiotic (G418; catalogue no. 11811-031) from Gibco; 400 nM BAF from Enzo Life Sciences; 0.1–1 µM SEL (KPT-330, SEL; catalogue no. S7252) from Selleck Chemicals; MEM AAs solution (catalogue no. 11130-051), nonessential AAs solution (catalogue no. 11140-050) from Thermo Scientific.

## Transfection
Trans IT-2020 reagent (catalogue no. MIR5400; Mirus) was used for DNA transfection of most cell lines, while GeneXPlus reagent (catalogue no. ACS-4004; ATCC) was used for SH-SY5Y cells, and TransIT-X2 reagent (catalogue no. MIR6003; Mirus) was used for MEFs, according to the manufacturer's instructions. Lipofectamine 2000 (catalogue no. 11668) or Lipofectamine RNAiMAX (catalogue no. 13778) (Invitrogen) were used for siRNA transfections, according to the manufacturer's instructions. For knockdown experiments, cells were transfected with 20–50 nM siRNA followed by another 20–50 nM siRNA transfection after 48 h. Cells were split once between both transfections, and collected 3 days post-transfection. The following DNA or siRNA/shRNA constructs were also used: empty pEGFP from Clontech; pcDNA3.1–myc–6XHis from Invitrogen; pcDNA3.1–p300 (catalogue no. 23252), pcDNA3.1–p300 DN (histone acetyltransferase (HAT)–) (catalogue no. 23254), pCMVβ–p300–myc (catalogue no. 30489), pSG5–HA–p300 (catalogue no. 89094), pRK5–HA–YFP–raptor (catalogue no. 73385), pRK5–HA–raptor (catalogue no. 8513), pcDNA3–AMPKα2 WT (catalogue no. 15991), pcDNA3–AMPKα2 K45R (catalogue no. 15992), pcDNA3–FLAG–LKB1 (catalogue no. 8590), pcDNA3.1–Myc–14-3-3ζ (catalogue no. 48798), pcDNA3.1–HA–14-3-3ε (catalogue no. 48797),

lentiviral-S-tdTomato (catalogue no. 112579), pRK5–HA–GST–RagB (catalogue no. 19301) and pLJM1–FLAG–RagD (catalogue no. 19316) from Addgene. Predesigned siRNAs (SMARTpool oligos ON-TARGET plus four non-targeting controls: #D-001810-10, p300 #L-003486-00-0005, CBP #L-003477-00-0005, 14-3-3ζ #L-003332-00-0005, 14-3-3ε #L-017302-02-0005, BAG6 #L-005062-01-0005, ALX1 #L-012333-02-0005, KAT2A #L-005055-00-0005, KAT2B #L-009722-02-0005, PP2Ac #L-003598-01-0005 and PP1 #L-006827-00-0005) were obtained from Dharmacon–Thermo Scientific. For knockdown of AMPKα1/2, we used siRNA from SantaCruz Biotechnology (#sc-45312).

## Western blot analysis
Cells were washed with ice-cold PBS and directly lysed with 2× Laemmli buffer and boiled at 100 °C for 10 min or lysed with radioimmuno-precipitation assay buffer (50 mM Tris–HCl pH 7.4, 150 mM NaCl, 1% NP-40, 0.5% sodium deoxycholate monohydrate and 0.1% sodium dodecyl-sulfate (SDS), supplemented with protease and phosphatase inhibitor cocktails (Roche)). When lysed in radioimmunoprecipitation assay buffer, cells were incubated on ice for 10 min, centrifuged at 16,100$g$ for 10 min and protein concentrations of supernatants were determined using a Bradford assay kit (Bio-Rad). Lysates were then denatured with 2× Laemmli buffer and boiled at 100 °C for 10 min, separated by SDS–polyacrylamide gel electrophoresis (PAGE), transferred onto polyvinylidene difluoride membranes, subjected to western blot analysis and visualized using an ECL enhanced chemiluminescence detection kit (GE Healthcare), or with direct infra-red fluorescence detection on an Odyssey Infrared Imaging System. Densitometric analysis of immunoblots was performed using the ImageJ program or IMAGE STUDIO Lite software.

## Mutagenesis
Mutagenesis of p300 was performed using the QuikChange Site-Directed Mutagenesis Kit (Agilent Stratagene), according to the manufacturer's instructions. Mutagenesis primers were designed using the web-based QuikChange Primer Design program (Agilent Technologies):

1.  5′-ccggagatgagagcggcccccggttcc-3′ (forward), 5′-ggaaccggggccgctctcatctccgg-3′ (reverse) to delete the residues from Phe11 to Lys17 (11PSAKRPK17) (p300 dNLS)
2.  5′-catattgaggttaggggaagcaccagatcgcagcaattct-3′ (forward), 5′-agaattgctgcgatctggtgcttcccctaacctcaatatg-3′ (reverse) to convert the residue Ser89 into Ala (p300 S89A)
3.  5′-ccatattgaggttaggggaatcaccagatcgcagcaattctg-3′ (forward), 5′-cagaattgctgcgatctggtgattcccctaacctcaatatgg-3′ (reverse) to convert the residue Ser89 into Asp (p300 S89D)
4.  5′-aaggccactggctccagcctgctgtccaggattc-3′ (forward), 5′-gaatcctggacagcaggctggagccagtggcctt-3′ (reverse) to convert the residue Ile into Ala (p300 IA)
5.  5′-ctgaatctggagaccagcgccactggctccaatc-3′ (forward), 5′-gattggagccagtggcgctggtctccagattcag-3′ (reverse) to convert the residue Leu into Ala (p300 L270A)
6.  5′-ttttgtctgaatctgggcaccaaggccactggctcc-3′ (forward), 5′-ggagccagtggccttggtgcccagattcagacaaaa-3′ (reverse) to convert the residue Leu into Ala (p300 LA2)

DPNI digestion was performed after PCR and XL-10 gold-competent cells were transformed and, through sequencing, the mutants from a positive clone were selected. All DNA constructs were verified by sequencing.

## Immunofluorescence
For immunofluorescence, cells were fixed for 5 min with ice-cold methanol or for 10 min with 4% paraformaldehyde. Antibody concentrations are described below. The mounting solution was from Molecular Probes.

Dilution of primary antibodies. 1:300 rabbit anti-mTOR, 1:600 mouse anti-LAMP1, 1:300 mouse anti-p300, 1:300 rabbit anti-p300, 1:200 rabbit anti-phospho-p300, 1:400 rabbit anti-14-3-3ζ, 1:400 mouse anti-BAG6, 1:300 mouse anti-ALX1, 1:200 mouse anti-GFP, 1:100 mouse anti-CRM1, 1:100 mouse anti-RAN, 1:100 mouse anti-TOM20, 1:200 mouse anti-LC3, 1:300 rabbit anti-lamin B1, 1:200 rabbit anti-p-histone H2A.X (S139), 1:200 rabbit anti-histone H3 (tri-methyl K9), 1:200 rabbit anti-lamin A/C, 1:200 rabbit anti-p-S6 (S235/236) and 1:1,000 mouse anti-HA.11 clone 16B12.

The secondary antibodies Alexa 488, 555, 568, 594 or 647 goat anti-mouse, goat anti-rabbit or rabbit anti-goat were obtained from Molecular Probes and used at 1:400. Imaging was conducted with LSM710, LSM780 or LSM880 Zeiss confocal with a ×63 oil-immersion lens. Colocalization was measured in a blinded manner using Volocity software for Mander's Overlap Coefficient (MOC).

### Live-cell imaging for monitoring nucleocytoplasmic shuttling

Nucleocytoplasmic shuttling using the NLS–TdTomato–NES construct was assessed as described previously[52]. RPE1 cells with inducible GFP–lamin A/C and GFP–progerin expression were transfected with the NLS–TdTomato–NES construct (provided by J. Rothstein; catalogue no. 112579; Addgene). After a 48 h incubation, cells were plated onto MatTek Petri dishes (MatTek). Following 48 h Dox treatment, they were imaged at 37 °C, utilizing an LSM780 Zeiss confocal equipped with a ×63 oil-immersion lens.

### Autophagy flux assay

Autophagy flux was determined with the SRAI–hLC3B construct, as previously described[27]. Both control and p300 KO cells were transfected with the SRAI–hLC3B construct. Following 48 h incubation, cells were fixed using 4% paraformaldehyde for 5 min, and coverslips were mounted with ProLong Gold Antifade Reagent (without 4,6-diamidino-2-phenylindole (DAPI); catalogue no. P10144; Molecular Probes). The mRFP–GFP–LC3 stably expressing HeLa cells were imaged on at least 20 fields, and analysed with Zen software. The numbers of yellow vesicles dots (GFP-positive and mRFP-positive dots; autophagosomes) and red-only vesicles (mRFP-only dots; autolysosomes) were counted in a blinded fashion using Zeiss LSM780 or LSM880 confocal microscopes and quantified using ImageJ.

### Cytosolic/nuclear fractionation

Cells were washed twice with ice-cold PBS and lysed with buffer A (10 mM 4-(2-hydroxyethyl)-1-piperazineethanesulfonic acid (HEPES), 10 mM KCl, 0.1 mM ethylenediaminetetraacetic acid (EDTA), 0.4% NP-40, 1 mM dithiothreitol (DTT) and protease/phosphatase inhibitors cocktail) and incubated on ice for 20 min. After homogenization, lysates were centrifuged at 16,100$g$ at 4 °C for 10 min. Supernatants containing cytosolic proteins were collected and nuclear pellets were resuspended with buffer B (20 mM HEPES, 0.4 M NaCl, 1 mM EDTA, 10% glycerol, 1 mM DTT and protease/phosphatase inhibitors cocktail) and incubated for 1 h on ice. After centrifugation at 16,100$g$ for 10 min at 4 °C, supernatants containing the nuclear proteins were collected. Protein concentration was determined using the Bicinchoninic Acid Protein Assay Kit (catalogue no. 23227; Pierce). Histone H3 or lamin B1 were used as nuclear controls, and GAPDH or S6 were used as cytosolic controls.

### Mitochondrial morphology analysis

Mitochondrial morphology was investigated using TOM20 antibody (sc-17764; 1:100). Images were captured with a LSM880 Zeiss confocal with a ×63 oil-immersion lens and quantified using the ImageJ program. These procedures were performed in a blinded fashion.

### p300 acetyltransferase activity assay

p300 activity was determined using the SensoLyte EP300 assay kit (AS-72172, Anaspec) with some modifications. Briefly, after cytosolic p300 was immunoprecipitated from cells, the lysates were incubated with AcCoA solution and substrates (H3 or p53) for 15 min at 37 °C. Developer solution was added and incubated for 30 min at room temperature. After the reaction was stopped with stop solution, the fluorescence was measured using a Spark multimode microplate reader (TECAN Trading AG) with the following settings: $\lambda_{ex}$ 389 nm and $\lambda_{em}$ 513 nm.

### PP2A phosphatase kinetics assay

PP2A phosphatase activity was measured as previously described[68] using a serine–threonine phosphatase assay kit (catalogue no. 17–127; Sigma-Aldrich) with some modifications. For PP2A immunoprecipitation, rabbit antibodies against PP2Ac (catalogue no. 2038; Cell Signaling Technology) were added to a total or nuclear fraction of 100 μg per 300 μl of lysate, followed by Dynabeads-protein G (Life Technologies) and incubation for 1–2 h at 4 °C. The beads were washed twice with PBS, followed by a single wash in assay buffer before the phosphopeptide was added to a final concentration of 0.75 mM and incubated for 10 min at 30 °C. For statistical analysis, three independent enzyme assays were performed and fold changes at various timepoints were measured as the average absorbance value.

### Measurement of Glc uptake

Intracellular Glc uptake was measured using cell lysates with a Glc uptake assay kit (catalogue no. ab136955; Abcam) according to the manufacturer's instruction. Insulin (1 μM for 20 min) was used to activate Glc transport. Relative Glc uptake was determined using a standard curve.

### Co-immunoprecipitation

Cells in 100 mm or 140 mm dishes were washed twice with PBS and lysed in ice-cold lysis buffer (40 mM HEPES (pH 7.4), 2 mM EDTA, 10 mM pyrophosphate, 10 mM glycerophosphate and 0.3% CHAPS (3-((3-cholamidopropyl)-dimethylammonio)-1-propanesulfonate) or 0.5% Triton X-100 and protease inhibitors (Roche) and phosphatase inhibitors (Sigma-Aldrich), and further supplemented with 10 mM sodium butyrate and 1 mM TSA for immunoprecipitation to detect acetylated proteins. Lysates were incubated on ice for 20 min and isolated by centrifugation at 16,100$g$ for 10 min. Supernatants were transferred to new tubes, and 1/10 of the sample was kept as input control, while the remaining lysate was overnight incubated with primary antibodies at 4 °C with gentle agitation. Thereafter, Dynabeads-protein G (Life Technologies) were added to the samples and incubated at 4 °C for 2 h. The beads were washed more than three times with lysis buffer and the immunoprecipitated proteins were eluted and denatured with 2× Laemmli buffer and boiled for 10 min at 100 °C, separated by SDS–PAGE. The immunoprecipitation of Myc-, FLAG- or HA-fusion proteins was performed using Myc-Trap (catalogue no. ytma-100; ChromoTek), FLAG-magnetic beads (catalogue no. M8823; Millipore) or HA-magnetic beads (catalogue no. 88836; Thermo Scientific), according to the manufacturer's instructions, with some modifications. Briefly, cells in 100 mm dishes were lysed in ice-cold lysis buffer (10 mM Tris–HCl, pH 7.4, 150 mM NaCl, 0.5 mM EDTA and 0.3% CHAPS and protease/phosphatase inhibitors cocktail). Then, 0.6 ml of cell lysate was incubated for 1 h at 4 °C with 20 μl of Myc-Trap, FLAG- or HA-magnetic beads, and then the beads were washed twice with the wash buffer (10 mM Tris–HCl, pH 7.4, 150 mM NaCl and 0.5 mM EDTA). Next, 50 μl of 2× Laemmli buffer was added and boiled for 10 min at 100 °C, then separated by SDS–PAGE. To determine raptor acetylation, cells in 100 mm dishes were washed twice with PBS and lysed in ice-cold lysis buffer (40 mM HEPES pH 7.4, 2 mM EDTA, 10 mM pyrophosphate, 10 mM glycerophosphate and 0.3% CHAPS and protease/phosphatase inhibitors cocktail), and further supplemented with 10 mM sodium butyrate and 1 mM TSA.

### Image analysis

Volocity software (PerkinElmer) was used for analysis and processing of confocal images. For colocalization analysis of confocal images, we

used MOC. At least 30 cells were examined per condition. All experiments were repeated at least three times. The background was fixed for all within-experiment analyses.

## Statistics and reproducibility
Significance levels were determined using GraphPad Prism 7 (GraphPad Software) or Excel (Microsoft office). Western blot protein levels were normalized to total forms or a housekeeping protein, such as tubulin. All data were expressed as means ± s.e.m. or s.d., as stated in figure legends. $P$ values of <0.05 were considered statistically significant. Statistical analysis was performed using Student's $t$-tests or one-way or two-way analysis of variance (ANOVA) followed by the appropriate post hoc test for multiple comparisons as detailed in the figure legends and source data.

Sample sizes were chosen on the basis of extensive experience with the assays we have performed. All in vivo experiments and tests were randomly assigned, but no randomization was performed for cell culture experiments. No data were excluded from the analyses. Staining and analysis were performed in a blinded fashion. Investigators were not blinded during the other experiments.

## Reporting summary
Further information on research design is available in the Nature Portfolio Reporting Summary linked to this article.

## Data availability
All data supporting the findings of this study are available from the corresponding author upon reasonable request. Source data are provided with this paper.

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

## Acknowledgements

We thank F. Siddiqi and L. Wrobel for help with mouse samples. We thank E. Stamatakou for advice and technical support for KO cell lines using CRISPR–Cas9 editing. We thank A. Djajadikerta for making the SRAI–hLC3B construct. We are grateful for funding from UK Dementia Research Institute (funded by the Medical Research Council, Alzheimer's Research UK and the Alzheimer's Society) and Parkinson's UK (D.C.R.). D.L. and S.Y.B. were funded by a Sir Henry Dale Fellowship jointly funded by the Wellcome Trust and the Royal Society grant no. 206242/Z/17/Z. This research was also supported by the NIHR Cambridge Biomedical Research Centre (grant no. NIHR203312). The views expressed are those of the authors and not necessarily those of the NIHR or the Department of Health and Social Care.

## Author contributions

S.M.S. and D.C.R. developed the study rationale. S.M.S. and D.C.R. wrote the manuscript, which was commented on by all authors. S.M.S. designed and performed most of the experiments. S.J.P. performed experiments and assisted with data analysis. D.C.R. supervised the study. D.L. contributed reagents and scientific insight. S.Y.B. engineered the inducible lamin A/C and progerin cell lines.

## Competing interests
D.C.R. is a consultant for Aladdin Healthcare Technologies Ltd, Mindrank AI, Nido Biosciences, Drishti Discoveries, Retro Biosciences Inc. and PAQ Therapeutics. D.L. is a cofounder of Adrestia Therapeutics and a scientific advisor for Shift Bioscience and Adrestia Therapeutics. The other authors declare no competing interests.

## Additional information
**Extended data** is available for this paper at https://doi.org/10.1038/s41556-023-01338-y.

**Correspondence and requests for materials** should be addressed to David C. Rubinsztein.

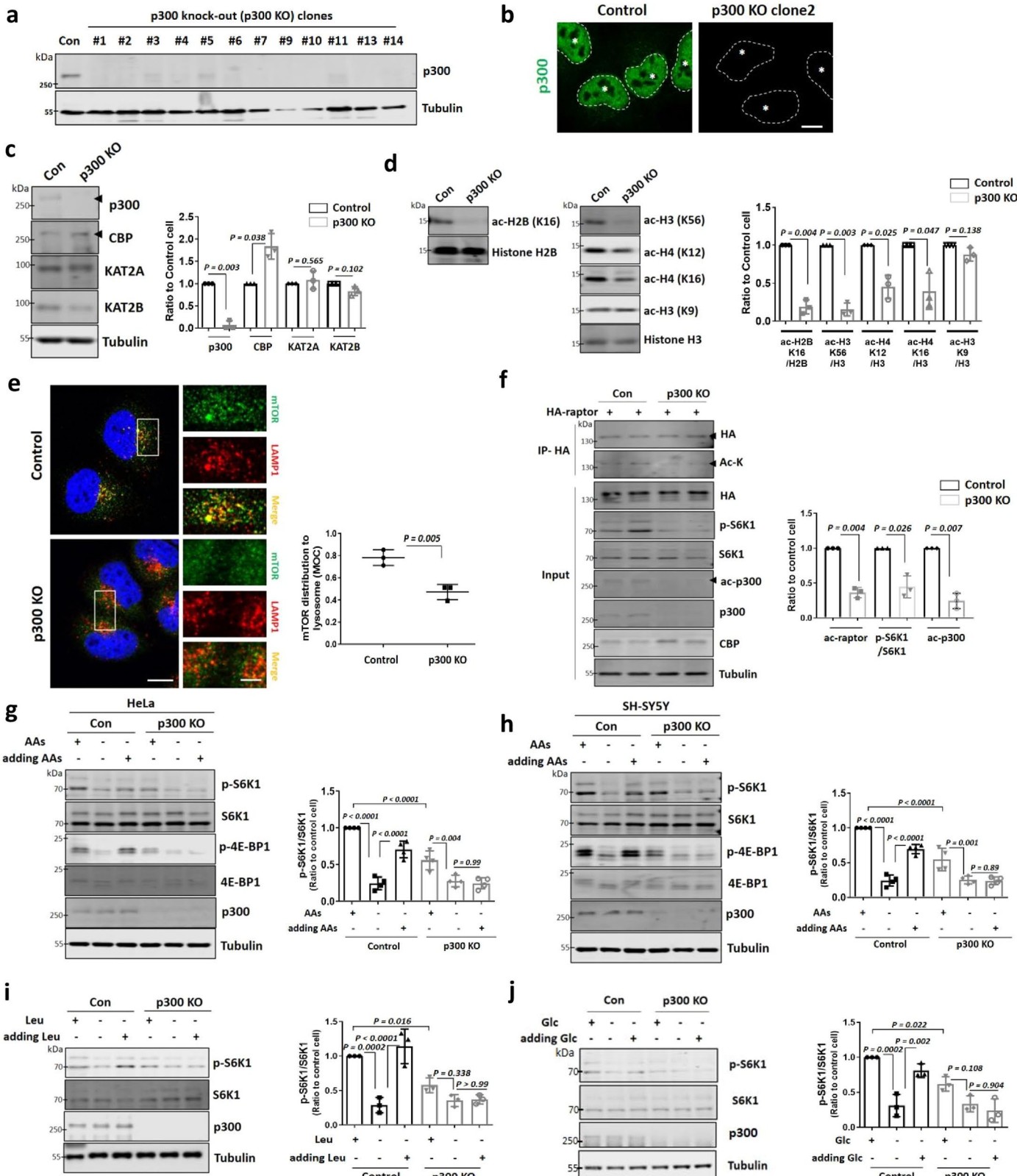

**Extended Data Fig. 1 | See next page for caption.**

**Extended Data Fig. 1 | p300 is a key acetyltransferase for mTORC1 regulation in response to nutrient status. a**, Validation of p300 CRISPR/Cas9 knockout (p300 KO) HeLa cells using anti-p300 antibody. Con is sgRNA non-targeting control. The number reflects labels of individual p300 KO cell clones. We used p300 KO clone #2 for further work unless otherwise specified. Blots are representative of at least 3 independent experiments (N = 3). **b**, Immunostaining with anti-p300 antibody. N = 3, n = about 40 cells analysed per condition per experiment. Scale bar, 5 μm. **c**, Control and p300 KO HeLa cells were used to determine whether p300 affects the expression levels of other KATs. N = 3. Two-tailed paired t test. **d**, Acetylation of histones in p300 KO cells compared to control cells. N = 3. Two-tailed paired t test. **e**, Control and p300 KO HeLa cells were immunostained with mTOR and LAMP1 antibodies as shown. Co-localization panels show an overlap between mTOR and LAMP1 signals. The fraction of mTOR-positive lysosomes was determined using Volocity software. N = 3, n = about 30 cells analysed per condition per experiment (two-tailed

unpaired t test). Scale bar, 5 μm, 1 μm (enlarged images). **f**, Reduced raptor acetylation, mTORC1 (p-S6K1 at Thr389)/S6K1) and p300 activity (acetylated p300 at Lys1499; ac-p300) in p300 KO HeLa cells. Cells were transfected with HA-raptor, and following incubation for 48 h cells were lysed and raptor was immunoprecipitated using an anti-HA antibody. N = 3. One sample t-test. **g,h**, Re-stimulation of AAs to starved cells did not restore mTORC1 (p-S6K1 at Thr389 / S6K1 and p-4E-BP1 (Thr37/46) / 4E-BP1) in p300 KO HeLa (**g**) or SH-SY5Y cells (**h**). Blots are representative of four biologically independent experiments (N = 4). One-way ANOVA with post hoc Tukey test. **i,j**, Re-stimulation of Leu (**i**) or Glc (**j**) to starved cells did not restore mTORC1 in p300 KO HeLa cells. Blots are representative of three biologically independent experiments (N = 3). One-way ANOVA with post hoc Tukey test. Data are presented as mean values +/- s.d. unless otherwise specified. Source numerical data and unprocessed blots are available in source data.

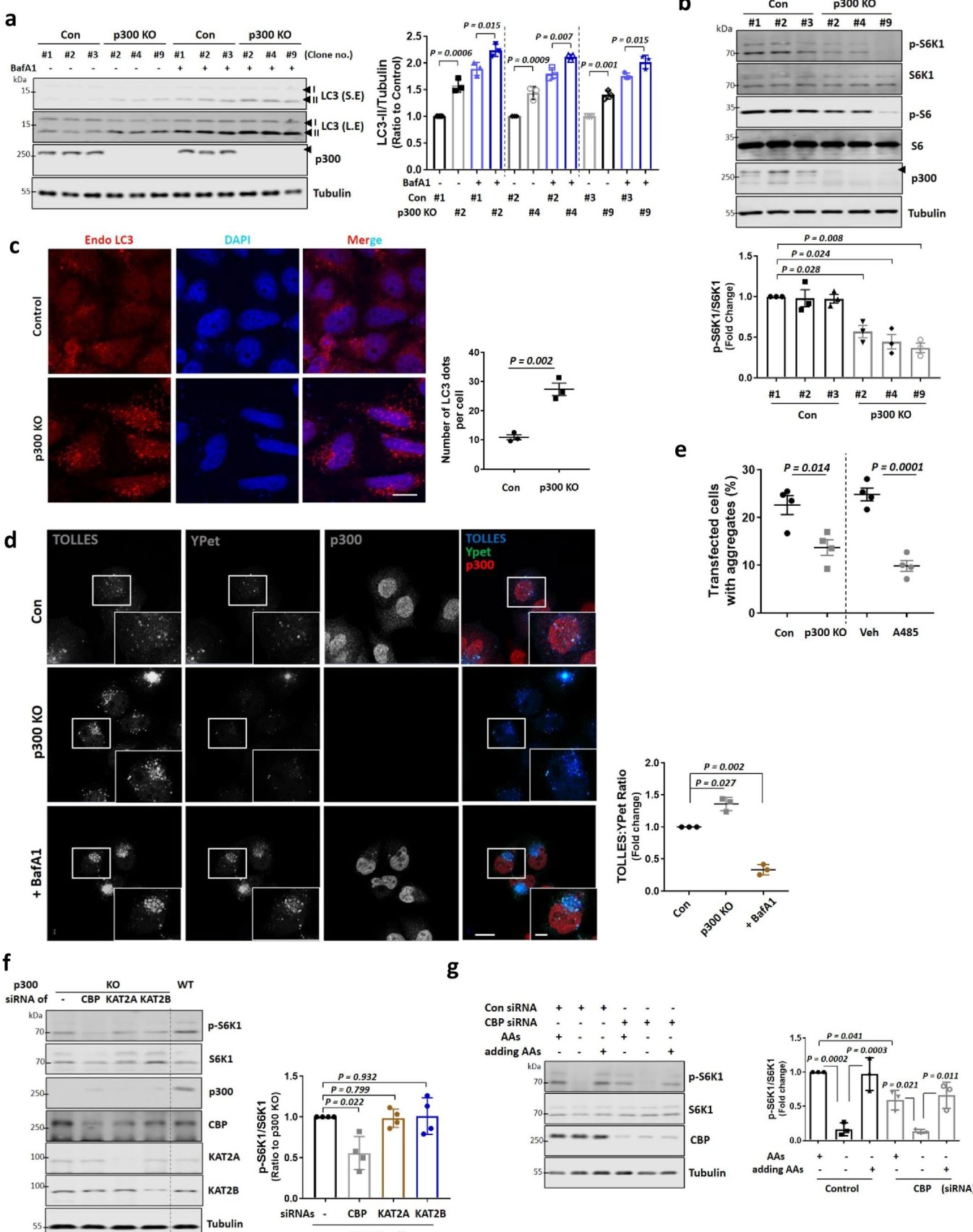

**Extended Data Fig. 2 | See next page for caption.**

**Extended Data Fig. 2 | Autophagy activation in p300 KO cells. a**, Autophagy activation in p300 KO clones (#2, #4 and #9; See Extended Data Fig. 1a). The cells were treated with vehicle (DMSO) or bafilomycin A1 (BafA1) at 400 nM for 4 h. Short exposure (S.E); Long exposure (L.E). Blots are representative of three biologically independent experiments (N = 3). One-way ANOVA with post hoc Tukey test. **b**, Reduced mTORC1 (p-S6K1 at Thr389)/S6K1) in p300 KO clones (#2, #4 and #9; See Extended Data Fig. 1a). N = 3. Two-tailed paired t test. **c**, Immunostaining of control or p300 KO HeLa cells using LC3 antibody, nuclei are stained with DAPI (blue). Scale bar, 5 μm. N = 3; about 70 cells scored per condition per experiment. Two-tailed unpaired t test. **d**, Increase in autophagic flux by p300 KO, as measured by the SRAI-LC3B reporter. Scale bar, 5 μm, 1 μm (enlarged images). N = 3; about 30 cells scored per condition per experiment. Two-tailed paired t test. **e**, Reduced mutant huntingtin aggregates in p300 KO or p300 inhibitor A485 treated HeLa cells. N = 4, 30–50 cells scored per condition per experiment. Two-tailed unpaired t test. **f**, mTORC1 activity in KAT knockdown p300 KO cells. N = 4. Two-tailed paired t test. **g**, Re-stimulation of AAs to starved cells restored mTORC1 in CBP knockdown HeLa cells. N = 3. One-way ANOVA with post hoc Tukey test. Data are presented as mean values +/- s.d. unless otherwise specified. Source numerical data and unprocessed blots are available in source data.

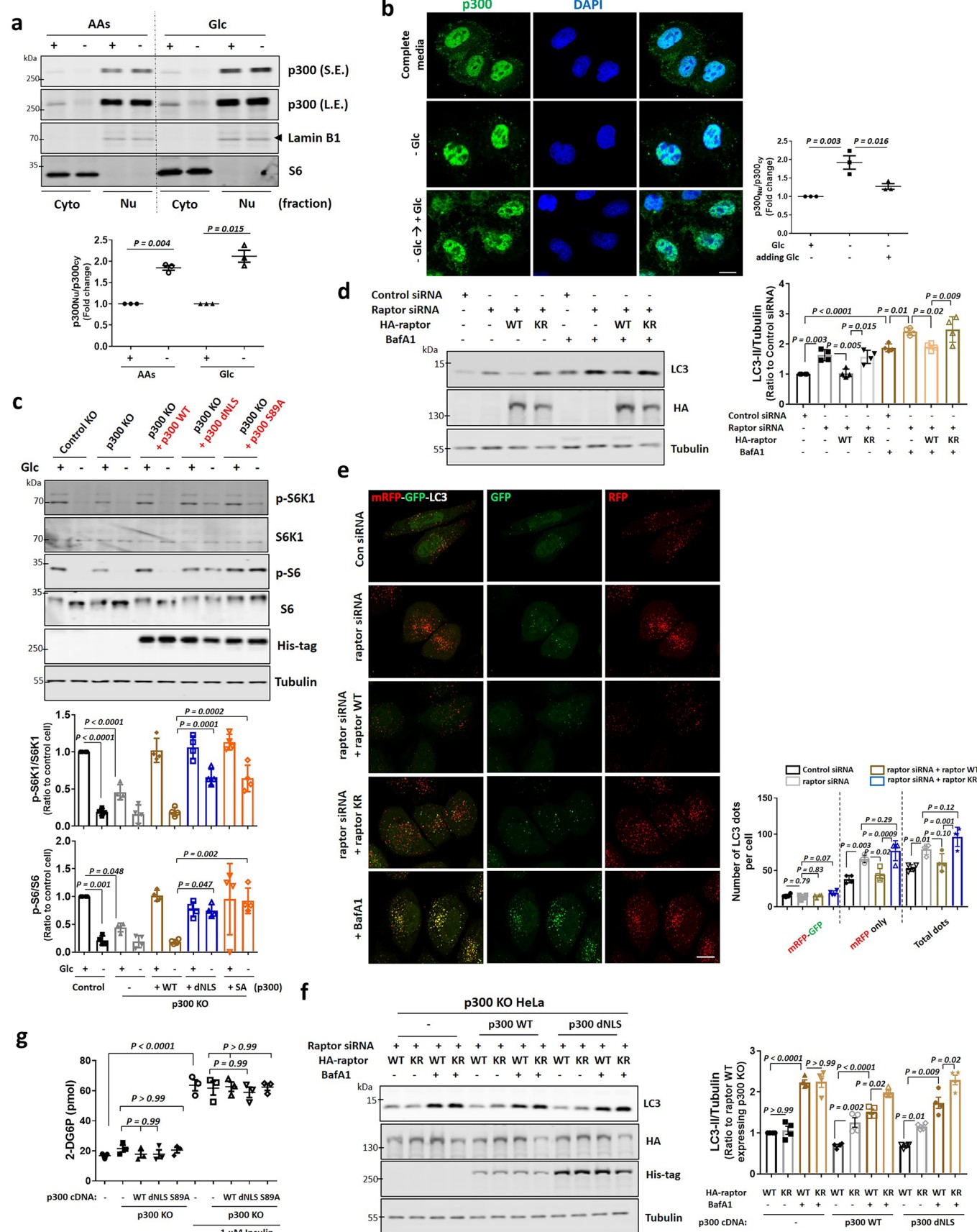

**Extended Data Fig. 3 | See next page for caption.**

**Extended Data Fig. 3 | p300 nuclear transport under glucose starvation.**
**a**, p300 nucleocytoplasmic shuttling in AAs or Glc-depleted media for 1 h using fractionation. Blots are representative of three biologically independent experiments (N = 3). One sample t-test. **b**, p300 nucleocytoplasmic shuttling in Glc-depleted media for 1 h, and then re-addition of Glc to the media for 1 h. N = 3, about 50 cells scored per condition per experiment. One-way ANOVA with post hoc Tukey test. Scale bar, 5 μm. **c**, Remaining mTORC1 activity in His-tagged p300 dNLS or S89A expressing p300 KO cells in Glc-depleted media for 1 h. The expression levels following transfection are depicted by the quantification of the His-tag. N = 4. One-way ANOVA with post hoc Tukey test. **d**, Rescue of autophagy activation in raptor knockdown cells by reconstituting with wild-type (WT) raptor, not KR mutant (K1097R). N = 4. One-way ANOVA with post hoc Tukey test.

**e**, Autophagic flux with HeLa cells stably expressing mRFP-GFP-LC3 reporter. Representative confocal z-stack images (left panel) and total number of mRFP/GFP dots (autophagosomes) and mRFP-only dots (autolysosomes). N = 4; 60-80 cells analysed per condition per experiment. Scale bar, 5 μm. One-way ANOVA with post hoc Tukey test. **f**, Increased LC3-II levels by expressing raptor KR mutant in p300 KO cells reconstituted with p300 WT or dNLS. N = 4. One-way ANOVA with post hoc Tukey test. **g**, Glucose uptake assay in p300 KO cells expressing His-tagged WT, dNLS or S89A. Insulin (1 μM) was used to stimulate glucose uptake into cells. N = 4 in each group. One-way ANOVA with post hoc Tukey test. Data are presented as mean values +/- s.d. unless otherwise specified. Source numerical data and unprocessed blots are available in source data.

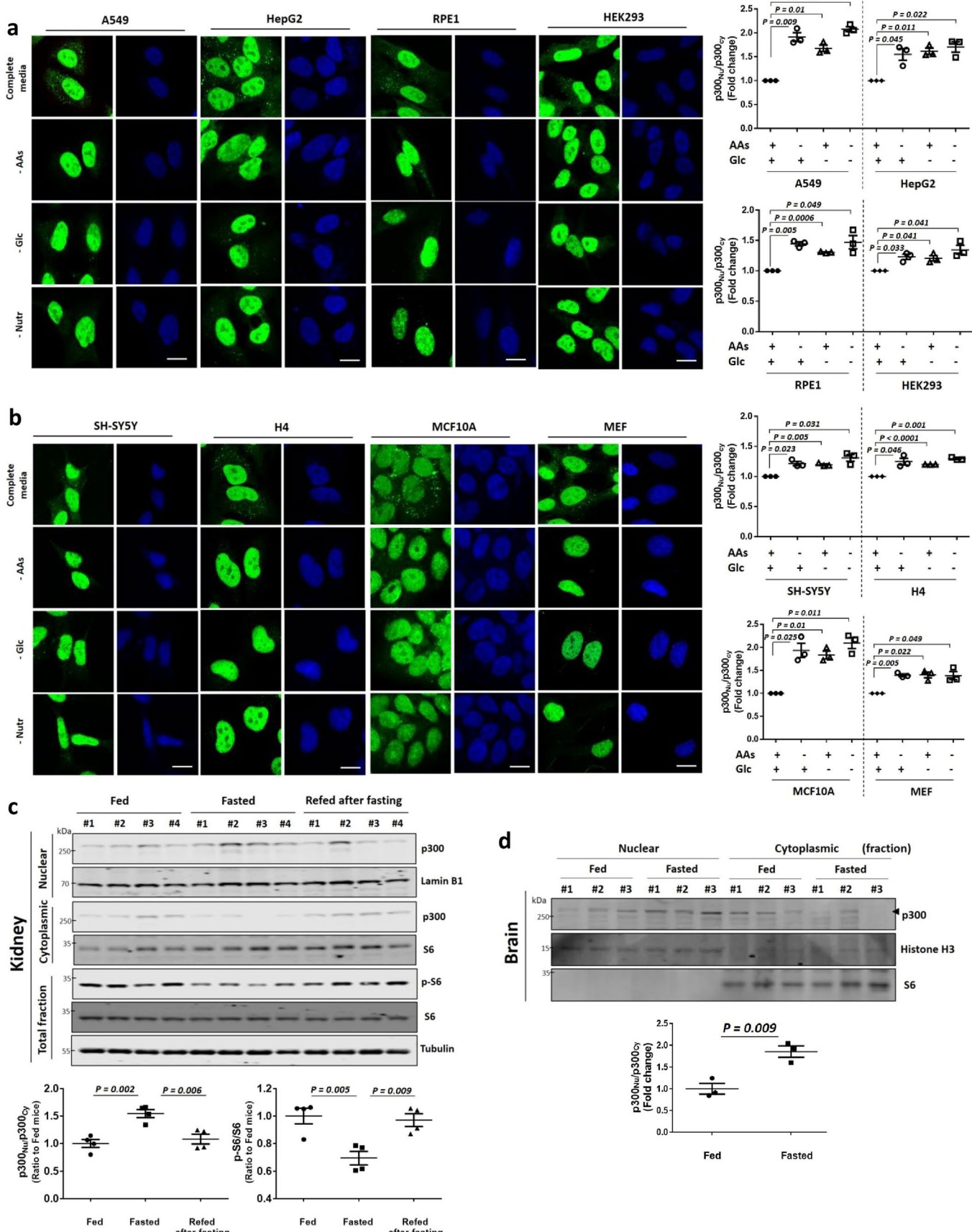

**Extended Data Fig. 4 | See next page for caption.**

**Extended Data Fig. 4 | p300 shuttling depending nutrient status in most cell types and in mice tissues. a**, **b**, p300 nucleus-cytoplasm shuttling in multiple cell lines (A549, HepG2, RPE1 and HEK293 (**a**), SH-SY5Y, H4, MCF10A and MEF (**b**)) in AAs-, Glc- or nutrient -depleted media (no AAs, no serum with low Glc) for 1 h. N = 3, about 30-50 cells scored per condition per experiment. Two-tailed paired t test. Scale bar, 5 μm. **c**, p300 nucleocytoplasmic shuttling in kidney from mice fasted or refed after fasting. After 24 h starvation, mice were given free access to food for 2 h. The tissue samples from fed (n = 4) and fasted mice (n = 4) were analyzed for mTORC1 and p300 levels in nuclear or cytoplasmic fractions. One-way ANOVA with post hoc Tukey test. **d**, Increased nuclear translocation of p300 in fasted mouse brains. After 22.5 h starvation, mice were given free access to food for 1.5 h followed by a second round of starvation for another 22.5 h (since brains respond more slowly to starvation than most peripheral tissues). The tissue samples from fed (n = 3) and fasted mice (n = 3) were analyzed for p300 levels in nuclear or cytoplasmic fraction. Note that fasted mice brains showed increased nuclear transport of p300 by 24 h starvation, however, 2 h refeeding after fasting might be not enough time for mTORC1 re-stimulation in mouse brains (as compared to peripheral tissues). Two-tailed unpaired t test. Data are presented as mean values +/- s.d. unless otherwise specified. Source numerical data and unprocessed blots are available in source data.

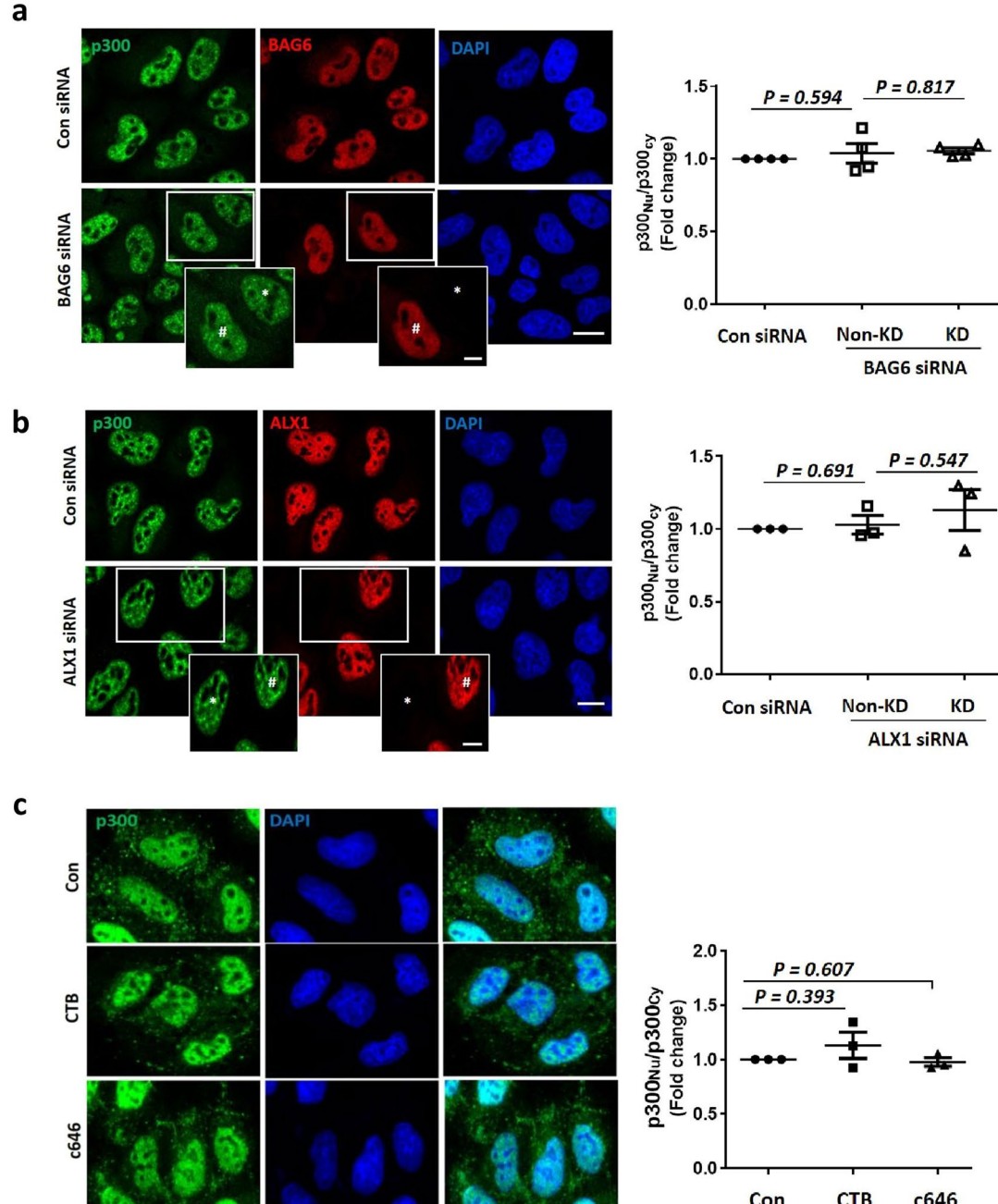

**Extended Data Fig. 5 | p300 shuttling is not mediated by known interactors or their activities. a, b,** No effect in p300 nucleocytoplasmic shuttling by depleting BAG6 (**a**) or ALX1 (**b**) in AAs starved cells. * knockdown (KD) cells, #non-knockdown (Non-KD) cells. N = 4 or N = 3, respectively, about 50 cells scored per condition per experiment. Two-tailed paired t test. Scale bar, 5 μm, 2 μm (enlarged images). **c,** No change in p300 nucleocytoplasmic shuttling after treatment with a p300 activator (0.5 mM CTB) or a p300 inhibitor (20 μM c646). N = 3, about 50 cells scored per condition per experiment. Two-tailed paired t test. Scale bar, 5 μm. Data are presented as mean values +/- s.d. unless otherwise specified. Source numerical data are available in source data.

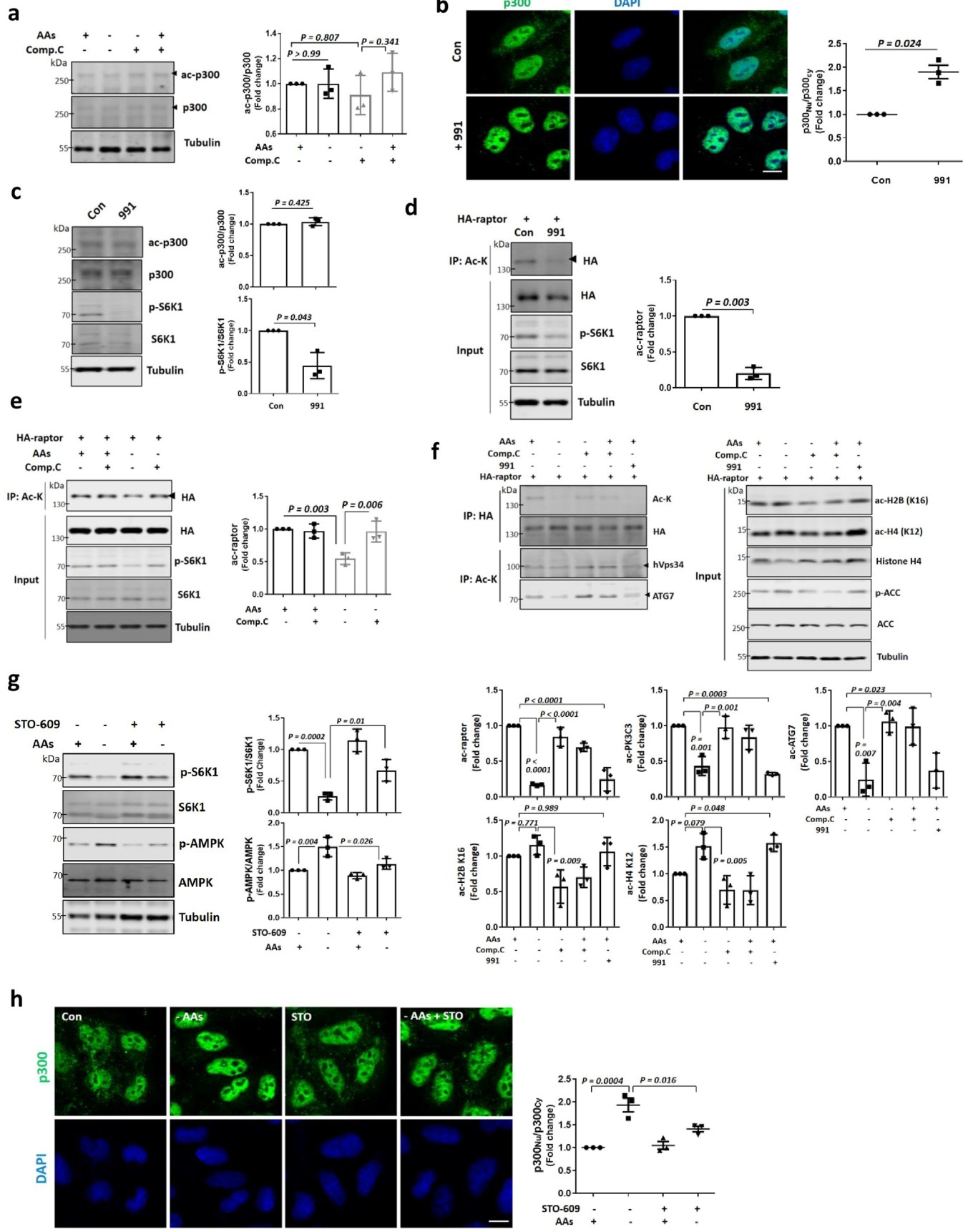

**Extended Data Fig. 6 | See next page for caption.**

**Extended Data Fig. 6 | p300 shuttling is associated with AMPK activity.**
**a**, No alteration in p300 activity by 5 µM Comp. C with or without AA starvation for 1 h. Blots are representative of three biologically independent experiments (N = 3). One-way ANOVA with post hoc Tukey test. **b**, Increased nuclear transport of p300 by 10 µM 991. N = 3, about 50 cells scored per condition per experiment. One sample t-test. Scale bar, 5 µm. **c**, No change in p300 activity, but reduced mTORC1 activity mediated by 10 µM 991. N = 3. One sample t-test. **d**, Decreased raptor acetylation by 10 µM 991. N = 3. One sample t-test. **e**, Rescue of decreased raptor acetylation in AAs-depleted cells by 5 µM Comp.C. N = 3. One-way ANOVA with post hoc Tukey test. **f**, Acetylation of p300-targeted proteins in nucleus and cytoplasm in an AMPK activity-dependent manner. N = 3. One-way ANOVA with post hoc Tukey test. **g**, Rescue of mTORC1 inhibition and AMPK activation by AA starvation after treatment with 25 µM STO-609. N = 3. One-way ANOVA with post hoc Tukey test. **h**, Inhibition of nuclear transport of p300 by 25 µM STO-609. N = 3, about 30 cells scored per condition per experiment. One-way ANOVA with post hoc Tukey test. Scale bar, 5 µm. Data are presented as mean values +/- s.d. unless otherwise specified. Source numerical data and unprocessed blots are available in source data.

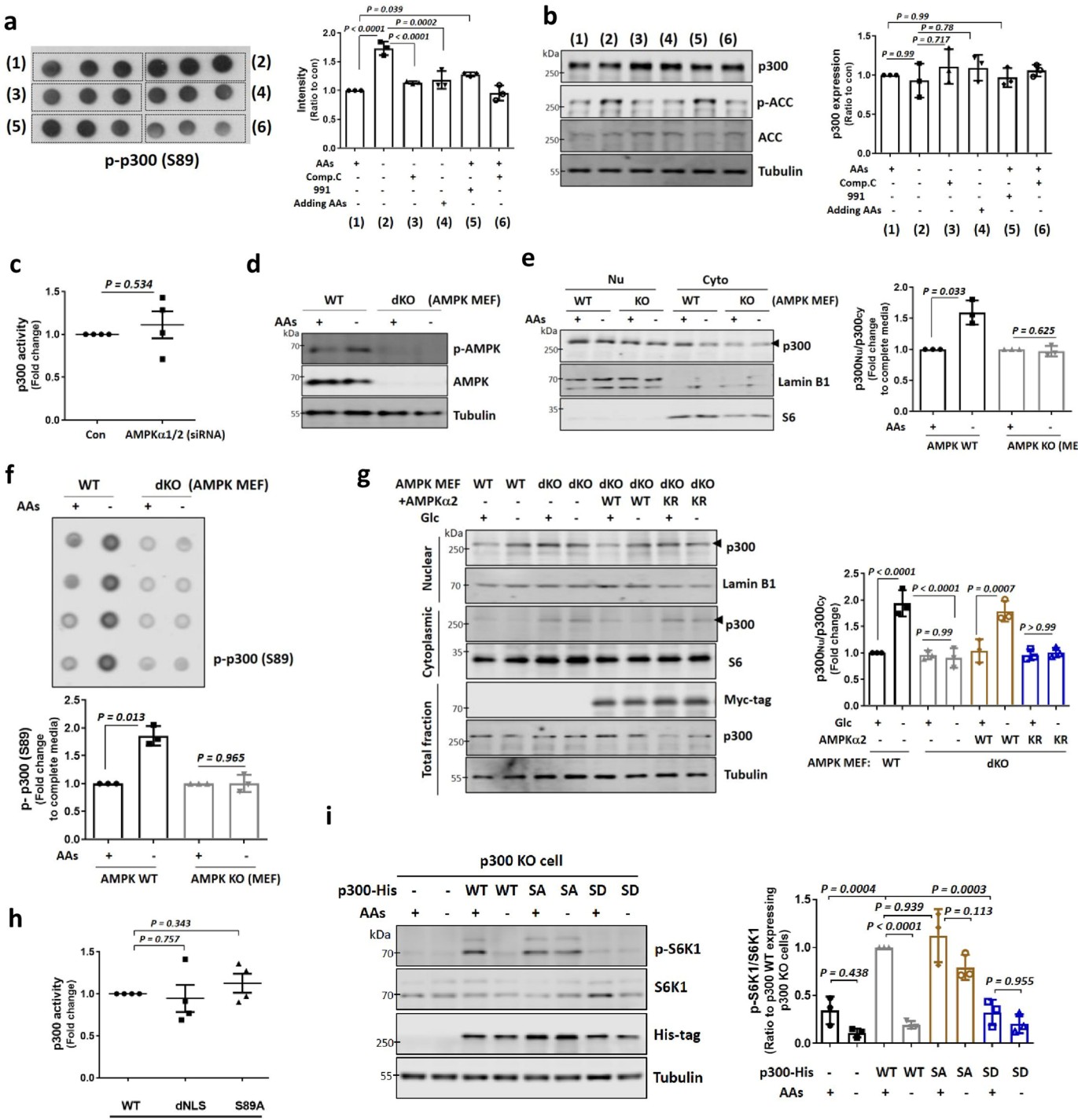

**Extended Data Fig. 7 | AMPK-mediated p300 nucleocytoplasmic shuttling.**
**a**, **b**, Phosphorylation of p300 (p-p300) at S89 in an AMPK-dependent manner.
Blots are representative of three biologically independent experiments (N = 3).
One-way ANOVA with post hoc Tukey test. **c**, p300 activity in control or AMPKα1/2
knockdown cells. N = 4. One sample t-test. **d**, Validation of AMPK WT and α1/α2
double knock-out (dKO) MEF using anti-total AMPKα and anti-phosphorylated
AMPKα antibody. N = 3. **e**, Inhibition of p300 nuclear transport in starved AMPK
dKO MEF. N = 3. One sample t-test. **f**, Failure of increased phosphorylation of
p300 at S89 in AMPK dKO MEF by depleting AAs. N = 3. One sample t-test.
**g**, Rescue of p300 shuttling in Glc-depleted AMPK dKO MEFs after reconstitution

with myc-tagged AMPKα2, but not myc-tagged AMPK α2 K45R. The expression
levels following transfection are depicted by the quantification of the myc-
tag. N = 3. One-way ANOVA with post hoc Tukey test. **h**, Measurement of
acetyltransferase activity in p300 WT, dNLS or S89A construct. N = 4. Two-tailed
paired t test. **i**, mTORC1 activity in His-tagged p300 WT, SA or SD expressing
p300 KO cells. The expression levels following transfection are depicted by the
quantification of the His-tag. N = 3. One-way ANOVA with post hoc Tukey test.
Data are presented as mean values +/- s.d. unless otherwise specified. Source
numerical data and unprocessed blots are available in source data.

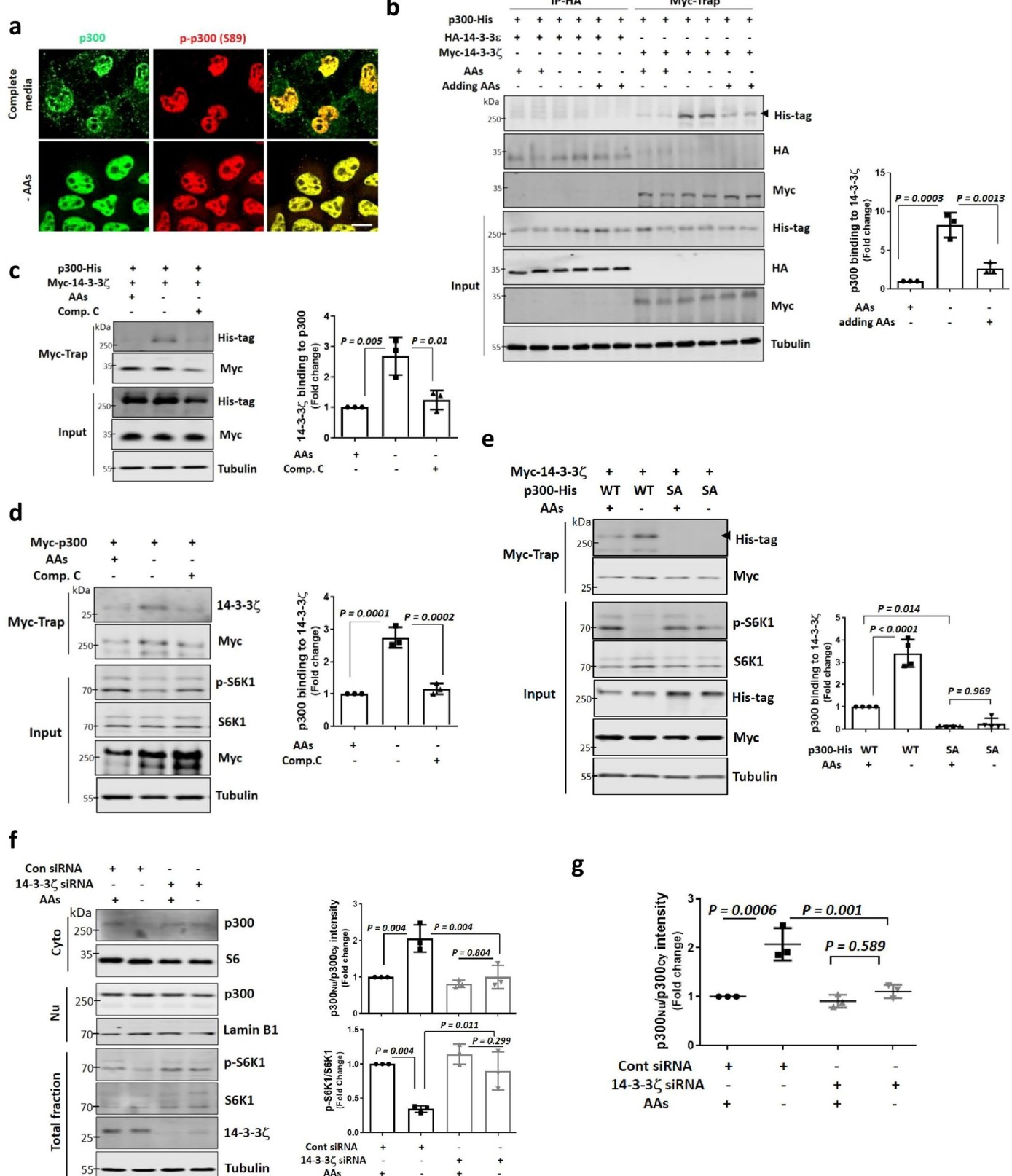

**Extended Data Fig. 8 | See next page for caption.**

**Extended Data Fig. 8 | 14-3-3 protein is crucial for p300 accumulation in nucleus. a**, Localisation of phosphorylated p300 (p-p300) in complete and AA-depleted media using a p300 phospho-specific antibody. N = 3, n = about 30 cells analysed per condition per experiment. Scale bar, 5 μm. **b**, Interaction of His-tagged p300 with myc-tagged 14-3-3ζ and its increase in AA-depleted conditions. The expression levels following transfection are determined by using antibodies specific to the tagged protein. Blots are representative of three biologically independent experiments (N = 3). One-way ANOVA with post hoc Tukey test. **c,d**, Inhibition of interaction of p300 with 14-3-3ζ by 5 μM Comp. C. The expression levels following transfection are determined by using antibodies specific to the tagged protein. N = 3. One-way ANOVA with post hoc Tukey test.

**e**, Inhibition of His-tagged p300 SA mutant binding with myc-tagged 14-3-3ζ. The expression levels following transfection are determined by using antibodies specific to the tagged protein. N = 4. One-way ANOVA with post hoc Tukey test. **f**, Remaining cytoplasmic p300 levels and mTORC1 activity in 14-3-3ζ knockdown cells in AA-depleted media for 1 h. N = 3. One-way ANOVA with post hoc Tukey test. **g**, Inhibition of p300 nuclear accumulation by depletion of AAs in 14-3-3ζ siRNA transfected HeLa cells. N = 3, about 30 cells scored per condition per experiment. One-way ANOVA with post hoc Tukey test. Data are presented as mean values +/- s.d. unless otherwise specified. Source numerical data and unprocessed blots are available in source data.

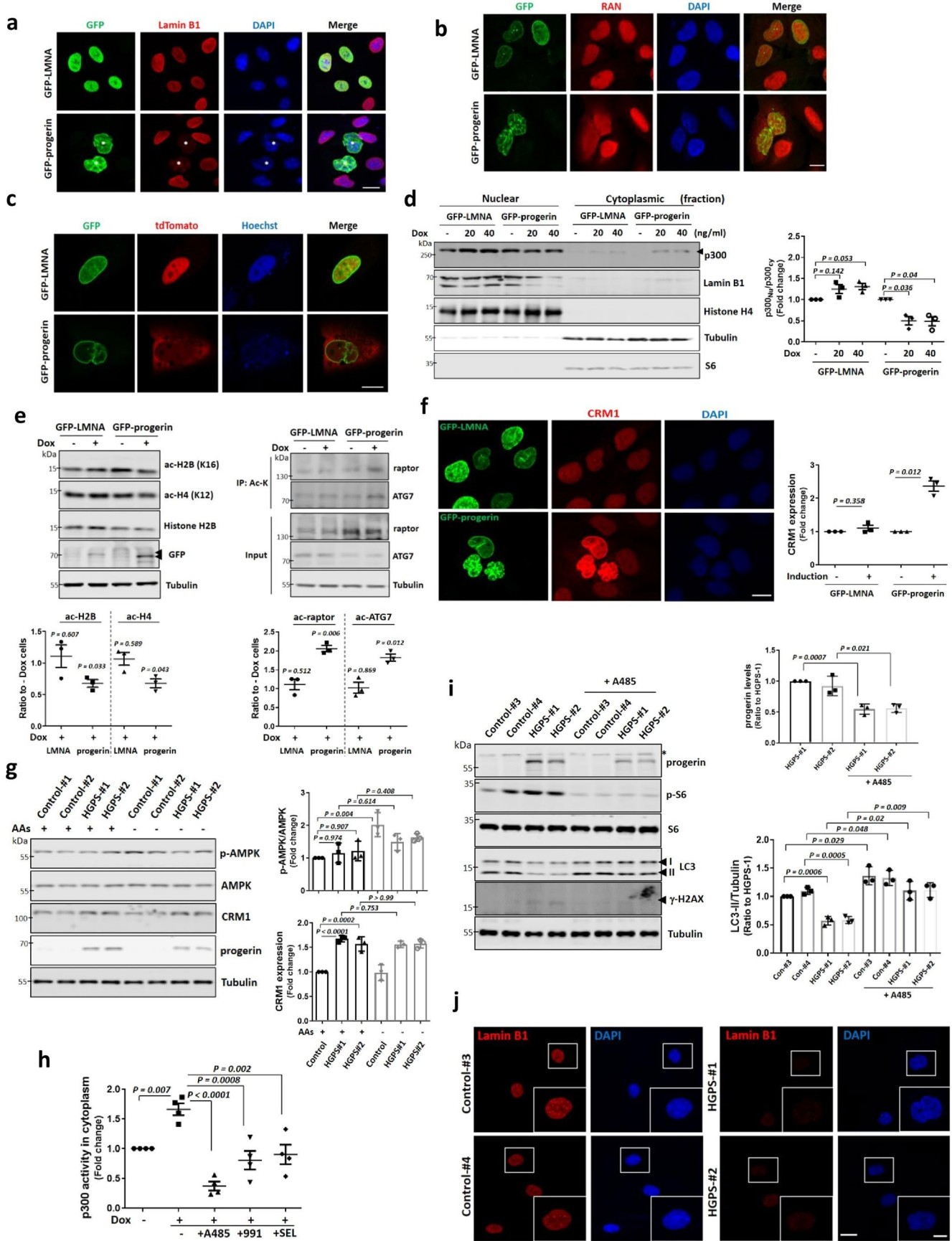

**Extended Data Fig. 9 | See next page for caption.**

**Extended Data Fig. 9 | Altered p300 localisation, AMPK activity and CRM1 expression in HGPS cells. a**, Decreased Lamin B1 expression level in HGPS cells. * GFP-progerin induced cells. Images are representative of three biologically independent experiments (N = 3). Scale bar, 5 µm. **b**, **c**, Defect in nucleus-cytoplasmic transport by progerin using RAN staining (**b**) or shuttling-reporter construct (NLS-tdTomato-NES) (**c**). N = 3. Scale bar, 5 µm. **d**, Altered p300 localisation by progerin expression using fractionation. N = 3. Two-tailed paired t test. **e**, Differential regulation of acetylated proteins between nucleus and cytoplasm in progerin-expressing cells. N = 3. One sample t-test. **f**, Enhanced CRM1 expression in HGPS cells. N = 3, about 50 cells scored per condition. One sample t-test. Scale bar, 5 µm. **g**, Defect in AMPK activation and increased CRM1

expression in HGPS fibroblasts. N = 3. One-way ANOVA with post hoc Tukey test. Data from control vials #1 and #2 were pooled as these are from the same individual (see Supplementary Table 1). **h**, Rescue of enhanced p300 activity in cytoplasm after treatment with A485, 991 or SEL. N = 4 each group. One-way ANOVA with post hoc Tukey test. **i**, Reduced progerin expression levels in HGPS fibroblasts by treatment with p300 inhibitor A485 for 24 h. N = 3. Two-tailed paired t test. **j**, Reduced Lamin B1 levels in HGPS fibroblasts. Scale bar, 5 µm, 2 µm (enlarged images). N = 3, about 50 cells scored per condition. Data are presented as mean values +/- s.d. unless otherwise specified. Source numerical data and unprocessed blots are available in source data.

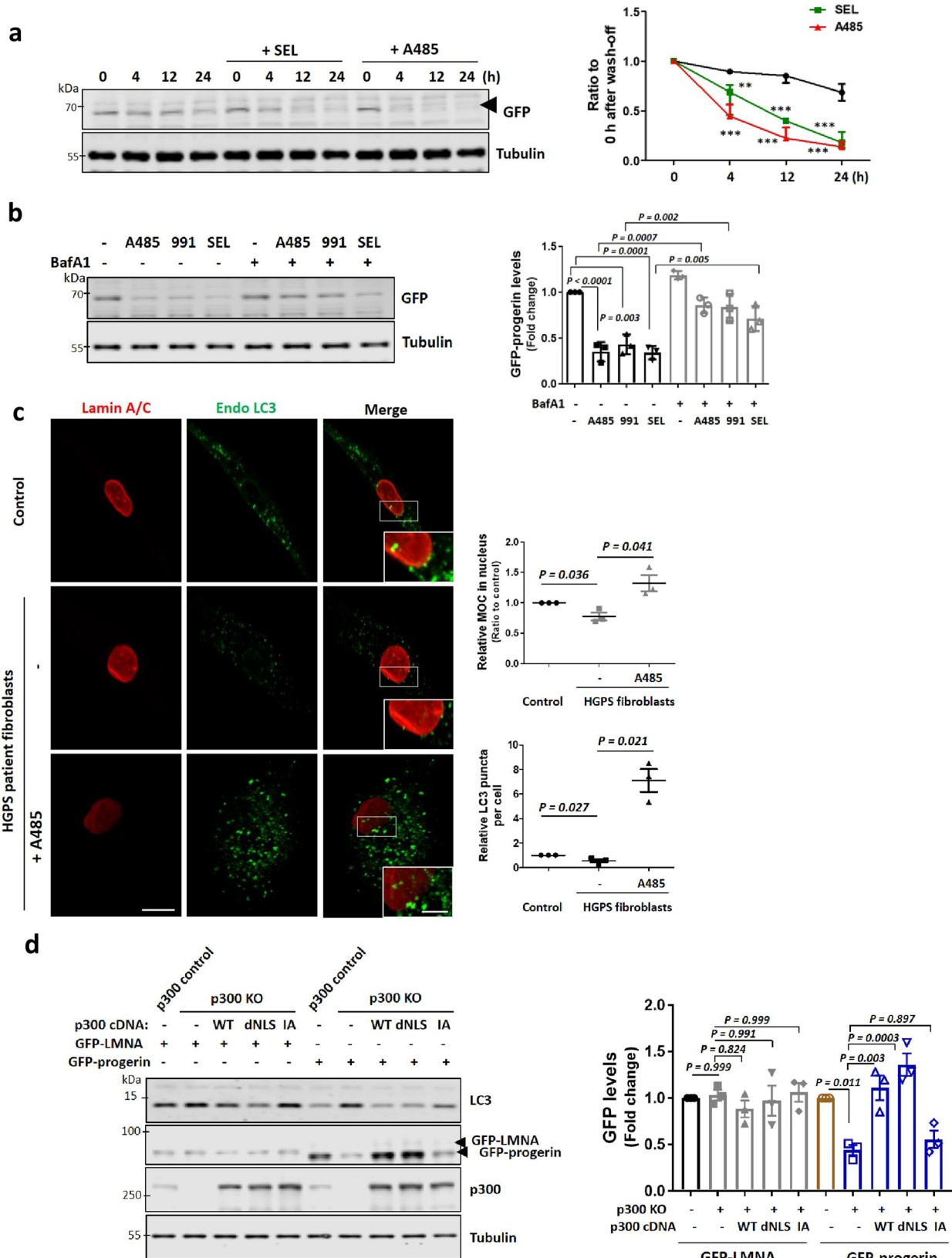

**Extended Data Fig. 10 | See next page for caption.**

**Extended Data Fig. 10 | Enhanced degradation of progerin by autophagy.**
**a**, Enhanced degradation of progerin protein by treatment with SEL or A485. Representative western blots of GFP-progerin inducible cell lines following different times of Dox removal. Blots are representative of three biologically independent experiments (N = 3). ** p < 0.01, *** p < 0.001 vs. vehicle (DMSO) treated cells (two-way ANOVA with Bonferroni post-test). **b**, Autophagy mediated progerin degradation by treatment with A485, 991 or SEL. N = 3. One-way ANOVA with post hoc Tukey test. **c**, More colocalisation of progerin with endogenous LC3-positive vesicles in HGPS fibroblasts after treatment with A485. N = 3, about 50 cells scored per condition per experiment. Two-tailed paired t-test. Scale bar, 5 μm, 2 μm (enlarged images). **d**, Regulation of progerin degradation in p300 KO expressing His-tagged p300 WT, dNLS or I265A (IA). N = 3. One-way ANOVA with post hoc Tukey test. Data are presented as mean values +/- s.d. unless otherwise specified. Source numerical data and unprocessed blots are available in source data.

# Reporting Summary

## Statistics

For all statistical analyses, confirm that the following items are present in the figure legend, table legend, main text, or Methods section.

| n/a | Confirmed | |
|---|---|---|
| ☐ | ☒ | The exact sample size (*n*) for each experimental group/condition, given as a discrete number and unit of measurement |
| ☐ | ☒ | A statement on whether measurements were taken from distinct samples or whether the same sample was measured repeatedly |
| ☐ | ☒ | The statistical test(s) used AND whether they are one- or two-sided *Only common tests should be described solely by name; describe more complex techniques in the Methods section.* |
| ☒ | ☐ | A description of all covariates tested |
| ☒ | ☐ | A description of any assumptions or corrections, such as tests of normality and adjustment for multiple comparisons |
| ☐ | ☒ | A full description of the statistical parameters including central tendency (e.g. means) or other basic estimates (e.g. regression coefficient) AND variation (e.g. standard deviation) or associated estimates of uncertainty (e.g. confidence intervals) |
| ☐ | ☒ | For null hypothesis testing, the test statistic (e.g. *F*, *t*, *r*) with confidence intervals, effect sizes, degrees of freedom and *P* value noted *Give P values as exact values whenever suitable.* |
| ☒ | ☐ | For Bayesian analysis, information on the choice of priors and Markov chain Monte Carlo settings |
| ☒ | ☐ | For hierarchical and complex designs, identification of the appropriate level for tests and full reporting of outcomes |
| ☐ | ☒ | Estimates of effect sizes (e.g. Cohen's *d*, Pearson's *r*), indicating how they were calculated |

*Our web collection on statistics for biologists contains articles on many of the points above.*

## Software and code

Policy information about availability of computer code

| Data collection | Gels were imaged using LICOR-Odyssey apparatus using IMAGE STUDIO Lite Licor ver 5.2, Inc. Confocal Carl Zeiss LSM710, LSM780 and LSM880 operated with ZEN Black imaging software 2.3. |
|---|---|
| Data analysis | IMAGE STUDIO Lite LI-COR ver 5.2, Inc and Image J (National Institute of Health, USA) for gel analysis. ZEN imaging software (ZEN Black 2.3 Carl Zeiss Microscopy) and Image J ver 1.54f for microscopic image analysis. Volocity 6.3 Software (PerkinElmer) for Mander's Overlap Coefficient (MOC) or Pearson's correlation coefficient (PCC). Microsoft Excel (Excel 2016 Microsoft office) and GraphPad Prism v7 (GraphPad Software) for statistical analysis. |

For manuscripts utilizing custom algorithms or software that are central to the research but not yet described in published literature, software must be made available to editors and reviewers. We strongly encourage code deposition in a community repository (e.g. GitHub). See the Nature Portfolio guidelines for submitting code & software for further information.

## Data

Policy information about availability of data

All manuscripts must include a data availability statement. This statement should provide the following information, where applicable:
- Accession codes, unique identifiers, or web links for publicly available datasets
- A description of any restrictions on data availability
- For clinical datasets or third party data, please ensure that the statement adheres to our policy

> All data supporting the findings of this study are available from the corresponding author upon reasonable request. Source data are provided with this paper.

## Research involving human participants, their data, or biological material

Policy information about studies with human participants or human data. See also policy information about sex, gender (identity/presentation), and sexual orientation and race, ethnicity and racism.

| | |
|---|---|
| Reporting on sex and gender | not applicable |
| Reporting on race, ethnicity, or other socially relevant groupings | not applicable |
| Population characteristics | not applicable |
| Recruitment | not applicable |
| Ethics oversight | not applicable |

Note that full information on the approval of the study protocol must also be provided in the manuscript.

# Field-specific reporting

Please select the one below that is the best fit for your research. If you are not sure, read the appropriate sections before making your selection.

☒ Life sciences ☐ Behavioural & social sciences ☐ Ecological, evolutionary & environmental sciences

For a reference copy of the document with all sections, see nature.com/documents/nr-reporting-summary-flat.pdf

# Life sciences study design

All studies must disclose on these points even when the disclosure is negative.

| | |
|---|---|
| Sample size | Sample sizes were chosen on the basis of extensive experience with the assays we have performed. (e.g. Son et al. Cell Metabolism 2019; Son et al. Nature Communications 2020; Wrobel et al. Nature Communications 2022) |
| Data exclusions | No data were excluded from the analysis. |
| Replication | All experiments were repeated by at least three times, and all experiments were reproducible. We used the average of each triplicate as a biological replicate for statistical analyses. |
| Randomization | All in vivo experiments and tests were randomly assigned, but no randomization was performed for cell culture experiments. |
| Blinding | Immunofluorescence analysis was blinded when possible. Western blot analysis was not blinded as it was not possible as the gel loading order needs to be defined. Investigators were not blinded during the other experiments. |

# Reporting for specific materials, systems and methods

We require information from authors about some types of materials, experimental systems and methods used in many studies. Here, indicate whether each material, system or method listed is relevant to your study. If you are not sure if a list item applies to your research, read the appropriate section before selecting a response.

## Materials & experimental systems

| n/a | Involved in the study |
|-----|----------------------|
| ☐ | ☒ Antibodies |
| ☐ | ☒ Eukaryotic cell lines |
| ☒ | ☐ Palaeontology and archaeology |
| ☐ | ☒ Animals and other organisms |
| ☒ | ☐ Clinical data |
| ☒ | ☐ Dual use research of concern |
| ☒ | ☐ Plants |

## Methods

| n/a | Involved in the study |
|-----|----------------------|
| ☒ | ☐ ChIP-seq |
| ☒ | ☐ Flow cytometry |
| ☒ | ☐ MRI-based neuroimaging |

## Antibodies

**Antibodies used**

Primary antibodies: mouse anti-Flag M2 (#F3165, RRID:AB_262044, WB 1:2000), rabbit anti-Actin (#A2066, RRID AB_476693, WB 1:2000), rabbit anti-HA tag (#AP1012A, RRID:AB_352510, WB 1:2000), mouse anti-progerin (#05-1231, RRID:AB_1587236, WB 1:1000) and mouse anti-α-Tubulin (#T9026, RRID:AB_477593, WB 1:3000) from Sigma Aldrich; mouse anti-GAPDH clone 6C5 (#ab8245, RRID:AB_2107448, WB 1:3000), rabbit anti-p300 (#ab10485, RRID:AB_297224, WB 1:1000, IF 1:200), rabbit anti-phospho-p300 (Ser89; #ab135554, DB 1:500, IF 1:100), rabbit anti-CBP (#ab137334, RRID:AB_2916306, WB 1:1000), rabbit anti-GFP tag (#ab6556, RRID:AB_305564, WB 1:2000, IF 1:200), rabbit anti-H2B (acK16; #ab177427; WB 1:1000), rabbit anti-H4 (#ab7311,RRID:AB_305837, WB 1:1000), rat anti-LAMP1 (#ab25630, RRID:AB_470708, IF 1:300), rabbit anti-LAMP1 (#ab24170, RRID:AB_775978, IF 1:400), rabbit anti-acetylated-Lysine (Ac-K) (#ab21623, RRID:AB_446436, WB 1:500), mouse anti-6X His tag (#ab18184, RRID:AB_444306, WB 1:1000), rabbit anti-Lamin B1 (#ab16048, RRID:AB_443298, WB 1:1000), mouse anti-Myc tag (#ab32, RRID:AB_303599, WB 1:1000), rabbit anti-14-3-3zeta (#ab51129, RRID:AB_867447, WB 1:1000 ), rabbit anti-Histone H3 (tri-methyl K9; #ab8898, RRID:AB_306848, WB 1:1000, IF 1:200), mouse anti-BAT3 (#ab88292, RRID:AB_2040593, IF 1:300), rabbit anti-Vps34 (# ab227861, RRID: AB_2827796; WB 1:1000), rabbit anti-ATG7 (Cat# ab133528, RRID:AB_2532126; WB 1:1000), rabbit anti-LC3B (#ab51520, RRID:AB_881429; IF 1:400) and rabbit anti-ALX1 (#ab181101, IF 1:300) from Abcam; goat anti-HA (#NB600-362, RRID:AB_10124937, 1:1000) and rabbit anti-LC3B (#NB100-2220, RRID: AB_10003146, WB 1:1000) from Novus Biologicals; rabbit anti-p300 (#sc-585, RRID:AB_2231120, #sc-48343, RRID:AB_628075, IF 1:50), goat anti-Lamin B (#sc-6217, RRID:AB_648158, WB 1:1000), rabbit anti-p-PP2Ac (#sc-271903, RRID:AB_10611810, WB 1:1000), mouse anti-PP1γ (#sc-515943, RRID:AB_2909495, WB 1:1000), mouse anti-CRM1 (#sc-74454, RRID:AB_1122704, WB 1:500, IF 1:100), mouse anti-TOM20 (#sc-17764, RRID:AB_628381, IF 1:100) from SantaCruz Biotechnology; mouse anti-GFP (#632375 and #632592, RRID:AB_2756343, IF 1:100) from Clontech; mouse anti-HA.11 clone 16B12 (#MMS-101P, RRID:AB_10064068, WB 1:1000) from Covance; mouse anti-p300 (#05-257, RRID:AB_309670, WB 1:1000, IF 1:100) from Millipore; mouse anti- acetylated-Lysine (Ac-K) (#AAC01, RRID:AB_2884959, WB 1:1000) from Cytoskeleton, Inc; mouse anti-GFP tag (#66002-1-Ig, RRID:AB_11182611, WB 1:1000, IF 1:100), rabbit anti-Lamin A/C (#10298-1-AP, RRID:AB_2296961, IF 1:100), rabbit anti-Lamin B1 (#12987-1-AP, RRID:AB_2136290, IF 1:100) and mouse anti-mTOR (#66888-1-Ig, RRID:AB_2882219, IF 1:400) from Proteintech; rabbit anti-p300 (#86377, RRID:AB_2800077, IF 1:300), rabbit anti-CBP (#7389, RRID:AB_2616020, WB 1:1000), rabbit anti-acetyl-CBP (Lys1535)/p300 (Lys1499) (#4771, RRID:AB_2262406, WB 1:1000), rabbit anti-GCN5 (#3305, RRID:AB_2128281, 1:1000), rabbit anti-PCAF (#3378, RRID:AB_2128409, WB 1:1000), rabbit anti-Histone H2B (#12364, RRID:AB_2714167, WB 1:1000), rabbit anti-Histone H3 (#9715, RRID:AB_331563, WB 1:1000), rabbit anti-H3 (acK9; #9649, RRID:AB_823528, WB 1:1000), rabbit anti-H3 (acK56; #4243, RRID:AB_10548193, WB 1:1000), rabbit anti-H4 (acK12; #2591, RRID:AB_2118617, WB 1:1000), rabbit anti-H4 (acK16; #13534, RRID:AB_2687581, WB 1:1000), rabbit anti-LAMP1 (#9091, RRID:AB_2687579, WB 1:1000, IF 1:200), rabbit anti-mTOR (#2972, RRID:AB_330978, WB 1:1000; #2983, RRID:AB_2105622, IF 1:400), rabbit anti-raptor (#2280, RRID:AB_561245, WB 1:1000, IF 1:100), rabbit anti-phospho-S6K1 (Thr389; #9234, RRID:AB_2269803, WB 1:1000), anti-total S6K1 (#9202, RRID:AB_331676, WB 1:1000), rabbit anti-phospho-S6 Ribosomal Protein (p-S6) (Ser235/236; #4856, RRID:AB_2181037, WB 1:1000, IF 1:200), rabbit anti-S6 Ribosomal Protein (S6) (#2217, RRID:AB_331355, WB 1:1000), rabbit anti-phospho-4E-BP1 (Thr37/46; #9459, RRID:AB_330985, WB 1:1000), rabbit anti-4E-BP1 (#9452, RRID:AB_331692, WB 1:1000), rabbit anti-acetylated-Lysine (Ac-K) (#9814, RRID:AB_10544700; #9441, RRID:AB_331805, WB 1:1000), rabbit anti-PP2Ac (#2259, RRID:AB_561239, WB 1:1000), rabbit anti-phospho-ACC1 (Ser79; #11818, RRID:AB_2687505, WB 1:1000), rabbit anti-ACC (#3676, RRID:AB_2219397, WB 1:1000), rabbit anti-phospho-AMPK (Thr172; #2531, RRID:AB_330330, 1:1000), rabbit anti-AMPK (#2532, RRID:AB_330331, WB 1:1000), rabbit anti-phospho-Histone H2A.X (Ser139; #2577, RRID:AB_2118010, WB 1:1000; #9718, RRID:AB_2118009, IF 1:200) from Cell Signaling Technology.
Secondary Antibodies: anti-mouse (#NA931V, RRID:AB_772210) and anti-rabbit (#NA934V) horseradish peroxidise (HRP)-conjugated secondary antibodies (GE Healthcare); anti-goat horseradish peroxidise (HRP)-conjugated secondary antibody (#611620, RRID:AB_87867, Invitrogen/Life Technologies). For immunoflourescence, goat-anti-mouse Alexa Fluor 488 (#A11029, RRID:AB_2534088, 1:400), 555 (#A21147, RRID:AB_1500897, 1:400) and 594 (#A11032, RRID:AB_2534091, 1:400), goat-anti-rabbit Alexa Fluor 488 (#A32731, RRID:AB_2633280, 1:400) and 555 (#A21428, RRID:AB_141784, 1:400) from ThermoFisher Scientific.

**Validation**

All antibodies used in this study were purchased from commercial vendors who had validated specificity in human cells/ mouse tissues for the specific assays (Western blot, immunoprecipitation and/or immunofluorescence). It is described on data sheets and online.
We have confirmed that the LC3 antibody does not detect an LC3-II band in autophagy null cells.

## Eukaryotic cell lines

Policy information about cell lines and Sex and Gender in Research

**Cell line source(s)**

Human cervical epithelium HeLa (ATCC; #CCL-2; CVCL_0030), human neuroblastoma SH-SY5Y (ECACC; #94030304), human embryonic kidney cell line HEK293 (ECACC; #85120602), Human retinal pigment epithelium RPE1 (ATCC, #CRL-4000), human breast cancer cell line MCF7 (ATCC; #HTB-22), human lung carcinoma A549 cells (kindly provided by Dr. F. Buss (University of Cambridge, UK)), human neuroglioma H4 cells (ATCC; #HTB-148), Human hepatoma HepG2 cells (ECACC; #85011430), and

| | |
|---|---|
| | Human mammary epithelium MCF10A cells were purchased from Horizon (#HD PAR-058).<br>AMPK α1/α2 double knockout (dKO) MEFs were a gift from B. Viollet (Universite´ Paris Descartes).<br>Primary mouse embryonic fibroblasts (MEFs) were isolated from wild-type C57BL/6 mouce embryos at E12.<br>Unaffected fibroblast control (Coriell Institute #GM05565, #GM02036, #GM00969)<br>Primary human dermal fibroblasts from patients with HGPS (1) (Coriell Institute #AG01972)<br>Primary human dermal fibroblasts from patients with HGPS (2) (Coriell Institute #AG11513) |
| Authentication | The cell lines were ordered from ATCC, Horizon or Coriell Institute with authentication.<br>HeLa authenticatd by ATCC (by Short Tandem Repeat (STR) profiling; FTA barcode:STRA1466)<br>HEK293 authenticatd by LGC (STR profiling, FTA barcode:STRA1472)<br>SH-SY5Y authenticatd by LGC (STR profiling, FTA barcode:STRA1440)<br>RPE1 authenticatd by ATCC (STR profiling)<br>MCF7 authenticatd by ATCC (STR profiling)<br>H4 authenticatd by ATCC (STR profiling)<br>HepG2 authenticatd by ECACC (STR profiling)<br>MCF10A authenticatd by Horizon (STR profiling)<br>AMPK dKO MEF were authenticatd by Western blot analysis with AMPK antibody.<br>Primary MEFs were not authenticated by STR. |
| Mycoplasma contamination | All the cells were regularly tested using EZ-PCR Mycoplasma Test Kit (Biological Industries; cat#20-700-20). Cells used in this study were mycoplasma negative. |
| Commonly misidentified lines<br>(See ICLAC register) | no commonly misidentified cell lines were used in the study. |

# Animals and other research organisms

Policy information about studies involving animals; ARRIVE guidelines recommended for reporting animal research, and Sex and Gender in Research

| | |
|---|---|
| Laboratory animals | Wild type C57Bl/6J mice (6 to 7-weeks-old)<br>Mice were housed in individually ventilated cages with free access to standard animal food chow (#R105; SAFE) and water, in a climate-controlled room with a 12 h light/dark cycle, except when subjected to starvation-refeeding protocols. |
| Wild animals | No wild animals were used in the study. |
| Reporting on sex | The ratio of sexes of used mice was 1:1 and the number of the mice used for the experiments are indicated for each experiment in the figure legends (in general, n=6). |
| Field-collected samples | No field collected samples were used in the study. |
| Ethics oversight | Mouse studies and procedures were performed in accordance with the UK Animals (Scientific Procedures) Act with appropriate Home Office Project and Personal animal licenses and with the approval of the University of Cambridge Animal welfare and Ethical Review Body. |

Note that full information on the approval of the study protocol must also be provided in the manuscript.

