## [Peer Review File · Nature Cell Biology]

Peer Review Information

Journal: Nature Cell Biology

Manuscript Title: p300 nucleocytoplasmic shuttling underlies mTORC1 hyperactivation in Hutchinson-Gilford Progeria Syndrome

Corresponding author name(s): Professor David Rubinsztein

Editorial Notes:

**Redactions –
published data**

Reviewer Comments & Decisions:

Decision Letter, initial version:

Dear Professor Rubinsztein,

Thank you for submitting your manuscript, "p300 cytoplasm-nucleus shuttling underlies mTORC1 hyperactivation in Hutchinson-Gilford Progeria Syndrome", to Nature Cell Biology. I am so sorry for the long delay before we could communicate our decision to you and sincerely apologize for the long duration of the process. The manuscript has now been seen by 3 referees, who are experts in lysosomes, mTOR, aging, neurodegeneration (Referee #1); HGPS (Referee #2); and mTORC in disease (Referee #3). As you will see from their comments (attached below), they found the work of

potential interest but have raised valid concerns that in our view would need to be addressed thoroughly before we can consider publication in Nature Cell Biology.

As per our standard process, we have now editorially discussed the reviews to determine which reviewer points should be addressed with priority to strengthen the core conclusions. Please see below for our thoughts on the directions for revision. Our standard revision period is six months, and as you know, we are committed to providing a fair and constructive peer-review process, so please do not hesitate to reach out to discuss any concerns or questions about the reviews or revision process.

In our view, it will be essential to address the reviews as follows:

A. The link between p300, mTORC1, and autophagy, including in the context of HGPS needs to be strengthened with additional data, as recommended by all reviewers:

Rev#1 "One area in the manuscript that needs strengthening is how autophagy is analysed. I only say this here because autophagy is used as an explanatory mechanism for how the p300-mTORC1 axis leads to enhanced biological ageing. Autophagy is even invoked in the concluding paragraph. Specifically with regards to extended data figure 2 – a flux assay needs to be done here, say +/- p300, +/- bafilomycin and WB for LC3BII. A flux assay needs to be performed because although the current results are suggestive, they do have more than one interpretation."

Rev#2 "In Fig 7, Based on the data enough evidence was provided to believe that progerin is degraded by activation of autophagy (which was also reported by others). However, the claim that progerin itself inhibits autophagy just because of p300 mislocalization is not consistent with the data presented."

Rev3 "Considering the multifaceted nature of mTOR signaling, it appears that there should be at least some consideration that mTORC1 outputs other than autophagy may contribute to the ensuing phenotypes. Indeed, the support for the role of autophagy in HGPS cellular phenotypes is based on correlative data and use of highly pleiotropic Bafilomycin A1 (that this same group has demonstrated may interfere with proteasomal protein degradation that is also affected by mTORC1). To this end, it was thought that the implication of autophagy should be somewhat toned down, or that additional evidence should be provided to support this conclusion."

B. The reviewers additionally felt that other aspects of the proposed mechanism of mTORC1 regulation needed to be evaluated more carefully:

Rev#1 "a key part of the mechanism proposed in this paper posits that p300 activates mTORC1 by acetylating RAPTOR, and this suppresses autophagy. The authors convincingly show that p300 acetylates RAPTOR, and this is modified by relevant nutrient-related conditions, especially in conjunction with a delta-NLS p300 mutant. What I couldn't find in the paper were experiments where lysines on RAPTOR were mutated to prove unequivocally that acetylation of RAPTOR was responsible for the downstream mechanisms proposed by the paper. This is important because p300 has many substrates, and maybe acetylation of one of those other substrates was responsible for the effect on autophagy for example. Previous work from the authors showed p300 acetylates K1097 on RAPTOR and this is required for mTORC1 activation (PMID: 32561715). What I couldn't find was a link between mutation of this residue and autophagic flux. If the relationship between K1097 acetylation and

autophagic flux has already been demonstrated in the literature, please state this in the manuscript. If not, I would really appreciate the inclusion of this experiment to tie important parts of the mechanism together."

Rev#2 points #3, 6

Rev#3:

"In the light of some recent studies, the authors should exclude the possibility that modulating p300 levels and/or activity may affect uptake and/or transport of glucose or amino acids (e.g., Martins et al, JCI insight 2022) and thereby affect mTORC1 signaling indirectly."

"Most of the readouts of mTORC1 activity are limited to S6K1/rpS6 phosphorylation. Considering the emerging findings showing potential modularity of mTORC1 signaling, it is advised that other mTORC1 substrates (e.g., 4E-BP1) are also used for the readout of mTORC1 activity at least in the initial experiments. It should also be noted that rpS6 p-S235/236 Ab is not a very good readout of mTORC1 signaling as this site is also phosphorylated by AGC kinases other than S6Ks including RSKs. Accordingly, the phosphorylation of rpS6 at this site is not affected in cells that are devoid of S6Ks (Pende et al MCB 2004)."

"It appears that in some cases the increase in cytoplasmic p300 is not accompanied by corresponding decrease in nuclear p300 upon amino acid or glucose repletion. What happens to total p300 levels upon amino acid or glucose repletion?"

"The authors should likely discuss why the focus was on glucose and amino acids and not on other stimuli known to alter mTORC1 signaling."

"Conclusion that upregulation of CRM1 and lower AMPK activity are responsible to increased mTORC1 activity in HGPS fibroblasts was thought to be somewhat premature considering the evidence, and previous literature showing that AMPK can suppress mTORC1 signaling in a variety of ways, whereby it was thought that the authors do not provide sufficient data to conclude that the effects of dampened AMPK signaling on mTORC1 are exclusively mediated by p300, in the context of HGPS. Some toning down of the statements in this section of the manuscript appears to be warranted.'

C. Lastly, please pay special attention to the reviewers' concerns about the controls used (including Rev#2 point#4), statistical analyses, quality of the data and please address their other points aimed at strengthening the current dataset, clarifying and streamlining the manuscript, and enriching the discussion.

D. Finally, please pay close attention to our guidelines on statistical and methodological reporting (listed below) as failure to do so may delay the reconsideration of the revised manuscript. In particular, please provide:

- a Supplementary Table including all numerical source data in Excel format, with data for different figures provided as different sheets within a single Excel file. The file should include source data giving rise to graphical representations and statistical descriptions in the paper and for all instances where the figures present representative experiments of multiple independent repeats, the source data of all

repeats should be provided.

We would be happy to consider a revised manuscript that would satisfactorily address these points, unless a similar paper is published elsewhere, or is accepted for publication in Nature Cell Biology in the meantime.

- ensure that it conforms to our format instructions and publication policies (see below and www.nature.com/nature/authors/).
- provide a point-by-point rebuttal to the full referee reports verbatim, as provided at the end of this letter.
- provide the completed Editorial Policy Checklist (found here <https://www.nature.com/authors/policies/Policy.pdf>), and Reporting Summary (found here <https://www.nature.com/authors/policies/ReportingSummary.pdf>). This is essential for reconsideration of the manuscript and these documents will be available to editors and referees in the event of peer review. For more information see <http://www.nature.com/authors/policies/availability.html> or contact me.

Nature Cell Biology is committed to improving transparency in authorship. As part of our efforts in this direction, we are now requesting that all authors identified as 'corresponding author' on published papers create and link their Open Researcher and Contributor Identifier (ORCID) with their account on the Manuscript Tracking System (MTS), prior to acceptance. ORCID helps the scientific community achieve unambiguous attribution of all scholarly contributions. You can create and link your ORCID from the home page of the MTS by clicking on 'Modify my Springer Nature account'. For more information please visit www.springernature.com/orcid.

[redacted]

We hope that you will find our referees' comments and editorial guidance helpful. Please do not hesitate to contact me if there is anything you would like to discuss. Thank you again for thinking of NCB for this work, and I apologize again for the delay in sending our decision to you.

Best wishes,

Melina

Melina Casadio, PhD

Senior Editor, Nature Cell Biology
ORCID ID: <https://orcid.org/0000-0003-2389-2243>

Reviewers' Comments:

Reviewer #1:

Remarks to the Author:

Summary of results: the study by Son and colleagues reports a nutrient sensitive system for the activation of mTORC1 activity. This contends that nutrient-mediated regulation of p300 localisation results in acetylation of RAPTOR that activates mTORC1. This mechanism hinges on AMPK-mediated phosphorylation of p300 for the negative regulation of the proposed p300-mTORC1 axis, and the nutrient-mediated activation of PP2A for nuclear efflux of p300 and re-establishment of mTORC1 activity. This system is dysfunctional in a progeria caused by mutation of Lamin A (Hutchinson-Gilford Progeria Syndrome), and this results in activation and loss of nutrient-mediated regulation of mTORC1 activity. The study by Son et al provides a mechanism for mTORC1 activation in this context. mTORC1 activation is a good explanation for the pro-ageing effects observed upon mutation of Lamin A, as mTORC1 accelerates biological ageing. mTORC1 activity accelerates biological ageing in part (but likely not solely) through suppression of autophagy.

This study provides a novel mechanistic explanation for this progeria, but also a novel and coherent nutrient-sensing mechanism for the regulation of mTORC1. The identification of nutrient sensing mechanisms for the activation of mTORC1 and the suppression of autophagy are important because they will likely form the basis of interventions that delay the onset of age-related disease in the future.

On originality: elements of the mechanism presented in the current study do of course appear in the literature, as has been cited in the current study. P300 has been reported to acetylate RAPTOR (at a specific residue) to regulate mTORC1 activity (PMID: 30197302). This topic was reviewed by the authors in 2021 (PMID: 33483607). Further, the link between p300 and autophagy has already been noted (PMID: 24560926). Although previous work has investigated AMPK, EP300, and autophagy (PMID: 24560926), the mechanistic links between AMPK and EP300 were speculative in nature the firm molecular links presented in the current study hadn't been established. Indeed the inspiration for the current study for investigation of S89 in EP300 as an AMPK target (PMID: 21940946) dealt with entirely different biology from that considered by the current study. The interaction between CRM1 and EP300 was taken from an 'omics resource paper (PMID: 26673895). This paper was broad in scope and didn't delve into the specific EP300-CRM1 relationship so I don't think this paper diminishes the novelty of the current study. The interaction between PP2A was previously documented (PMID: 15632055), but in a wildly different context. Likewise for the relationship between p300 and 14-3-3 proteins (PMID: 33799418, PMID: 20141511 that is a large unspecific bioinformatic research paper). mTORC1 activation in a model of this progeria has also been noted before (PMID: 34453483).

Although some elements of the study have been investigated before, I argue that the current study by Son and colleagues presents a novel coherent mechanism that when pulled together appears broadly relevant for research on ageing, and age-related disease. I expect the current study will attract a lot of citations because it presents a highly experimentally tractable system that should garner a lot of interest. 'Uptake' of this paper will be further enhanced because it demonstrates external validity – use of animal tissues especially and different cell lines was great to see.

Data and methodology: the data in this manuscript overall was of high quality, with striking examples of how AMPK regulates subcellular localisation of p300 in particular. The use of animal tissues was appreciated as noted above. One area in the manuscript that needs strengthening is how autophagy is analysed. I only say this here because autophagy is used as an explanatory mechanism for how the p300-mTORC1 axis leads to enhanced biological ageing. Autophagy is even invoked in the concluding paragraph. Specifically with regards to extended data figure 2 – a flux assay needs to be done here, say +/- p300, +/- bafilomycin and WB for LC3BII. A flux assay needs to be performed because although the current results are suggestive, they do have more than one interpretation.

Statistics: although data in this study is generally of high quality, statistical tests that are employed are likely inappropriate. For example, on numerous occasions, the controls are 'set to 1' and obviously paired data are divided by the cognate control and plotted accordingly. This removes all variation from the control group and therefore violates several underpinning assumptions of t-tests (equal variance, and normal distribution). I understand it is a common practice in cell biology, but it is still wrong.

For paired data, perhaps a paired t-test on uncorrected data could be used. If you choose to divide data by cognate controls to set controls to 1, a one-sample t-test that compares the test data set to 1 could be more appropriate. You can do both in Prism. Further, non-parametric tests do not require normal distributions – something to consider. However, I'm not a statistician and it would be best to consult a suitably qualified person on this matter.

Multiple comparisons are also not addressed in a statistically appropriate manner – for example in Fig. 3G, multiple comparisons are made using a t-test. There is no correction for multiple comparisons. This also needs to be addressed.

Conclusions: the conclusions of this manuscript are easy to follow, and I can draw direct lines between conclusions made and supporting evidence presented in the paper.

Further suggested improvements: a key part of the mechanism proposed in this paper posits that p300 activates mTORC1 by acetylating RAPTOR, and this suppresses autophagy. The authors convincingly show that p300 acetylates RAPTOR, and this is modified by relevant nutrient-related conditions, especially in conjunction with a delta-NLS p300 mutant. What I couldn't find in the paper were experiments where lysines on RAPTOR were mutated to prove unequivocally that acetylation of RAPTOR was responsible for the downstream mechanisms proposed by the paper. This is important because p300 has many substrates, and maybe acetylation of one of those other substrates was responsible for the effect on autophagy for example. Previous work from the authors showed p300 acetylates K1097 on RAPTOR and this is required for mTORC1 activation (PMID: 32561715). What I couldn't find was a link between mutation of this residue and autophagic flux. If the relationship between K1097 acetylation and autophagic flux has already been demonstrated in the literature, please state this in the manuscript. If not, I would really appreciate the inclusion of this experiment to tie important parts of the mechanism together.

In paragraph one of the discussion you state "While it may not be the sole determinant of mTORC1-autophagy responses in this context, it is clearly rate-limiting and appears to be relevant in diverse cell types in culture and diverse tissues of mice after fasting or refeed after fasting.". You hint here at what I thought was an elephant in the room with regards to nutrient-based regulation of mTORC1 and autophagy. I understand you lightly touched on this in the introduction, but could you please expand

on this point and introduce nutrient sensors like Sestrin2, and clearly elaborate on what you think the relationship is between the GATOR-RAGULATOR-RAGs axis and the AMPK-p300-RAPTOR axis is.

Kind regards

Tim Sargeant

Reviewer #2:

Remarks to the Author:

Min Son et al. manuscript describes how depletion of nutrients such as AA and glucose levels causes nucleus shuttling of p300. Increased nucleus p300 level decreases mTORC1 acetylation and, therefore, its activity, eventually activating the autophagy pathway. Lack of nutrients activates AMPK, which phosphorylates p300 at ser89. Phosphorylated p300 is transferred into the nucleus and decreases acetylation of mTORC1, which further activates autophagy. They also showed that the addition of nutrients to the starved cells activates phosphatase 2A which de-phosphorylates nuclear p300 and that causes its export to the cytoplasm. This shuttling regulates mTORC1 reactivation. The authors emphasized that this shuttling is altered in progeria patients, too, due to the presence of progerin which causes mTORC1 activation and autophagy inhibition.

The authors started testing their hypothesis by showing the importance of p300 in mTORC1 regulation. They effectively included many different controls and showed the rescue effect by restoring AAs in the media. including NLS deleted P300 construct and further showing this pattern in many different cell lines made their aim stronger. They showed this pattern in vivo using cytoplasmic and nuclear fractions of the liver and muscle and the brain of fasted versus refed mice. Using different inhibitors to kinases that can phosphorylate p300, they showed AMPK inhibition inhibited nuclear transport of p300 and its activation induces the nuclear shuttling of p300. Using additional experiments, including AMPK KD and pull-down, the authors further strengthened these claims. Next, these authors investigated how p300 gets transported back to cytoplasm after AA addition, focusing on PP2A based on LC-MC data and CRM1 (exportin). In the last two figures, the authors examined p300 in the progeria cells, in which, dysregulated mTOR activities have been observed previously.

This was a well-written report on thorough research. The data appear appropriate for testing the authors' hypothesis, and they have a straightforward research question and use appropriate methods to test this hypothesis. By channeling their findings into the progeria field, they also provided mechanistic insights into previous findings. Their studies supported the therapeutic application of mTOR inhibitors, such as rapamycin and its analogs, in treating progeria.

Concerns:

1. Many key data were shown in extended data figures (ED Figs), and the number of extended figures exceeded the main figure numbers.
2. In ED fig1, it is unclear the rationale underlying picking clone #2 of p300 k/o for future analysis. The experiments were conducted in an unstable HeLa cell line. What about other clones? Have the authors analyzed the potential off-target effects of CRISPR? In my view, ED figs 1 & 2 are important and can be moved to the main figures.

3. In Figure 4, the peak of PP2A and p300 binding occurred 15 mins after AA addition, and this association decreased afterward. Why?
4. A normal human fibroblast of a 3-year-old boy was used as a control. It was not age-matched or gender controlled. In the ideal situation, an ABE-corrected patient cell lines should be used.
5. In Fig 7, Based on the data enough evidence was provided to believe that progerin is degraded by activation of autophagy (which was also reported by others). However, the claim that progerin itself inhibits autophagy just because of p300 mislocalization is not consistent with the data presented.
6. Please discuss why only specific AA shortage in the media can induce p300 mislocalization..

Reviewer #3:

Remarks to the Author:

In this article Son et al, build on their previous findings showing that the acetyl-CoA/p300 axis may play a role in leucine sensing by mTORC1 by providing evidence suggesting that nucleo-cytoplasmic shuttling of p300 may play a role in amino acid- or glucose-induced activation of mTORC1. The authors propose a model whereby glucose or amino acid depletion sequesters p300 in the nucleus, thereby decreasing acetylation of raptor, reducing mTORC1 activity and bolstering autophagy. Some evidence is provided that nucleo-cytoplasmic shuttling of p300 may be regulated by AMPK and PP2A mediated phosphorylation and dephosphorylation of p300, respectively. Finally, Son et al., present findings suggesting that nucleo-cytoplasmic transport of p300 may be perturbed in Hutchinson-Gilford Progeria Syndrome that is characterized by elevated mTORC1 levels. Overall, it was thought that this study is of high potential interest to a broad spectrum of research fields inasmuch as it highlights a hitherto unappreciated mechanisms of mTORC1 activation with the potential implication in human diseases. Moreover, this study was thought to provide alternative explanation to some long-standing paradigms in mTORC1 signaling that were derived from limited experimental systems that were mostly based on HEK293 cell line variants. In general, it was thought that the experimental evidence adequately supports authors' conclusions. Nonetheless, it was thought that several apparent issues should be addressed to further improve data quality, experimental rigor and further strengthen the proposed model. These specific comments and concerns were outlined below.

Major comments:

-Control blots to estimate levels of exogenous p300 variants are missing in a number of figures (e.g., Fig 1e, 3g. etc). It was thought that these control western blots were warranted to exclude potential confounding stemming from different levels of overexpressed p300 variants.

-Quality of the western blots throughout the article seems inconsistent, whereby some blots appear to be of sub-par quality (e.g., Fig. 1e – total S6K1; Fig 4a, PP2Ac, Extended Data Fig. 1c, p300 etc.).

-In the extended figure 2d other KATs were depleted using siRNA-based approach and this was compared to the effects of p300 KO on mTORC1 activity. Considering that the depletion of other KATs does not seem to be that strong (i.e., a significant amount of targeted protein is still present), it was thought that the direct comparisons with the complete KO of p300 is somewhat inappropriate. To this end, the conclusion that other KATs are not involved should be toned down. Alternatively, an experiment comparing the effects of similar extent of depletion of p300 and other KATs on mTORC1 signaling should be performed. Finally, basal mTORC1 activity appears to be reduced in cells in which CBP was depleted by RNAi (extended data figure 2E, lane 1 vs. 4). This should be commented on in

the text.

-Fig. 1h/5g– appear to show different expression levels of the two p300 mutants (anti-His-tag Western blots). This should be perhaps noted and/or commented on.

-In the light of some recent studies, the authors should exclude the possibility that modulating p300 levels and/or activity may affect uptake and/or transport of glucose or amino acids (e.g., Martins et al, JCI insight 2022) and thereby affect mTORC1 signaling indirectly.

-Considering the multifaceted nature of mTOR signaling, it appears that there should be at least some consideration that mTORC1 outputs other than autophagy may contribute to the ensuing phenotypes. Indeed, the support for the role of autophagy in HGPS cellular phenotypes is based on correlative data and use of highly pleiotropic Bafilomycin A1 (that this same group has demonstrated may interfere with proteasomal protein degradation that is also affected by mTORC1). To this end, it was thought that the implication of autophagy should be somewhat toned down, or that additional evidence should be provided to support this conclusion.

-Most of the readouts of mTORC1 activity are limited to S6K1/rpS6 phosphorylation. Considering the emerging findings showing potential modularity of mTORC1 signaling, it is advised that other mTORC1 substrates (e.g., 4E-BP1) are also used for the readout of mTORC1 activity at least in the initial experiments. It should also be noted that rpS6 p-S235/236 Ab is not a very good readout of mTORC1 signaling as this site is also phosphorylated by AGC kinases other than S6Ks including RSKs. Accordingly, the phosphorylation of rpS6 at this site is not affected in cells that are devoid of S6Ks (Pende et al MCB 2004).

-A number of rescue experiments appear to be based on transient overexpression of e.g., pcDNA-based constructs. If this is true, this should be clearly indicated in the text considering the well-recognized issues associate with this approach. Moreover, most of the interactions/associations appear to be investigated using forced expression of exogenous versions of tagged proteins. This should be clearly indicated in the text as otherwise the readers may be led to believe that these interactions/associations were established using endogenous proteins.

Minor concerns:

-It appears that in some cases the increase in cytoplasmic p300 is not accompanied by corresponding decrease in nuclear p300 upon amino acid or glucose repletion. What happens to total p300 levels upon amino acid or glucose repletion?

-The authors should likely discuss why the focus was on glucose and amino acids and not on other stimuli known to alter mTORC1 signaling.

-The article may benefit from some careful editing. E.g., Page 4, line 101, “the” seems to be misplaced and should be moved to read “Previously, depletion of leucine (Leu) was shown to inhibit mTORC1 activity by multiple pathways, including the Leu-MCCC1-p300 axis.”; Page 4, line 129, a space is missing between “regulate” and “mTORC1”. etc.

-Page 10, lines 304-306, “Progerin-mediated DNA damage (assessed using an antibody against the DNA double strand break marker γ -H2AX) in HGPS fibroblasts was also reduced by treatment with

A485, 991 (AMPK activator) or Selinexor (SEL; nuclear export inhibitor) (Fig. 7b)". The reference to Figure 7b does not seem appropriate here, as the indicated compounds appear to be used in the experiments shown in Figure 7c, where LMNB, not γ -H2AX, is assessed.

-Figure legends are missing detail. For instance, it should be indicated that the p300 constructs in figure 1a are His-tagged, etc. In general, it was thought that it would be beneficial to indicate the name of the proteins and not just tags in the figures. Specific phosphorylation sites of substrates assessed by immunoblotting should also be indicated throughout the article.

-Employing SEM for Western blot densitometry data generated from the replicates using the same cell line was also thought to be somewhat dubious and likely to inflate the p values, but it was thought that the differences between experimental conditions were rather clear, and thus this was not a major source of concern. In addition, some of the statistical tests that were employed are not identified in neither the figure legends nor methods.

-Conclusion that upregulation of CRM1 and lower AMPK activity are responsible to increased mTORC1 activity in HGPS fibroblasts was thought to be somewhat premature considering the evidence, and previous literature showing that AMPK can suppress mTORC1 signaling in a variety of ways, whereby it was thought that the authors do not provide sufficient data to conclude that the effects of dampened AMPK signaling on mTORC1 are exclusively mediated by p300, in the context of HGPS. Some toning down of the statements in this section of the manuscript appears to be warranted.

Curious question:

-I understand that this is out of the scope of the manuscript, but did the authors attempt to establish the role of p300 in mTORC1 regulation in TSC1 or 2 KO cells in e.g., response to glucose depletion and repletion? This again is a personal curiosity of the reviewer (not associated with the evaluation of this current article) and is based on some previous findings showing that p300 may be increased in TSC2 KO cells, whereby AMPK was thought to act via TSC at least in some contexts.

I hope that the authors will find these comments constructive and of sufficient pathos.

Sincerely

I/Topisirovic

READABILITY OF MANUSCRIPTS – Nature Cell Biology is read by cell biologists from diverse backgrounds, many of whom are not native English speakers. Authors should aim to communicate

their findings clearly, explaining technical jargon that might be unfamiliar to non-specialists, and avoiding non-standard abbreviations. Titles and abstracts should concisely communicate the main findings of the study, and the background, rationale, results and conclusions should be clearly explained in the manuscript in a manner accessible to a broad cell biology audience. Nature Cell Biology uses British spelling.

REFERENCES – are limited to a total of 70 for Articles, Resources, Technical Reports; and 40 for Letters. This includes references in the main text and Methods combined. References must be numbered sequentially as they appear in the main text, tables and figure legends and Methods and must follow the precise style of Nature Cell Biology references. References only cited in the Methods should be numbered consecutively following the last reference cited in the main text. References only associated with Supplementary Information (e.g. in supplementary legends) do not count toward the total reference limit and do not need to be cited in numerical continuity with references in the main

text. Only published papers can be cited, and each publication cited should be included in the numbered reference list, which should include the manuscript titles. Footnotes are not permitted.

Methods should be written concisely, but should contain all elements necessary to allow interpretation and replication of the results. As a guideline, Methods sections typically do not exceed 3,000 words. The Methods should be divided into subsections listing reagents and techniques. When citing previous methods, accurate references should be provided and any alterations should be noted. Information must be provided about: antibody dilutions, company names, catalogue numbers and clone numbers for monoclonal antibodies; sequences of RNAi and cDNA probes/primers or company names and catalogue numbers if reagents are commercial; cell line names, sources and information on cell line identity and authentication. Animal studies and experiments involving human subjects must be reported in detail, identifying the committees approving the protocols. For studies involving human subjects/samples, a statement must be included confirming that informed consent was obtained. Statistical analyses and information on the reproducibility of experimental results should be provided in a section titled "Statistics and Reproducibility".

All Nature Cell Biology manuscripts submitted on or after March 21 2016 must include a Data availability statement at the end of the Methods section. For Springer Nature policies on data availability see <http://www.nature.com/authors/policies/availability.html>; for more information on this particular policy see <http://www.nature.com/authors/policies/data/data-availability-statements-data-citations.pdf>. The Data availability statement should include:

- Accession codes for primary datasets (generated during the study under consideration and designated as "primary accessions") and secondary datasets (published datasets reanalysed during the study under consideration, designated as "referenced accessions"). For primary accessions data should be made public to coincide with publication of the manuscript. A list of data types for which submission to community-endorsed public repositories is mandated (including sequence, structure, microarray, deep sequencing data) can be found here <http://www.nature.com/authors/policies/availability.html#data>.
- Unique identifiers (accession codes, DOIs or other unique persistent identifier) and hyperlinks for datasets deposited in an approved repository, but for which data deposition is not mandated (see here for details <http://www.nature.com/sdata/data-policies/repositories>).
- At a minimum, please include a statement confirming that all relevant data are available from the authors, and/or are included with the manuscript (e.g. as source data or supplementary information), listing which data are included (e.g. by figure panels and data types) and mentioning any restrictions on availability.
- If a dataset has a Digital Object Identifier (DOI) as its unique identifier, we strongly encourage including this in the Reference list and citing the dataset in the Methods.

We recommend that you upload the step-by-step protocols used in this manuscript to the Protocol Exchange. More details can be found at www.nature.com/protocolexchange/about.

All imaging data should be accompanied by scale bars, which should be defined in the legend. Cropped images of gels/blots are acceptable, but need to be accompanied by size markers, and to retain visible background signal within the linear range (i.e. should not be saturated). The boundaries of panels with low background have to be demarked with black lines. Splicing of panels should only be considered if unavoidable, and must be clearly marked on the figure, and noted in the legend with a statement on whether the samples were obtained and processed simultaneously. Quantitative comparisons between samples on different gels/blots are discouraged; if this is unavoidable, it should only be performed for samples derived from the same experiment with gels/blots were processed in parallel, which needs to be stated in the legend.

The total number of Supplementary Figures (not including the "unprocessed scans" Supplementary Figure) should not exceed the number of main display items (figures and/or tables (see our Guide to

Authors and March 2012 editorial <http://www.nature.com/ncb/authors/submit/index.html#suppinfo>; <http://www.nature.com/ncb/journal/v14/n3/index.html#ed>). No restrictions apply to Supplementary Tables or Videos, but we advise authors to be selective in including supplemental data.

GUIDELINES FOR EXPERIMENTAL AND STATISTICAL REPORTING

REPORTING REQUIREMENTS – To improve the quality of methods and statistics reporting in our papers we have recently revised the reporting checklist we introduced in 2013. We are now asking all life sciences authors to complete two items: an Editorial Policy Checklist (found here <https://www.nature.com/authors/policies/Policy.pdf>) that verifies compliance with all required editorial policies and a reporting summary (found here <https://www.nature.com/authors/policies/ReportingSummary.pdf>) that collects information on experimental design and reagents. These documents are available to referees to aid the evaluation of the manuscript. Please note that these forms are dynamic 'smart pdfs' and must therefore be downloaded and completed in Adobe Reader. We will then flatten them for ease of use by the reviewers. If you would like to reference the guidance text as you complete the template, please access these flattened versions at <http://www.nature.com/authors/policies/availability.html>.

Author Rebuttal to Initial comments

9 October 2023

Dear Dr Casadio

RE: NCB-A51241

Many thanks for your email of 28 June 2023 regarding this paper. We have tried to address all comments from the reviewers. First we have addressed the specific comments that you highlighted in your letter. After this we have responded to each comment in order – thus, the comments/suggestions that you highlighted are duplicated when we consider the comments point-by-point.

We hope that our revisions are satisfactory.

KEY POINTS HIGHLIGHTED BY EDITOR:

In our view, it will be essential to address the reviews as follows:

A. The link between p300, mTORC1, and autophagy, including in the context of HGPS needs to be strengthened with additional data, as recommended by all reviewers:

Rev#1 "One area in the manuscript that needs strengthening is how autophagy is analysed. I only say this here because autophagy is used as an explanatory mechanism for how the p300-mTORC1 axis leads to enhanced biological ageing. Autophagy is even invoked in the concluding paragraph. Specifically with regards to extended data figure 2 – a flux assay needs to be done here, say +/- p300, +/- bafilomycin and WB for LC3BII. A flux assay needs to be performed because although the current results are suggestive, they do have more than one interpretation."

→ To assess autophagic flux in p300 knockout cells compared to control cells, we performed three methods: 1) Treatment with bafilomycin A1 (BafA1), 2) SRAI-LC3B assay, and 3) Measurement of degradation of autophagic substrates, specifically Htt Q74. As expected, p300 knockout cells exhibited elevated levels of LC3-II under both basal conditions and following treatment with BafA1. We added these data and text to the manuscript (see Extended Data Figure 2a; page 4, line 22).

The SRAI-hLC3B assay utilizes the SRAI reporter, a tandem construct comprising TOLLES (a blue fluorescent protein resistant to acid-denaturation and proteolysis) and YPet (a yellow fluorescent protein, which undergoes acid-denaturation and proteolysis in lysosomes).

The principle behind this assay involves the delivery of the SRAI reporter to lysosomes, where YPet degradation occurs. This process leads to a noticeable shift in fluorescence of the tandem construct, as the FRET-associated quenching of the TOLLES signal is relieved following YPet degradation. To adapt this tool for autophagy analysis, we fused the SRAI reporter to the N-terminus of LC3B, a well-established autophagy marker protein. The ratio of blue to yellow fluorescence now reflects the proportion of LC3B undergoing lysosomal degradation. This assay was validated in our previous study (Wrobel et al., 2022, Nature Communications). Notably, the SRAI-LC3B expression in p300 KO cells clearly results in an elevated TOLLES:YPet ratio (see Extended Figure 2d; page 4, line 29), indicating autophagy flux activation in response to p300 KO.

In Extended Figure 2e, we observed that the autophagic substrate Htt Q74 underwent more significant degradation in p300 knockout (KO) cells, as well as when treated with the p300 inhibitor A485. (see Extended Figure 2e; page 5, line 6)

Rev#2 "In Fig 7, Based on the data enough evidence was provided to believe that progerin is degraded by activation of autophagy (which was also reported by others). However, the claim that progerin itself inhibits autophagy just because of p300 mislocalization is not consistent with the data presented."

➔ We agree with the reviewer's comment – it is very difficult to show that the p300 mislocalisation is the only cause of impaired autophagy caused by progerin, partly because progerin itself is degraded by autophagy. We have added some text to the discussion to make this clear (page 14, line 5).

"However, it is very difficult to show that the p300 mislocalisation is the only cause of impaired autophagy caused by progerin, partly because progerin itself is degraded by autophagy (Extended Data Fig. 10b). Thus, while we can rescue a number of mTORC1- and autophagy-related phenotypes in HGPS cells and models with strategies that reduce cytoplasmic p300 levels, we can not exclude additional mechanisms whereby progerin may compromise autophagy."

We have also amended the abstract (page 1, line 28).

"These results reveal how nutrients regulate mTORC1, a cytoplasmic complex, by shuttling its positive regulator p300 in and out of the nucleus and how this pathway is misregulated in HGPS contributing to mTORC1 hyperactivation and defective autophagy."

We have added a new experiment (Extended Fig. 10d) to try to further fortify our data in this context. Here, we have looked that the levels of progerin (an autophagy substrate) in wild-type, p300 knockout cells and p300 knockout cells reconstituted with wild-type (WT), cytoplasmic (dNLS) and nuclear (IA; exportin binding deficient) p300. It is clear that the effects of p300 absence are rescued by the wild-type and cytoplasmic p300 but not nuclear-only p300. However, while these results suggest that p300 shuttling can indeed modulate progerin levels through its influence on autophagic activity, as mentioned above, this does not exclude other potential mechanisms.

Extended Fig. 10d. Regulation of progerin degradation in p300 KO expressing p300 WT, dNLS or I265A (IA). Blots are representative of three biologically independent experiments (N=3). * p < 0.05 vs. control cells; ## p < 0.01, ### p < 0.001 vs. p300 KO cells (one-way ANOVA with post hoc Tukey test).

Rev3 "Considering the multifaceted nature of mTOR signaling, it appears that there should

be at least some consideration that mTORC1 outputs other than autophagy may contribute to the ensuing phenotypes. Indeed, the support for the role of autophagy in HGPS cellular phenotypes is based on correlative data and use of highly pleiotropic Bafilomycin A1 (that this same group has demonstrated may interfere with proteasomal protein degradation that is also affected by mTORC1). To this end, it was thought that the implication of autophagy should be somewhat toned down, or that additional evidence should be provided to support this conclusion."

→ We agree reviewer's comment – we have now stated this explicitly in the discussion (page 14, line 12).

"It is important to consider that processes other than autophagy that are regulated by mTORC1 may also contribute to HGPS cellular phenotypes."

B. The reviewers additionally felt that other aspects of the proposed mechanism of mTORC1 regulation needed to be evaluated more carefully:

Rev#1 "a key part of the mechanism proposed in this paper posits that p300 activates mTORC1 by acetylating RAPTOR, and this suppresses autophagy. The authors convincingly show that p300 acetylates RAPTOR, and this is modified by relevant nutrient-related conditions, especially in conjunction with a delta-NLS p300 mutant. What I couldn't find in the paper were experiments where lysines on RAPTOR were mutated to prove unequivocally that acetylation of RAPTOR was responsible for the downstream mechanisms proposed by the paper. This is important because p300 has many substrates, and maybe acetylation of one of those other substrates was responsible for the effect on autophagy for example. Previous work from the authors showed p300 acetylates K1097 on RAPTOR and this is required for mTORC1 activation (PMID: 32561715). What I couldn't find was a link between mutation of this residue and autophagic flux. If the relationship between K1097 acetylation and autophagic flux has already been demonstrated in the literature, please state this in the manuscript. If not, I would really appreciate the inclusion of this experiment to tie important parts of the mechanism together."

→ Thank you for your comment. In our previous work (Son et al. 2020, Nature Communications; PMID: 32561715), we observed that the K1097R mutant of raptor induced autophagy activation as a result of mTOR inhibition even in cells treated with the p300 activator CTB or after refeeding following starvation. This observation, along with other data in that paper, shows that p300-mediated regulation of autophagy is primarily driven by raptor acetylation at K1097.

[REDACTED]

We have now specified this biology in the introduction (page 2, line 21).

“Elevated acetyl-CoA levels stimulated mTORC1 activity and inhibited macroautophagy (henceforth autophagy) by inducing acetylation of raptor at K1097, which is necessary for its binding to Rag proteins and mTORC1 lysosomal localization, a requirement and driver for mTORC1 activation in response to nutrient replenishment. p300, rather than other lysine acetyltransferases (KATs), was a critical mediator of this process by regulating raptor acetylation. Indeed, mutating K1097 in raptor prevented the mTORC1 activation and autophagy inhibition that were otherwise seen in response to p300 activation or refeeding after starvation.”

Rev#2 points #3, 6

Rev#2 points #3) In Figure 4, the peak of PP2A and p300 binding occurred 15 mins after AA addition, and this association decreased afterward. Why?

→ Our data indicates that when amino acids are replenished to starved cells, there is an increase in the activity of the PP2A phosphatase (as shown in Fig. 4c, d). This increase leads to the dephosphorylation of p300 at Ser89, even within a short timeframe of 15-30 min. As a result of this dephosphorylation, the p300 interaction with PP2A is decreased, while p300 interacts better with CRM1 and can be exported from the nucleus to the cytoplasm. Therefore, we can conclude that the interaction between p300 and PP2Ac occurs very soon after amino acid re-stimulation, and this interaction leads to the dephosphorylation of p300 and might subsequently result in the loss of their interaction. We added this observation to the discussion (page 12, line 30).

“Export from the nucleus after nutrient replenishment is mediated by dephosphorylation of p300 by PP2A (even within a short timeframe of 15-30 min), which decreases the binding of p300 and PP2A (Fig. 4a) while increasing the interaction of p300 with with the exportin, CRM1 (Fig. 5b).”

Rev#2 points #6) Please discuss why only specific AA shortage in the media can induce p300 mislocalization.

→ The levels of Acetyl-CoA (Ac-CoA) play a crucial role in regulating p300 activity. When considering the branched-chain amino acids (BCAAs; Leu, Ile and Val, it is noteworthy that Val, unlike Leu and Ile, is recognized for its inability to generate Acetyl-CoA (the effector of mTORC1 regulation in this context), as reported by Adeva-Andany et al. in 2017. As depicted in Figure 3g, depleting Valine did not result in a reduction of mTORC1 activity. We added the text into the manuscript (page 8, line 18).

“To investigate which AAs regulate mTORC1 through AMPK, we tested single AA-depleted media (Arg-, Leu-, Ile- or Val-) on cells expressing p300 WT or SA. Depletion of Arg, Leu or Ile but not Val activated AMPK (p-ACC at Ser79 / total ACC) and inhibited mTORC1 in p300 WT expressing cells, but p300 SA expression blunted the mTORC1 inhibition by single AA depletion (Fig. 3g)”

In Figure 3g, it is evident that the depletion of Arginine (Arg), Leucine (Leu), or Isoleucine (Ile), but notably not Valine (Val), induces AMPK activation. Our study has uncovered that AMPK plays a role in mediating p300 nuclear transport in response to starvation, potentially leading to p300 accumulation in the nucleus.

Furthermore, there is existing literature supporting the activation of AMPK in response to the depletion of single amino acids. For instance, a previous study has demonstrated that Arg depletion can activate the AMPK signaling pathway in MEF cells (Carroll et al. 2016). Thus, the Arg-AMPK connection is likely an important reason why this amino acid regulates mTORC1 in response to its availability. We have added text to the discussion to summarise this issue (page 13, line 3).

“Indeed, our data suggest that both AMPK-PP2A and the BCAAs-acetyl-CoA pathways are rate-limiting for amino acid regulation of mTORC1 and autophagy. This likely explains why p300 shuttling and consequent mTORC1 and autophagy regulation are dependent on BCAAs (like Leu and Ile) that are catabolised to acetyl-CoA (a key effector of mTORC1) and why Arg, as an AMPK regulator is important for p300 shuttling.”

Rev#3:

"In the light of some recent studies, the authors should exclude the possibility that modulating p300 levels and/or activity may affect uptake and/or transport of glucose or amino acids (e.g., Martins et al, JCI insight 2022) and thereby affect mTORC1 signaling indirectly."

- ➔ To test the potential impairment in glucose uptake associated with depleting p300, we performed a glucose uptake assay in p300 KO cells. Our findings indicate that p300 KO cells did not show significant differences in glucose uptake levels when

compared to control cells. Additionally, the re-introduction of p300 constructs, including the wild-type (WT), dNLS, and S89A, did not have a noticeable impact on glucose uptake in both basal and insulin-treated conditions.

This implies that, instead of suppressing glucose uptake through p300 depletion, cytoplasmic-nuclear transport of p300 is important for mTORC1 regulation. We added these data and text to the manuscript (Extended fig. 3d; page 6, line 22).

“In a previous study, when p300 and CBP were simultaneously lost in skeletal muscle and adipocytes, it resulted in a complete absence of insulin-stimulated Glc uptake into cells. To explore the potential impairment in Glc uptake associated with depleting p300, we performed a Glc uptake assay in p300 KO cells. However, our findings indicate that p300 KO HeLa cells did not exhibit significant differences in Glc uptake levels when compared to control cells (Extended Data Fig. 3d). This implies that, instead of suppressing Glc uptake through p300 depletion, cytoplasmic-nuclear transport of p300 is important for mTORC1 regulation.”

Extended Data Fig. 3d. Glucose uptake assay in p300 KO cells expressing His-tagged WT, dNLS or S89A. Insulin (1 μ M) was used to stimulate glucose uptake into cells. N=4 in each group. *** $p < 0.001$ vs. control cells (one-way ANOVA with post hoc Tukey test).

“Most of the readouts of mTORC1 activity are limited to S6K1/rpS6 phosphorylation. Considering the emerging findings showing potential modularity of mTORC1 signaling, it is advised that other mTORC1 substrates (e.g., 4E-BP1) are also used for the readout of mTORC1 activity at least in the initial experiments. It should also be noted that rpS6 p-S235/236 Ab is not a very good readout of mTORC1 signaling as this site is also

phosphorylated by AGC kinases other than S6Ks including RSKs. Accordingly, the phosphorylation of rpS6 at this site is not affected in cells that are devoid of S6Ks (Pende et al MCB 2004)."

- ➔ We agree with reviewer's comment, so we have conducted additional experiments to include measurements of 4E-BP1 as an assessment of mTORC1 activity. We added these data and text to the manuscript, which can be found in Fig. 1a, as well as in Extended Fig. 1g and 1h.

"It appears that in some cases the increase in cytoplasmic p300 is not accompanied by corresponding decrease in nuclear p300 upon amino acid or glucose depletion. What happens to total p300 levels upon amino acid or glucose depletion?"

- ➔ As shown in Fig. 1e and Extended fig. 1g-j, the depletion of amino acids or glucose did not lead to any significant changes in p300 expression levels within total lysates. However, it resulted in a noticeable alteration in its subcellular localization, as shown in Figure 1e. It's important to note that p300 is primarily localized in the nucleus, and therefore, the Western blot data may not clearly demonstrate a decrease in cytoplasmic p300 accompanied by a corresponding increase in nuclear p300 in response to starvation. However, immunostaining images provide a more detailed view, indicating a reduction in cytoplasmic p300 and an increase in nuclear p300 upon starvation, as evidenced in Figure 1c, 1d, 3a, 3e, and 4f.

"The authors should likely discuss why the focus was on glucose and amino acids and not on other stimuli known to alter mTORC1 signaling."

- In response to the reviewer's comments, we have added additional text in the discussion to elucidate the other stimuli known to regulate mTORC1 activity. (page 13, line 10)

"In addition to nutrient status, mTORC1 also responds to intracellular and environmental stresses that hinder growth, such as hypoxia or DNA damage. Hypoxia inhibits mTORC1 partly by activating AMPK and also by inducing REDD1 (Regulated in DNA damage and development 1), which in turn activates TSC. The DNA damage response pathway inhibits mTORC1 by inducing p53 target genes, including the AMPK regulatory subunit (AMPK β) and PTEN. It's worth noting that other stimuli that regulate mTORC1 activity can also be influenced by AMPK. Therefore, it would be intriguing to investigate whether hypoxia or DNA damage can modulate mTORC1 by altering the localization of p300."

"Conclusion that upregulation of CRM1 and lower AMPK activity are responsible to increased mTORC1 activity in HGPS fibroblasts was thought to be somewhat premature considering the evidence, and previous literature showing that AMPK can suppress mTORC1 signaling in a variety of ways, whereby it was thought that the authors do not provide sufficient data to conclude that the effects of dampened AMPK signaling on mTORC1 are

exclusively mediated by p300, in the context of HGPS. Some toning down of the statements in this section of the manuscript appears to be warranted.'

→ We agree reviewer's comment and have added text to make this explicit. (page 11, line 1)

“These data suggest that upregulated CRM1 and less AMPK activation in HGPS cells induce more cytoplasmic p300 localization, which would contribute to mTORC1 activation and autophagy inhibition. While AMPK is rate-limiting for mTORC1 regulation in this context, it also impacts mTORC1 via other routes.”

C. Lastly, please pay special attention to the reviewers' concerns about the controls used (including Rev#2 point#4), statistical analyses, quality of the data and please address their other points aimed at strengthening the current dataset, clarifying and streamlining the manuscript, and enriching the discussion.

→ We agree with the reviewer's comment. Now we have added new data with an expanded set of control lines which are age- and gender-matched. The details of the fibroblasts we used can be found in Extended data Table 1 (See the table below).

Table 1. List of human dermal fibroblast cultures used in this study

No #	Culture	LMNA mutation	Donor age at biopsy	Gender	Origin
Control-#1	GM05565	None	3	M	Coriell
Control-#2	GM05565	None	3	M	Coriell
Control-#3	GM02036	None	11	F	Coriell
Control-#4	GM00969	None	2	F	Coriell
HGPS-#1	AG01972	G608G	14	F	Coriell
HGPS-#2	AG11513	G608G	8	F	Coriell

* Control-#1 and #2 both originate from the same individual (identified as #GM05565), however, they belong to different batches.

In the comparison between Control #3 and #4 and HGPS-#1 and #2, we assessed several parameters, including autophagy, mTORC1 activity, progerin levels (as shown in extended Figure 9i), as well as the intensity of LMNB1 (as depicted in extended

Figure 9j). As expected, our results revealed that HGPS fibroblasts displayed activated mTORC1, inhibition of autophagy (as observed in extended Figure 9i), and reduced LMNB1 expression levels (as shown in extended Figure 9j) when compared to age- and gender-matched controls. We added these data and text to the manuscript (see Extended Data Figure 9i, j and table 1; page 11, line 7).

Extended Fig. 9i, Reduced progerin expression levels in HGPS fibroblasts by treatment with p300 inhibitor A485 for 24 h. Blots are representative of three biologically independent experiments (N=3). Two-tailed t-test. *** $p < 0.001$ vs. HGPS-1 fibroblasts; # $p < 0.05$ vs. HGPS-1 fibroblasts (for progerin levels); * $p < 0.05$, *** $p < 0.001$ vs. control fibroblasts; # $p < 0.05$, ## $p < 0.001$ vs. vehicle treated HGPS fibroblasts (for LC3-II levels). j, Reduced Lamin B1 (LMNB1) levels in HGPS fibroblasts. Scale bar, 5 μ m. N=3, about 50 cells scored per condition.

D. Finally, please pay close attention to our guidelines on statistical and methodological reporting (listed below) as failure to do so may delay the reconsideration of the revised manuscript. In particular, please provide:

➔ This has been done and all source data are provided as a Source data file.

➔ This has been done and all source data are provided as a Source data file.

We would be happy to consider a revised manuscript that would satisfactorily address these points, unless a similar paper is published elsewhere, or is accepted for publication in Nature Cell Biology in the meantime.

- ensure that it conforms to our format instructions and publication policies (see below and www.nature.com/nature/authors/).

- provide a point-by-point rebuttal to the full referee reports verbatim, as provided at the end of this letter.

POINT-BY-POINT RESPONSES TO ALL REVIEWERS' COMMENTS:

Reviewer #1:

Remarks to the Author:

Summary of results: the study by Son and colleagues reports a nutrient sensitive system for the activation of mTORC1 activity. This contends that nutrient-mediated regulation of p300 localisation results in acetylation of RAPTOR that activates mTORC1. This mechanism hinges on AMPK-mediated phosphorylation of p300 for the negative regulation of the proposed p300-mTORC1 axis, and the nutrient-mediated activation of PP2A for nuclear efflux of p300 and re-establishment of mTORC1 activity. This system is dysfunctional in a progeria caused by mutation of Lamin A (Hutchinson-Gilford Progeria Syndrome), and this results in activation and loss of nutrient-mediated regulation of mTORC1 activity. The study by Son et al provides a mechanism for mTORC1 activation in this context. mTORC1 activation is a

good explanation for the pro-ageing effects observed upon mutation of Lamin A, as mTORC1 accelerates biological ageing. mTORC1 activity accelerates biological ageing in part (but likely not solely) through suppression of autophagy.

This study provides a novel mechanistic explanation for this progeria, but also a novel and coherent nutrient-sensing mechanism for the regulation of mTORC1. The identification of nutrient sensing mechanisms for the activation of mTORC1 and the suppression of autophagy are important because they will likely form the basis of interventions that delay the onset of age-related disease in the future.

On originality: elements of the mechanism presented in the current study do of course appear in the literature, as has been cited in the current study. P300 has been reported to acetylate RAPTOR (at a specific residue) to regulate mTORC1 activity (PMID: 30197302). This topic was reviewed by the authors in 2021 (PMID: 33483607). Further, the link between p300 and autophagy has already been noted (PMID: 24560926). Although previous work has investigated AMPK, EP300, and autophagy (PMID: 24560926), the mechanistic links between AMPK and EP300 were speculative in nature the firm molecular links presented in the current study hadn't been established. Indeed the inspiration for the current study for investigation of S89 in EP300 as an AMPK target (PMID: 21940946) dealt with entirely different biology from that considered by the current study. The interaction between CRM1 and EP300 was taken from an 'omics resource paper (PMID: 26673895). This paper was broad in scope and didn't delve into the specific EP300-CRM1 relationship so I don't think this paper diminishes the novelty of the current study. The interaction between PP2A was previously documented (PMID: 15632055), but in a wildly different context. Likewise for the relationship between p300 and 14-3-3 proteins (PMID: 33799418, PMID: 20141511 that is a large unspecific bioinformatic research paper). mTORC1 activation in a model of this progeria has also been noted before (PMID: 34453483).

Although some elements of the study have been investigated before, I argue that the current study by Son and colleagues presents a novel coherent mechanism that when pulled together appears broadly relevant for research on ageing, and age-related disease. I expect the current study will attract a lot of citations because it presents a highly experimentally tractable system that should garner a lot of interest. 'Uptake' of this paper will be further enhanced because it demonstrates external validity – use of animal tissues especially and

different cell lines was great to see.

Data and methodology: the data in this manuscript overall was of high quality, with striking examples of how AMPK regulates subcellular localisation of p300 in particular. The use of animal tissues was appreciated as noted above.

Q1-1) One area in the manuscript that needs strengthening is how autophagy is analysed. I only say this here because autophagy is used as an explanatory mechanism for how the p300-mTORC1 axis leads to enhanced biological ageing. Autophagy is even invoked in the concluding paragraph. Specifically with regards to extended data figure 2 – a flux assay needs to be done here, say +/- p300, +/- bafilomycin and WB for LC3BII. A flux assay needs to be performed because although the current results are suggestive, they do have more than one interpretation.

→ To assess autophagic flux in p300 knockout cells compared to control cells, we performed three methods: 1) Treatment with bafilomycin A1 (BafA1), 2) SRAI-LC3B assay, and 3) Measurement of degradation of autophagic substrates, specifically Htt Q74. As expected, p300 knockout cells exhibited elevated levels of LC3-II under both basal conditions and following treatment with BafA1. We added these data and text to the manuscript (see Extended Data Figure 2a; page 4, line 22).

The SRAI-hLC3B assay utilizes the SRAI reporter, a tandem construct comprising TOLLES (a blue fluorescent protein resistant to acid-denaturation and proteolysis) and YPet (a yellow fluorescent protein, which undergoes acid-denaturation and proteolysis in lysosomes).

The principle behind this assay involves the delivery of the SRAI reporter to lysosomes, where YPet degradation occurs. This process leads to a noticeable shift in fluorescence of the tandem construct, as the FRET-associated quenching of the TOLLES signal is relieved following YPet degradation. To adapt this tool for autophagy analysis, we fused the SRAI reporter to the N-terminus of LC3B, a well-established autophagy marker protein. The ratio of blue to yellow fluorescence now reflects the proportion of LC3B undergoing lysosomal degradation. This assay was

validated in our previous study (Wrobel et al., 2022, Nature Communications). Notably, the SRAI-LC3B expression in p300 KO cells clearly results in an elevated TOLLES:YPet ratio (see Extended Figure 2d; page 4, line 29), indicating autophagy flux activation in response to p300 KO.

In Extended Figure 2e, we observed that the autophagic substrate Htt Q74 underwent more significant degradation in p300 knockout (KO) cells, as well as when treated with the p300 inhibitor A485. (see Extended Figure 2e; page 5, line 6)

Q1-2) Statistics: although data in this study is generally of high quality, statistical tests that are employed are likely inappropriate. For example, on numerous occasions, the controls are 'set to 1' and obviously paired data are divided by the cognate control and plotted

accordingly. This removes all variation from the control group and therefore violates several underpinning assumptions of t-tests (equal variance, and normal distribution). I understand it is a common practice in cell biology, but it is still wrong.

For paired data, perhaps a paired t-test on uncorrected data could be used. If you choose to divide data by cognate controls to set controls to 1, a one-sample t-test that compares the test data set to 1 could be more appropriate. You can do both in Prism. Further, non-parametric tests do not require normal distributions – something to consider. However, I'm not a statistician and it would be best to consult a suitably qualified person on this matter. Multiple comparisons are also not addressed in a statistically appropriate manner – for example in Fig. 3G, multiple comparisons are made using a t-test. There is no correction for multiple comparisons. This also needs to be addressed.

→ We have meticulously reevaluated all the data in our study to conduct a thorough analysis using the appropriate statistical tests. Subsequently, we have added this information into the figure legends. We have employed One-sample t-tests, where appropriate and these are specified in the figure legends. (Please note that we have previously sought advice from a statistician regarding the most appropriate tests to use for biological replicates done on different days where the baseline control score can vary between experiments - e.g. the western blot data where exposure influences the quantification).

Conclusions: the conclusions of this manuscript are easy to follow, and I can draw direct lines between conclusions made and supporting evidence presented in the paper.

Q1-3) Further suggested improvements: a key part of the mechanism proposed in this paper posits that p300 activates mTORC1 by acetylating Raptor, and this suppresses autophagy. The authors convincingly show that p300 acetylates Raptor, and this is modified by relevant nutrient-related conditions, especially in conjunction with a delta-NLS p300 mutant. What I couldn't find in the paper were experiments where lysines on Raptor were mutated to prove unequivocally that acetylation of Raptor was responsible for the downstream mechanisms proposed by the paper. This is important because p300 has many substrates, and maybe acetylation of one of those other substrates was responsible for the effect on autophagy for example. Previous work from the authors showed p300 acetylates K1097 on Raptor and this is required for mTORC1 activation (PMID: 32561715). What I

couldn't find was a link between mutation of this residue and autophagic flux. If the relationship between K1097 acetylation and autophagic flux has already been demonstrated in the literature, please state this in the manuscript. If not, I would really appreciate the inclusion of this experiment to tie important parts of the mechanism together.

- ➔ Thank you for your comment. In our previous work (Son et al. 2020, Nature Communications; PMID: 32561715), we observed that the K1097R mutant of raptor induced autophagy activation as a result of mTOR inhibition even in cells treated with the p300 activator CTB or after refeeding following starvation. This observation, along with other data in that paper strongly suggests that p300-mediated regulation of autophagy is primarily driven by raptor acetylation at K1097.

[REDACTED]

We have now specified this biology in the introduction (page 2, line 21).

“Elevated acetyl-CoA levels stimulated mTORC1 activity and inhibited macroautophagy (henceforth autophagy) by inducing acetylation of raptor at K1097, which is necessary for its binding to Rag proteins and mTORC1 lysosomal localization, a requirement and driver for mTORC1 activation in response to nutrient replenishment. p300, rather than other lysine acetyltransferases (KATs), was a critical mediator of this process by regulating raptor acetylation. Indeed, mutating K1097 in raptor prevented the mTORC1

activation and autophagy inhibition that were otherwise seen in response to p300 activation or refeeding after starvation.”

Q1-4) In paragraph one of the discussion you state “While it may not be the sole determinant of mTORC1-autophagy responses in this context, it is clearly rate-limiting and appears to be relevant in diverse cell types in culture and diverse tissues of mice after fasting or refeed after fasting.”. You hint here at what I thought was an elephant in the room with regards to nutrient-based regulation of mTORC1 and autophagy. In understand you lightly touched on this in the introduction, but could you please expand on this point and introduce nutrient sensors like Sestrin2, and clearly elaborate on what you think the relationship is between the GATOR-RAGULATOR-RAGs axis and the AMPK-p300-RAPTOR axis is.

→ We added additional text in the introduction following the reviewer’s helpful suggestion comments (page 2, line 28).

“ In most cell lines and primary cells, this is the dominant mechanism regulating mTORC1 and autophagy in response to branched chain amino acids (BCAAs; like leucine) starvation and refeeding. While inhibition of the enzymes that catabolise leucine to acetyl-CoA, or mutation of K1097 in raptor abrogated the switches in mTORC1 and autophagy in response to starvation and refeeding of these amino acids, these effects still occurred in such cell types when previously-described leucine sensors in HEK293 cells (like Sestrin2) were depleted. However, the pathway we have described is not limiting in rare cell types like HEK293 cells or mouse embryonic fibroblasts (MEFs).”

We have now expanded the discussion to consider the requirement for both the Leucine-acetyl-CoA and AMPK-PP2A pathways for mTORC1 and autophagy regulation in response to amino acid abundance (page 12, line 15).

“Multiple studies conducted by our research group and others have underscored the importance of post-translational modifications of raptor, including acetylation, in regulating mTORC1. These modifications influence the binding of raptor to Rag GTPases on the lysosomal membrane, which enables mTORC1 activation. Raptor primarily resides in the cytoplasm and can be acetylated by the p300 acetyltransferase at Lys1097. While p300 is predominantly localised in the nucleus, it can shuttle between the

nucleus and cytoplasm. The dynamic shuttling of p300 between the nucleus and cytoplasm assumes a critical role in facilitating its interaction with raptor and, consequently, the acetylation of raptor. However, the mechanisms regulating this shuttling and the consequences of shifting p300 localisation were not well understood. Previous studies have shown that p300 activates mTORC1, and we confirm the expectation that this is mediated by cytoplasmic p300 (Fig. 7i), suggesting that p300 cytosolic localization is essential for AcCoA-mediated mTORC1 activation. Our data suggest that p300 shuttling from the cytoplasm to the nucleus in response to nutrient (amino acids or glucose) deprivation is regulated by AMPK-dependent phosphorylation of p300, which enables 14-3-3 binding and nuclear entrapment of p300. Export from the nucleus after nutrient replenishment is mediated by dephosphorylation of p300 by PP2A (even within a short timeframe of 15-30 min), which decreases the binding of p300 and PP2A (Fig. 4a) while increasing the interaction of p300 with the exportin, CRM1 (Fig. 5b). This AMPK-PP2A-mTORC1 regulatory mechanism is rate-limiting with regard to mTORC1 activity and autophagy regulation in response to amino acid/glucose deprivation and refeeding appears to be relevant in a variety of cell types, including neurons, and in most tissues of mice fasted or refed after fasting. Indeed, our data suggest that both AMPK-PP2A and the BCAAs-acetyl-CoA pathways are rate-limiting for amino acid regulation of mTORC1 and autophagy. This likely explain why p300 shuttling and consequent mTORC1 and autophagy regulation are dependent on BCAAs (like Leu and Ile) that are catabolised to acetyl-CoA levels and why Arg, as an AMPK regulator is important for p300 shuttling."

Reviewer #2:

Remarks to the Author:

Min Son et al. manuscript describes how depletion of nutrients such as AA and glucose levels causes nucleus shuttling of p300. Increased nucleus p300 level decreases mTORC1 acetylation and, therefore, its activity, eventually activating the autophagy pathway. Lack of nutrients activates AMPK, which phosphorylates p300 at ser89. Phosphorylated p300 is transferred into the nucleus and decreases acetylation of mTORC1, which further activates autophagy. They also showed that the addition of nutrients to the starved cells activates phosphatase 2A which de-phosphorylates nuclear p300 and that causes its export to the cytoplasm. This shuttling regulates mTORC1 reactivation. The authors emphasized that this shuttling is altered in progeria patients, too, due to the presence of progerin which causes

mTORC1 activation and autophagy inhibition.

The authors started testing their hypothesis by showing the importance of p300 in mTORC1 regulation. They effectively included many different controls and showed the rescue effect by re-storing AAs in the media. including NLS deleted P300 construct and further showing this pattern in many different cell lines made their aim stronger. They showed this pattern in vivo using cytoplasmic and nuclear fractions of the liver and muscle and the brain of fasted versus refed mice. Using different inhibitors to kinases that can phosphorylate p300, they showed AMPK inhibition inhibited nuclear transport of p300 and its activation induces the nuclear shuttling of p300. Using additional experiments, including AMPK KD and pull-down, the authors further strengthened these claims. Next, these authors investigated how p300 gets transported back to cytoplasm after AA addition, focusing on PP2A based on LC-MC data and CRM1 (exportin). In the last two figures, the authors examined p300 in the progeria cells, in which, dysregulated mTOR activities have been observed previously.

This was a well-written report on thorough research. The data appear appropriate for testing the authors' hypothesis, and they have a straightforward research question and use appropriate methods to test this hypothesis. By channeling their findings into the progeria field, they also provided mechanistic insights into previous findings. Their studies supported the therapeutic application of mTOR inhibitors, such as rapamycin and its analogs, in treating progeria.

Concerns:

Q2-1) Many key data were shown in extended data figures (ED Figs), and the number of extended figures exceeded the main figure numbers.

→ We have conformed to the stipulated limitation of including a maximum of 6-8 main figures and up to 10 Extended Data figures in our manuscript.

Q2-2) In ED fig1, it is unclear the rationale underlying picking clone #2 of p300 k/o for future analysis. The experiments were conducted in an unstable HeLa cell line. What about other clones? Have the authors analyzed the potential off-target effects of CRISPR? In my view, ED figs 1 & 2 are important and can be moved to the main figures.

→ We conducted an assessment of mTORC1 and autophagy levels in both control and p300 KO HeLa clones (clones #1, #2, and #3 for the control group; clones #2, #4, and #9 for the p300 KO group (refer to Extended Figure 1a for details). Notably, like p300 KO #2 clone, the other KO clones (#4 and #9) also displayed mTORC1 inhibition (as shown in Extended Figure 2b) and autophagy activation (as shown in Extended Figure 2a). Our investigation included the use of the autophagy-lysosome inhibitor BafA1, revealing that all p300 KO clones exhibited activated autophagy flux (as depicted in Extended Figure 2a).

To examine any potential off-target effects of the CRISPR-Cas9 knockout system in our study, we introduced p300 wildtype cDNA into the p300 KO cells. Consequently, we observed that introducing p300 wildtype restored mTORC1 inhibition, autophagy activation, and raptor deacetylation upon p300 depletion (as shown in Figure 1a, 1h, 5g, and Extended Figures 3c, 7i, 10d).

We endeavored to incorporate some of the extended figures into the main figures; however, I apologize this proved unfeasible due to space constraints and so they remain in the extended data.

Q2-3) In Figure 4, the peak of PP2A and p300 binding occurred 15 mins after AA addition, and this association decreased afterward. Why?

→ Our data indicates that when amino acids are replenished to starved cells, there is an increase in the activity of the PP2A phosphatase (as shown in Fig. 4c, d). This increase leads to the dephosphorylation of p300 at Ser89, even within a short timeframe of 15-30 min. As a result of this dephosphorylation, the p300 interaction with PP2A is decreased, while p300 interacts better with CRM1 and can be exported from the nucleus to the cytoplasm. Therefore, we can conclude that the interaction between p300 and PP2Ac occurs very soon after amino acid re-stimulation, and this interaction leads to the dephosphorylation of p300 and might subsequently result in the loss of their interaction. We added this observation to the discussion (page 12, line 30).

“Export from the nucleus after nutrient replenishment is mediated by dephosphorylation of p300 by PP2A (even within a short timeframe of 15-30 min), which decreases the binding of p300 and PP2A (Fig. 4a) while increasing the interaction of p300 with with the exportin, CRM1 (Fig. 5b).”

Q2-4) A normal human fibroblast of a 3-year-old boy was used as a control. It was not age-matched or gender controlled. In the ideal situation, an ABE-corrected patient cell lines should be used.

→ We agree with the reviewer's comment. Now we have added new data with an expanded set of control lines which are age- and gender-matched. The details of the fibroblasts we used can be found in Extended Data Table 1 (See the table below).

Table 1. List of human dermal fibroblast cultures used in this study

No #	Culture	LMNA mutation	Donor age at biopsy	Gender	Origin
Control-#1	GM05565	None	3	M	Coriell
Control-#2	GM05565	None	3	M	Coriell
Control-#3	GM02036	None	11	F	Coriell
Control-#4	GM00969	None	2	F	Coriell
HGPS-#1	AG01972	G608G	14	F	Coriell
HGPS-#2	AG11513	G608G	8	F	Coriell

* Control-#1 and #2 both originate from the same individual (identified as #GM05565), however, they belong to different batches.

In the comparison between Control #3 and #4 and HGPS-#1 and #2, we assessed several parameters, including autophagy, mTORC1 activity, progerin levels (as shown in extended Figure 9i), as well as the intensity of LMNB1 (as depicted in extended Figure 9j). As expected, our results revealed that HGPS fibroblasts displayed activated mTORC1, inhibition of autophagy (as observed in extended Figure 9i), and reduced LMNB1 expression levels (as shown in extended Figure 9j) when compared to age- and gender-matched controls. We added these data and text to the manuscript (see Extended Data Figure 9i, j and table 1; page 11, line 7).

Extended Fig. 9i, Reduced progerin expression levels in HGPS fibroblasts by treatment with p300 inhibitor A485 for 24 h. Blots are representative of three biologically independent experiments

(N=3). Two-tailed t-test. *** $p < 0.001$ vs. HGPS-1 fibroblasts; # $p < 0.05$ vs. HGPS-1 fibroblasts (for progerin levels); * $p < 0.05$, *** $p < 0.001$ vs. control fibroblasts; # $p < 0.05$, ## $p < 0.001$ vs. vehicle treated HGPS fibroblasts (for LC3-II levels). j, Reduced Lamin B1 (LMNB1) levels in HGPS fibroblasts. Scale bar, 5 μm . N=3, about 50 cells scored per condition.

Q2-5) In Fig 7, Based on the data enough evidence was provided to believe that progerin is degraded by activation of autophagy (which was also reported by others). However, the claim that progerin itself inhibits autophagy just because of p300 mislocalization is not consistent with the data presented.

→ We agree with the reviewer's comment – it is very difficult to show that the p300 mislocalisation is the only cause of impaired autophagy caused by progerin, partly because progerin itself is degraded by autophagy. We have added some text to the discussion to make this clear (page 14, line 5).

“However, it is very difficult to show that the p300 mislocalisation is the only cause of impaired autophagy caused by progerin, partly because progerin itself is degraded by autophagy (Extended Data Fig. 10b). Thus, while we can rescue a number of mTORC1- and autophagy-related phenotypes in HGPS cells and models with strategies that reduce cytoplasmic p300 levels, we can not exclude additional mechanisms, whereby progerin may compromise autophagy.”

We have also amended the abstract (page 1, line 28).

“These results reveal how nutrients regulate mTORC1, a cytoplasmic complex, by shuttling its positive regulator p300 in and out of the nucleus and how this pathway is misregulated in HGPS contributing to mTORC1 hyperactivation and defective autophagy.”

We have added a new experiment (Extended Fig. 10d) to try to further fortify our data in this context. Here, we have looked that the levels of progerin (an autophagy substrate) in wild-type, p300 knockout cells and p300 knockout cells reconstituted with wild-type (WT), cytoplasmic (dNLS) and nuclear (IA; exportin binding deficient) p300.

It is clear that the effects of p300 absence are rescued by the wild-type and cytoplasmic p300 but not nuclear-only p300. However, while these results suggest that p300 shuttling can indeed modulate progerin levels through its influence on autophagic activity, as mentioned above, this does not exclude other potential mechanisms.

Extended Fig. 10d. Regulation of progerin degradation in p300 KO expressing p300 WT, dNLS or I265A (IA). Blots are representative of three biologically independent experiments (N=3). * $p < 0.05$ vs. control cells; ## $p < 0.01$, ### $p < 0.001$ vs. p300 KO cells (one-way ANOVA with post hoc Tukey test).

Q2-6) Please discuss why only specific AA shortage in the media can induce p300 mislocalization..

➔ The levels of Acetyl-CoA (Ac-CoA) play a crucial role in regulating p300 activity. When considering the branched-chain amino acids (BCAAs; Leu, Ile and Val, it is noteworthy that Val, unlike Leu and Ile, is recognized for its inability to generate Acetyl-CoA (the effector for mTORC1 regulation in this context), as reported by Adeva-Andany et al. in 2017. As depicted in Figure 3g, depleting Valine did not result in a reduction of mTORC1 activity. We added the text into the manuscript (page 8, line 18).

“To investigate which AAs regulate mTORC1 through AMPK, we tested single AA-depleted media (Arg-, Leu-, Ile- or Val-) on cells expressing p300 WT or SA. Depletion

of Arg, Leu or Ile but not Val activated AMPK (p-ACC at Ser79 / total ACC) and inhibited mTORC1 in p300 WT expressing cells, but p300 SA expression blunted the mTORC1 inhibition by single AA depletion (Fig. 3g)”

In Figure 3g, it is evident that the depletion of Arginine (Arg), Leucine (Leu), or Isoleucine (Ile), but notably not Valine (Val), induces AMPK activation. Our study has uncovered that AMPK plays a role in mediating p300 nuclear transport in response to starvation, potentially leading to p300 accumulation in the nucleus.

Furthermore, there is existing literature supporting the activation of AMPK in response to the depletion of single amino acids. For instance, a previous study has demonstrated that Arg depletion can activate the AMPK signaling pathway in MEF cells (Carroll et al. 2016). Thus, the Arg-AMPK connection is likely an important reason why this amino acid regulates mTORC1 in response to its availability. We have added text to the discussion to summarise this issue (page 13, line 3).

“Indeed, our data suggest that both AMPK-PP2A and the BCAAs-acetyl-CoA pathways are rate-limiting for amino acid regulation of mTORC1 and autophagy. This likely explain why p300 shuttling and consequent mTORC1 and autophagy regulation are dependent on BCAAs (like Leu and Ile) that are catabolised to acetyl-CoA (a key effector of mTORC1) and why Arg, as an AMPK regulator is important for p300 shuttling.”

Reviewer #3:

Remarks to the Author:

In this article Son et al, build on their previous findings showing that the acetyl-CoA/p300 axis may play a role in leucine sensing by mTORC1 by providing evidence suggesting that nucleo-cytoplasmic shuttling of p300 may play a role in amino acid- or glucose-induced activation of mTORC1. The authors propose a model whereby glucose or amino acid depletion sequesters p300 in the nucleus, thereby decreasing acetylation of raptor, reducing mTORC1 activity and bolstering autophagy. Some evidence is provided that nucleo-cytoplasmic shuttling of p300 may be regulated by AMPK and PP2A mediated phosphorylation and dephosphorylation of p300, respectively. Finally, Son et al., present findings suggesting that nucleo-cytoplasmic transport of p300 may be perturbed in

Hutchinson-Gilford Progeria Syndrome that is characterized by elevated mTORC1 levels. Overall, it was thought that this study is of high potential interest to a broad spectrum of research fields inasmuch as it highlights a hitherto unappreciated mechanisms of mTORC1 activation with the potential implication in human diseases. Moreover, this study was thought to provide alternative explanation to some long-standing paradigms in mTORC1 signaling that were derived from limited experimental systems that were mostly based on HEK293 cell line variants. In general, it was thought that the experimental evidence adequately supports authors' conclusions. Nonetheless, it was thought that several apparent issues should be addressed to further improve data quality, experimental rigor and further strengthen the proposed model. These specific comments and concerns were outlined below.

Major comments:

Q3-1) Control blots to estimate levels of exogenous p300 variants are missing in a number of figures (e.g., Fig 1e, 3g. etc). It was thought that these control western blots were warranted to exclude potential confounding stemming from different levels of overexpressed p300 variants.

- We agree with reviewer's comment, so we have incorporated measurements of endogenous and/or exogenous p300 expression levels in total lysates (See- Fig 1e, 3b, 3c, 4e, Extended Fig. 7g for endogenous p300; Fig. 3g, Extended Fig. 8c, 8d for exogenous p300). As shown in Fig. 1e, the depletion of amino acids did not result in any significant change in the overall p300 expression levels. However, it did lead to a noticeable alteration in its localization. In Figure 3g, we further assessed the activity of mTORC1 and AMPK in single amino acid depleted media on the same expression of p300 WT and p300 SA in p300 KO cells.

Q3-2) Quality of the western blots throughout the article seems inconsistent, whereby some blots appear to be of sub-par quality (e.g., Fig. 1e – total S6K1; Fig 4a, PP2Ac, Extended Data Fig. 1c, p300 etc.).

→ We acknowledge the reviewer's comment, so, we have included high-quality blot images in the manuscript (See Fig. 1e, 4a, Extended Fig. 1c). We hope that these images meet the required standards and are deemed acceptable.

Q3-3) In the extended figure 2d other KATs were depleted using siRNA-based approach and this was compared to the effects of p300 KO on mTORC1 activity. Considering that the depletion of other KATs does not seem to be that strong (i.e., a significant amount of targeted protein is still present), it was thought that the direct comparisons with the complete KO of p300 is somewhat inappropriate. To this end, the conclusion that other KATs are not involved should be toned down. Alternatively, an experiment comparing the effects of similar extent of depletion of p300 and other KATs on mTORC1 signaling should be performed. Finally, basal mTORC1 activity appears to be reduced in cells in which CBP was depleted by RNAi (extended data figure 2E, lane 1 vs. 4). This should be commented on in the text.

➔ We agree reviewer's comment, so, we have included data on the knockdown efficiency with siRNA and have made appropriate revisions to the manuscript to draw conclusions accordingly (page 5, line 11).

"We determined next whether other KATs had any effects on mTORC1 regulation by p300. CBP knockdown in p300 KO cells showed a slight additional effect on mTORC1 activity lowering, but this phenomenon was not seen with knockdowns of other KATs. As a small fraction of the proteins still remained, with approximately 20% of CBP, 10% of GCN5, and 30% of PCAF after transfection with siRNAs (Extended Data Fig. 2f), we can not definitively exclude small effects."

Also, we have incorporated the following text into the manuscript to address the baseline mTORC1 inhibition observed after CBP knockdown. (page 5, line 16)

"In CBP knockdown cells (with no p300 knockouts), mTORC1 activity was decreased after AAs depletion and was restored after adding back AAs to these cells, although the baseline mTORC1 activity in nutrient-replete medium was decreased compared to control (Extended Data Fig. 2g), suggesting that the highly homologous acetyltransferases p300 and CBP are non-redundant with respect to mTORC1 regulation by nutrients, consistent with previous reports."

Previous studies have indeed indicated CBP-dependent regulation of mTORC1, such as the observed reduction in acetyl-histone levels like H3K56 and the induction of autophagy, albeit to a lesser extent compared to p300 knockdown (Wan W et al. 2017). However, it's important to note that the underlying mechanisms behind these observations have not been fully understood. Further research is needed to explore and elucidate the intricate molecular pathways for CBP's role in mTORC1 regulation and its downstream effects on histone acetylation and autophagy.

Q3-4) Fig. 1h/5g– appear to show different expression levels of the two p300 mutants (anti-His-tag Western blots). This should be perhaps noted and/or commented on.

→ We added the text in the figure legend following reviewer's comments.

Q3-5) In the light of some recent studies, the authors should exclude the possibility that modulating p300 levels and/or activity may affect uptake and/or transport of glucose or amino acids (e.g., Martins et al, JCI insight 2022) and thereby affect mTORC1 signaling indirectly.

→ To test the potential impairment in glucose uptake associated with depleting p300, we performed a glucose uptake assay in p300 KO cells. Our findings indicate that p300 KO cells did not show significant differences in glucose uptake levels when compared to control cells. Additionally, the re-introduction of p300 constructs,

including the wild-type (WT), dNLS, and S89A, did not have a noticeable impact on glucose uptake in both basal and insulin-treated conditions.

This implies that, instead of suppressing glucose uptake through p300 depletion, cytoplasmic-nuclear transport of p300 is important for mTORC1 regulation. We added these data and text to the manuscript (Extended fig. 3d; page 6, line 22).

“In a previous study, when p300 and CBP were simultaneously lost in skeletal muscle and adipocytes, it resulted in a complete absence of insulin-stimulated Glc uptake into cells. To explore the potential impairment in Glc uptake associated with depleting p300, we performed a Glc uptake assay in p300 KO cells. However, our findings indicate that p300 KO HeLa cells did not exhibit significant differences in Glc uptake levels when compared to control cells (Extended Data Fig. 3d). This implies that, instead of suppressing Glc uptake through p300 depletion, cytoplasmic-nuclear transport of p300 is important for mTORC1 regulation.”

Extended Data Fig. 3d. Glucose uptake assay in p300 KO cells expressing His-tagged WT, dNLS or S89A. Insulin (1 μ M) was used to stimulate glucose uptake into cells. N=4 in each group. *** $p < 0.001$ vs. control cells (one-way ANOVA with post hoc Tukey test).

Q3-6) Considering the multifaceted nature of mTOR signaling, it appears that there should be at least some consideration that mTORC1 outputs other than autophagy may contribute to the ensuing phenotypes. Indeed, the support for the role of autophagy in HGPS cellular phenotypes is based on correlative data and use of highly pleiotropic Bafilomycin A1 (that this same group has demonstrated may interfere with proteasomal protein degradation that

is also affected by mTORC1). To this end, it was thought that the implication of autophagy should be somewhat toned down, or that additional evidence should be provided to support this conclusion.

➔ We agree reviewer's comment – we have now stated this explicitly in the discussion (page 14, line 12).

"It is important to consider that processes other than autophagy that are regulated by mTORC1 may also contribute to HGPS cellular phenotypes."

Q3-7) Most of the readouts of mTORC1 activity are limited to S6K1/rpS6 phosphorylation. Considering the emerging findings showing potential modularity of mTORC1 signaling, it is advised that other mTORC1 substrates (e.g., 4E-BP1) are also used for the readout of mTORC1 activity at least in the initial experiments. It should also be noted that rpS6 p-S235/236 Ab is not a very good readout of mTORC1 signaling as this site is also phosphorylated by AGC kinases other than S6Ks including RSKs. Accordingly, the phosphorylation of rpS6 at this site is not affected in cells that are devoid of S6Ks (Pende et al MCB 2004).

➔ We agree with reviewer's comment, so we have conducted additional experiments to include measurements of 4E-BP1 as an assessment of mTORC1 activity. We added these data and text to the manuscript, which can be found in Fig. 1a, as well as in Extended Fig. 1g and 1h.

Q3-8) A number of rescue experiments appear to be based on transient overexpression of

e.g., pcDNA-based constructs. If this is true, this should be clearly indicated in the text considering the well-recognized issues associated with this approach. Moreover, most of the interactions/associations appear to be investigated using forced expression of exogenous versions of tagged proteins. This should be clearly indicated in the text as otherwise the readers may be led to believe that these interactions/associations were established using endogenous proteins.

- In response to the reviewer's comments, we have added additional text in the figure legend to clarify whether endogenous or exogenous proteins are being referred to in the figure, enhancing the clarity and interpretability of the data presented.

Minor concerns:

Q3-9) It appears that in some cases the increase in cytoplasmic p300 is not accompanied by corresponding decrease in nuclear p300 upon amino acid or glucose repletion. What happens to total p300 levels upon amino acid or glucose repletion?

- As shown in Fig. 1e and Extended fig. 1g-j, the depletion of amino acids or glucose did not lead to any significant changes in p300 expression levels within total lysates. However, it resulted in a noticeable alteration in its subcellular localization, as shown in Figure 1e. It's important to note that p300 is primarily localized in the nucleus, and therefore, the Western blot data may not clearly demonstrate a decrease in cytoplasmic p300 accompanied by a corresponding increase in nuclear p300 in response to starvation. However, immunostaining images provide a more detailed view, indicating a reduction in cytoplasmic p300 and an increase in nuclear p300 upon starvation, as evidenced in Figure 1c, 1d, 3a, 3e, and 4f.

Q3-10) The authors should likely discuss why the focus was on glucose and amino acids and not on other stimuli known to alter mTORC1 signaling.

→ In response to the reviewer's comments, we have added additional text in the discussion to elucidate the other stimuli known to regulate mTORC1 activity. (page 13, line 10)

“In addition to nutrient status, mTORC1 also responds to intracellular and environmental stresses that hinder growth, such as hypoxia or DNA damage. Hypoxia inhibits mTORC1 partly by activating AMPK and also by inducing REDD1 (Regulated in DNA damage and development 1), which in turn activates TSC. The DNA damage response pathway inhibits mTORC1 by inducing p53 target genes, including the AMPK regulatory subunit (AMPK β) and PTEN. It's worth noting that other stimuli that regulate mTORC1 activity can also be influenced by AMPK. Therefore, it would be intriguing to investigate whether hypoxia or DNA damage can modulate mTORC1 by altering the localization of p300.”

Q3-11) The article may benefit from some careful editing. E.g., Page 4, line 101, “the” seems to be misplaced and should be moved to read “Previously, depletion of leucine (Leu) was shown to inhibit mTORC1 activity by multiple pathways, including the Leu-MCCC1-p300 axis.”; Page 4, line 129, a space is missing between “regulate” and “mTORC1”. etc.

→ Thank you for spotting our error. We have corrected this.

Q3-12) Page 10, lines 304-306, “Progerin-mediated DNA damage (assessed using an antibody against the DNA double strand break marker γ -H2AX) in HGPS fibroblasts was also reduced by treatment with A485, 991 (AMPK activator) or Selinexor (SEL; nuclear export inhibitor) (Fig. 7b)”. The reference to Figure 7b does not seem appropriate here, as the indicated compounds appear to be used in the experiments shown in Figure 7c, where LMNB, not γ -H2AX, is assessed.

→ Thank you for spotting our error. We have corrected this.

Q3-13) Figure legends are missing detail. For instance, it should be indicated that the p300 constructs in figure 1a are His-tagged, etc. In general, it was thought that it would be beneficial to indicate the name of the proteins and not just tags in the figures. Specific phosphorylation sites of substrates assessed by immunoblotting should also be indicated throughout the article.

→ We added the information to the figure legends following the reviewer’s comments.

Q3-14) Employing SEM for Western blot densitometry data generated from the replicates using the same cell line was also thought to be somewhat dubious and likely to inflate the p values, but it was thought that the differences between experimental conditions were rather clear, and thus this was not a major source of concern. In addition, some of the statistical tests that were employed are not identified in neither the figure legends nor methods.

→ Each time we show SEMs these are from 3 or more independent biological experiments from the same cell lines. This is standard practice. We have meticulously reevaluated all the data in our study to conduct a thorough analysis using the appropriate statistical tests. Subsequently, we have added this information into the figure legends.

Q3-15) Conclusion that upregulation of CRM1 and lower AMPK activity are responsible to increased mTORC1 activity in HGPS fibroblasts was thought to be somewhat premature considering the evidence, and previous literature showing that AMPK can suppress mTORC1

signaling in a variety of ways, whereby it was thought that the authors do not provide sufficient data to conclude that the effects of dampened AMPK signaling on mTORC1 are exclusively mediated by p300, in the context of HGPS. Some toning down of the statements in this section of the manuscript appears to be warranted.

→ We agree reviewer's comment and have added text to make this explicit. (page 11, line 1)

“These data suggest that upregulated CRM1 and less AMPK activation in HGPS cells induce more cytoplasmic p300 localization, which would contribute to mTORC1 activation and autophagy inhibition. However, we can not exclude additional effects of AMPK on mTORC1 via distinct pathways.”

Curious question:

Q3-16) I understand that this is out of the scope of the manuscript, but did the authors attempt to establish the role of p300 in mTORC1 regulation in TSC1 or 2 KO cells in e.g., response to glucose depletion and repletion? This again is a personal curiosity of the reviewer (not associated with the evaluation of this current article) and is based on some previous findings showing that p300 may be increased in TSC2 KO cells, whereby AMPK was thought to act via TSC at least in some contexts.

→ Thank you for the interesting question. We'll do this experiment as soon as we complete our current work.

References

Adeva-Andany, M.M., Lopez-Maside, L., Donapetry-Garcia, C., Fernandez-Fernandez, C., and Sixto-Leal, C. (2017). Enzymes involved in branched-chain amino acid metabolism in humans. *Amino Acids* 49, 1005-1028

Bernadette Carroll, Dorothea Maetzel, Oliver DK Maddocks, Gisela Otten, Matthew Ratcliff, Graham R Smith, Elaine A Dunlop, João F Passos, Owen R Davies, Rudolf Jaenisch, Andrew R Tee, Sovan Sarkar, Viktor I Korolchuk (2016) Control of TSC2-Rheb signaling axis by arginine regulates mTORC1 activity eLife 5:e11058

Cao K, Graziotto JJ, Blair CD, Mazzulli JR, Erdos MR, Krainc D, Collins FS. Rapamycin reverses cellular phenotypes and enhances mutant protein clearance in Hutchinson-Gilford progeria syndrome cells. *Sci Transl Med.* 2011 Jun 29;3(89):89ra58. doi: 10.1126/scitranslmed.3002346. PMID: 21715679.

Eliassen MM, Winkler W, Jordan V, Pokar M, Marchetti M, Roth E, Allmaier G, Oehler R. Adaptive cellular mechanisms in response to glutamine-starvation. *Front Biosci.* 2006 Sep 1;11:3199-211. doi: 10.2741/2043. PMID: 16720386.

Li W, Xie L, Chen Z, Zhu Y, Sun Y, Miao Y, Xu Z, Han X. Cantharidin, a potent and selective PP2A inhibitor, induces an oxidative stress-independent growth inhibition of pancreatic cancer cells through G2/M cell-cycle arrest and apoptosis. *Cancer Sci.* 2010 May;101(5):1226-33. doi: 10.1111/j.1349-7006.2010.01523.x. Epub 2010 Feb 5. PMID: 20331621.

Son SM, Park SJ, Stamatakou E, Vicinanza M, Menzies FM, Rubinsztein DC. Leucine regulates autophagy via acetylation of the mTORC1 component raptor. *Nat Commun.* 2020 Jun 19;11(1):3148. doi: 10.1038/s41467-020-16886-2. PMID: 32561715; PMCID: PMC7305105.

Wan W, You Z, Xu Y, Zhou L, Guan Z, Peng C, Wong CCL, Su H, Zhou T, Xia H, Liu W. mTORC1 Phosphorylates Acetyltransferase p300 to Regulate Autophagy and Lipogenesis. *Mol Cell.* 2017 Oct 19;68(2):323-335.e6. doi: 10.1016/j.molcel.2017.09.020. Epub 2017 Oct 12. PMID: 29033323.

Decision Letter, first revision:

*Please delete the link to your author homepage if you wish to forward this email to co-authors.

Dear Professor Rubinsztein,

Thank you for submitting your revised manuscript, "p300 cytoplasm-nucleus shuttling underlies mTORC1 hyperactivation in Hutchinson-Gilford Progeria Syndrome", to the journal. All original reviewers have now provided feedback on the revision (please see below). As you will see from their comments, they appreciated the revision efforts and found that most of their points have been addressed. However, Rev#1 shared a persistent concern that we found important. Although we also continue to be very interested in this study, we believe that their concern should be addressed before publication in *Nature Cell Biology*.

Please address the remaining concern from the rev with flux analyses to determine whether autophagic activity is in line with the LC3 blot and addresses their original concern linking acetylation

of RAPTOR to changes in autophagy flux.

As always, please pay close attention to our guidelines on statistical and methodological reporting (listed below) as failure to do so may delay the reconsideration of the revised manuscript. In particular, please provide:

- a Supplementary Figure including unprocessed images of all gels/blots in the form of a multi-page pdf file. Please ensure that blots/gels are labeled and the sections presented in the figures are clearly indicated.
- a Supplementary Table including all numerical source data in Excel format, with data for different figures provided as different sheets within a single Excel file. The file should include source data giving rise to graphical representations and statistical descriptions in the paper and for all instances where the figures present representative experiments of multiple independent repeats, the source data of all repeats should be provided.

We therefore invite you to take these points into account when revising the manuscript. In addition, when preparing the revision please:

- ensure that it conforms to our format instructions and publication policies (see below and <https://www.nature.com/nature/for-authors>).
- provide a point-by-point rebuttal to the full referee reports verbatim, as provided at the end of this letter.
- provide the completed Reporting Summary (found here <https://www.nature.com/documents/nr-reporting-summary.pdf>). This is essential for reconsideration of the manuscript and will be available to editors and referees in the event of peer review. For more information see <http://www.nature.com/authors/policies/availability.html> or contact me.

When submitting the revised version of your manuscript, please pay close attention to our [href="https://www.nature.com/nature-portfolio/editorial-policies/image-integrity">Digital Image Integrity Guidelines](https://www.nature.com/nature-portfolio/editorial-policies/image-integrity). and to the following points below:

Nature Cell Biology is committed to improving transparency in authorship. As part of our efforts in this

direction, we are now requesting that all authors identified as 'corresponding author' on published papers create and link their Open Researcher and Contributor Identifier (ORCID) with their account on the Manuscript Tracking System (MTS), prior to acceptance. ORCID helps the scientific community achieve unambiguous attribution of all scholarly contributions. You can create and link your ORCID from the home page of the MTS by clicking on 'Modify my Springer Nature account'. For more information please visit www.springernature.com/orcid.

This journal strongly supports public availability of data. Please place the data used in your paper into a public data repository, or alternatively, present the data as Supplementary Information. If data can only be shared on request, please explain why in your Data Availability Statement, and also in the correspondence with your editor. Please note that for some data types, deposition in a public repository is mandatory - more information on our data deposition policies and available repositories appears below.

[redacted]

We would like to receive the revision within four weeks. If submitted within this time period, reconsideration of the revised manuscript will not be affected by related studies published elsewhere, or accepted for publication in Nature Cell Biology in the meantime. We would be happy to consider a revision even after this timeframe, but in that case we will consider the published literature at the time of resubmission when assessing the file.

As always, please do not hesitate to reach out with any questions or concerns, and we look forward to your final revision.

Best wishes,

Melina

Melina Casadio, PhD
Senior Editor, Nature Cell Biology
ORCID ID: <https://orcid.org/0000-0003-2389-2243>

Reviewers' Comments:

Reviewer #1:

Remarks to the Author:

The Authors have addressed the comments I raised in my previous review. Here I address the author

responses to those comments. I have addressed novelty and significance in my previous review.

Q1-1. On strengthening autophagy-related data pertaining to loss of P300, the authors have supplied significant new data and have now shown autophagy has been increased using flux assays (both biochemical and ratiometric fluorescence-based techniques), and aggregate clearance assays. Collectively these data are strong.

Q1-2. Use of statistics - the authors have employed one-sample t-tests where appropriate.

Q1-3. Proof that acetylation of RAPTOR decreases autophagy - the authors refer to a previous publication where RAPTOR is modified such that it cannot be acetylated. This does prevent mTORC1 activation upon nutrient stimulation, however, only a western blot for LC3 is provided as proof for autophagy stimulation (with no flux measurement). LC3II (lipid bound marker for autophagosomes) increases upon leucine depletion in wild type RAPTOR expressing cells, and is also increased when mutant RAPTOR is expressed in the presence of nutrients. On balance, when it is shown that mutant RAPTOR cannot support mTORC1 activity upon nutrient stimulation, and LC3II is correspondingly increased, it is highly likely that autophagy is increased. However, this level of evidence falls short of the flux assays and aggregate clearance experiments that were provided as evidence in response to Q1-1.

Q1-4. Discussion of the GATOR-RAGULATOR-RAGs axis - text has been added describing experimental data that looked at the relationship between the GATOR2 inhibitor, SESTRIN2, and the action of P300 on mTORC1. This now adds some context.

Kind regards

Tim Sargeant

Reviewer #2:

Remarks to the Author:

The authors have addressed my concerns.

Reviewer #3:

Remarks to the Author:

I thought that the authors appropriately addressed all of my comments. To this end, I have no further concerns regarding this manuscript.

Sincerely

I/Topisirovic

GUIDELINES FOR SUBMISSION OF NATURE CELL BIOLOGY ARTICLES

ARTICLE FORMAT

ABSTRACT – should not exceed 150 words and should be unreferenced. This paragraph is the most visible part of the paper and should briefly outline the background and rationale for the work, and accurately summarize the main results and conclusions. Key genes, proteins and organisms should be specified to ensure discoverability of the paper in online searches.

TEXT – the main text consists of the Introduction, Results, and Discussion sections and must not exceed 3500 words including the abstract. The Introduction should expand on the background relating to the work. The Results should be divided in subsections with subheadings, and should provide a concise and accurate description of the experimental findings. The Discussion should expand on the findings and their implications. All relevant primary literature should be cited, in particular when discussing the background and specific findings.

REFERENCES – are limited to a total of 70 in the main text and Methods combined,. They must be numbered sequentially as they appear in the main text, tables and figure legends and Methods and must follow the precise style of Nature Cell Biology references. References only cited in the Methods should be numbered consecutively following the last reference cited in the main text. References only associated with Supplementary Information (e.g. in supplementary legends) do not count toward the total reference limit and do not need to be cited in numerical continuity with references in the main text. Only published papers can be cited, and each publication cited should be included in the numbered reference list, which should include the manuscript titles. Footnotes are not permitted.

Methods should be written concisely, but should contain all elements necessary to allow interpretation and replication of the results. As a guideline, Methods sections typically do not exceed 3,000 words. The Methods should be divided into subsections listing reagents and techniques. When citing previous methods, accurate references should be provided and any alterations should be noted. Information must be provided about: antibody dilutions, company names, catalogue numbers and clone numbers for monoclonal antibodies; sequences of RNAi and cDNA probes/primers or company names and catalogue numbers if reagents are commercial; cell line names, sources and information on cell line identity and authentication. Animal studies and experiments involving human subjects must be reported in detail, identifying the committees approving the protocols. For studies involving human subjects/samples, a statement must be included confirming that informed consent was obtained. Statistical analyses and information on the reproducibility of experimental results should be provided in a section titled "Statistics and Reproducibility".

All Nature Cell Biology manuscripts submitted on or after March 21 2016, must include a Data availability statement as a separate section after Methods but before references, under the heading "Data Availability". For Springer Nature policies on data availability see <http://www.nature.com/authors/policies/availability.html>; for more information on this particular policy see <http://www.nature.com/authors/policies/data/data-availability-statements-data-citations.pdf>. The Data availability statement should include:

- Accession codes for primary datasets (generated during the study under consideration and designated as "primary accessions") and secondary datasets (published datasets reanalysed during the study under consideration, designated as "referenced accessions"). For primary accessions data should be made public to coincide with publication of the manuscript. A list of data types for which submission to community-endorsed public repositories is mandated (including sequence, structure, microarray, deep sequencing data) can be found here <http://www.nature.com/authors/policies/availability.html#data>.

- Unique identifiers (accession codes, DOIs or other unique persistent identifier) and hyperlinks for datasets deposited in an approved repository, but for which data deposition is not mandated (see here for details <http://www.nature.com/sdata/data-policies/repositories>).
- At a minimum, please include a statement confirming that all relevant data are available from the authors, and/or are included with the manuscript (e.g. as source data or supplementary information), listing which data are included (e.g. by figure panels and data types) and mentioning any restrictions on availability.
- If a dataset has a Digital Object Identifier (DOI) as its unique identifier, we strongly encourage including this in the Reference list and citing the dataset in the Methods.

We recommend that you upload the step-by-step protocols used in this manuscript to the Protocol Exchange. More details can found at www.nature.com/protocolexchange/about.

DISPLAY ITEMS – main display items are limited to 6-8 main figures and/or main tables. For Supplementary Information see below.

FIGURES – Colour figure publication costs \$395 per colour figure. All panels of a multi-panel figure must be logically connected and arranged as they would appear in the final version. Unnecessary figures and figure panels should be avoided (e.g. data presented in small tables could be stated briefly in the text instead).

All imaging data should be accompanied by scale bars, which should be defined in the legend. Cropped images of gels/blots are acceptable, but need to be accompanied by size markers, and to retain visible background signal within the linear range (i.e. should not be saturated). The boundaries of panels with low background have to be demarked with black lines. Splicing of panels should only be considered if unavoidable, and must be clearly marked on the figure, and noted in the legend with a statement on whether the samples were obtained and processed simultaneously. Quantitative comparisons between samples on different gels/blots are discouraged; if this is unavoidable, it has to be performed for samples derived from the same experiment with gels/blots were processed in parallel, which needs to be stated in the legend.

- For line art, graphs, charts and schematics we prefer Adobe Illustrator (.AI), Encapsulated PostScript

(.EPS) or Portable Document Format (.PDF). Files should be saved or exported as such directly from the application in which they were made, to allow us to restyle them according to our journal house style.

Regardless of format, all figures must be vector graphic compatible files, not supplied in a flattened raster/bitmap graphics format, but should be fully editable, allowing us to highlight/copy/paste all text and move individual parts of the figures (i.e. arrows, lines, x and y axes, graphs, tick marks, scale bars etc). The only parts of the figure that should be in pixel raster/bitmap format are photographic images or 3D rendered graphics/complex technical illustrations.

Supplementary items should relate to a main text figure, wherever possible, and should be mentioned

sequentially in the main manuscript, designated as Supplementary Figure, Table, Video, or Note, and numbered continuously (e.g. Supplementary Figure 1, Supplementary Figure 2, Supplementary Table 1, Supplementary Table 2 etc.).

Unprocessed scans of all key data generated through electrophoretic separation techniques need to be presented in a supplementary figure that should be labeled and numbered as the final supplementary figure, and should be mentioned in every relevant figure legend. This figure does not count towards the total number of figures and is the only figure that can be displayed over multiple pages, but should be provided as a single file, in PDF or TIFF format. Data in this figure can be displayed in a relatively informal style, but size markers and the figures panels corresponding to the presented data must be indicated.

The total number of Supplementary Figures (not including the “unprocessed scans” Supplementary Figure) should not exceed the number of main display items (figures and/or tables (see our Guide to Authors and March 2012 editorial <http://www.nature.com/ncb/authors/submit/index.html#supinfo>; <http://www.nature.com/ncb/journal/v14/n3/index.html#ed>). No restrictions apply to Supplementary Tables or Videos, but we advise authors to be selective in including supplemental data.

GUIDELINES FOR EXPERIMENTAL AND STATISTICAL REPORTING

REPORTING REQUIREMENTS – We ask authors to complete a Reporting Summary that collects information on experimental design and reagents. We hope this will aid in your evaluation of the paper. The Reporting Summary can be found here <https://www.nature.com/documents/nr-reporting-summary.pdf>) Please note that these forms are dynamic ‘smart pdfs’ and must therefore be downloaded and completed in Adobe Reader. We will then flatten them for ease of use. If you would like to reference the guidance text as you complete the template, please access these flattened versions at <http://www.nature.com/authors/policies/availability.html>.

Information on how many times each experiment was repeated independently with similar results

needs to be provided in the legends and/or Methods for all experiments, and in particular wherever representative experiments are shown.

Author Rebuttal, first revision:

16 November 2023

Dear Dr Casadio

RE: NCB-A51241A

Many thanks for your email of 27 October 2023 regarding this paper. We have addressed the outstanding request from yourself and the reviewer using two approaches and have also added an extra experiment to fortify related conclusions.

KEY POINTS HIGHLIGHTED BY EDITOR:

Thank you for submitting your revised manuscript, "p300 cytoplasm-nucleus shuttling underlies mTORC1 hyperactivation in Hutchinson-Gilford Progeria Syndrome", to the journal. All original reviewers have now provided feedback on the revision (please see below). As you will see from their comments, they appreciated the revision efforts and found that most of their points have been addressed. However, Rev#1 shared a persistent concern that we found important. Although we also continue to be very interested in this study, we believe that their concern should be addressed before publication in Nature Cell Biology.

Please address the remaining concern from the rev with flux analyses to determine whether autophagic activity is in line with the LC3 blot and addresses their original concern linking acetylation of RAPTOR to changes in autophagy flux.

- ➔ We have added a new experiment (Extended Fig. 3d-f) to assess the effects of raptor acetylation of autophagic flux. In Extended Fig. 3d, we found that raptor knockdown increased LC3-II levels under both basal and BafA1-treated conditions. Thus this increases autophagic flux - as BafA1 blocks lysosomal degradation, LC3-II in BafA1 conditions reflect LC3-II/autophagosome synthesis. This increase was effectively reversed by reconstituting these raptor knockout cells with raptor WT, but not with the acetylation-dead mutant (KR; K1097R), consistent with our model and previous data showing that raptor acetylation increases mTORC1 activity and impedes autophagosome biogenesis.

To confirm these data, we performed an orthogonal autophagic flux assay using transgenically-expressed tandem-tagged mRFP-GFP-LC3 reporter as shown in Extended Fig. 3e. This assay allows to characterize the LC3-II vesicles and discriminate between autophagosomes or non-acidified lysosomes, and acidified autolysosomes. Due to the different pKa of the two LC3 tags (< 4.5 for mRFP and

around 6 for GFP), the GFP signal is rapidly quenched by the acidic lysosomal environment, while the mRFP persists. Thus, the vesicles that emit both red and green signals are autophagosomes or non-acidified lysosomes, while the vesicles that emit only the red signal (due to the GFP-signal being quenched by an acidic environment) are acidic autolysosomes (Ref below). Here, red-only LC3 vesicles represent end stage acidified autolysosomes and their numbers reflect autophagic flux. While overexpression of wild-type raptor decreased autophagy flux (red-only LC3 vesicle numbers) in raptor-knockdown cells, this was not seen with the raptor KR mutant that has a mutation at the lysine that is acetylated (Extended Data Fig. 3e).

Furthermore, we investigated whether p300 shuttling impacts autophagy via raptor acetylation. In Extended Fig. 3f, p300 KO cells expressing raptor KR showed no change in LC3-II (in the presence or absence of BafA1), compared to raptor WT. However, in p300 KO cells reconstituted with p300 WT or dNLS, we observed elevated LC3-II levels when raptor KR was transfected, compared to raptor WT. This suggests that cytoplasmic p300 negatively regulates autophagosome biogenesis (LC3-II in BafA1) in a manner that is dependent on the raptor acetylation sites K1097. We added these data and text to the manuscript (see Extended Data Figure 3d-f; page 6, line 13).

“Previously, we reported that p300 regulated mTORC1 and subsequently autophagy by acetylating raptor, rather than other substrates 8, 9. Consistent with our previous data, raptor knockdown increased LC3-II levels under both basal and BafA1-treated conditions (Extended Data Fig. 3d). This increase was effectively reversed by reconstituting these raptor knockout cells with raptor WT, but not with the acetylation-dead mutant (KR; K1097R 8) (Extended Data Fig. 3d), consistent with our model and previous data showing that raptor acetylation increases mTORC1 activity and impedes autophagosome biogenesis 8, 9. We confirmed these data with an autophagic flux assay using transgenically-expressed tandem-tagged mRFP-GFP-LC3 reporter 31, where unacidified autophagosomes are green and red = yellow, while acidified autolysosomes are red only. While overexpression of wild-type raptor decreased autophagy flux (red-only LC3 vesicle numbers) in raptor-knockdown cells, this was not seen with the raptor KR mutant (Extended Data Fig. 3e). To investigate whether p300 shuttling impacts autophagy via raptor acetylation, p300 KO cells expressing raptor KR showed no change in LC3-II (in the presence or absence of BafA1), compared to raptor WT. However, in p300 KO cells reconstituted with p300 WT or dNLS, we observed elevated LC3-II levels when raptor KR was transfected, compared to raptor WT (Extended Data Fig. 3f). This suggests that cytoplasmic p300 negatively regulates autophagosome biogenesis (LC3-II in BafA1) in a manner that is dependent on the raptor acetylation site K1097.”

Extended Data Fig. 3d

Extended Data Fig. 3e

Extended Data Fig. 3f

As always, please pay close attention to our guidelines on statistical and methodological reporting (listed below) as failure to do so may delay the reconsideration of the revised manuscript. In particular, please provide:

➔ This has been done and all source data are provided as a Source data file.

➔ This has been done and all source data are provided as a Source data file.

POINT-BY-POINT RESPONSES TO ALL REVIEWERS' COMMENTS:

Reviewers' Comments:

Reviewer #1:

Remarks to the Author:

The Authors have addressed the comments I raised in my previous review. Here I address the author responses to those comments. I have addressed novelty and significance in my previous review.

Q1-1. On strengthening autophagy-related data pertaining to loss of P300, the authors have supplied significant new data and have now shown autophagy has been increased using flux assays (both biochemical and ratiometric fluorescence-based techniques), and aggregate clearance assays. Collectively these data are strong.

→ Thank you very much.

Q1-2. Use of statistics - the authors have employed one-sample t-tests where appropriate.

→ Thank you very much.

Q1-3. Proof that acetylation of RAPTOR decreases autophagy - the authors refer to a previous publication where RAPTOR is modified such that it cannot be acetylated. This does prevent mTORC1 activation upon nutrient stimulation, however, only a western blot for LC3 is provided as proof for autophagy stimulation (with no flux measurement). LC3II (lipid bound marker for autophagosomes) increases upon leucine depletion in wild type RAPTOR expressing cells, and is also increased when mutant RAPTOR is expressed in the presence of nutrients. On balance, when it is shown that mutant RAPTOR cannot support mTORC1 activity upon nutrient stimulation, and LC3II is correspondingly increased, it is highly likely that autophagy is increased. However, this level of evidence falls short of the flux assays and aggregate clearance experiments that were provided as evidence in response to Q1-1.

→ Please see comments to editor related to new experiments in Extended data Figs 3d-f above.

Q1-4. Discussion of the GATOR-RAGULATOR-RAGs axis - text has been added describing experimental data that looked at the relationship between the GATOR2 inhibitor, SESTRIN2, and the action of P300 on mTORC1. This now adds some context.

→ Thank you very much.

< References >

Kimura, S., Noda, T. & Yoshimori, T. Dissection of the autophagosome maturation process by a novel reporter protein, tandem fluorescent-tagged LC3. *Autophagy* 3, 452–460 (2007)

Decision Letter, second revision:

Our ref: NCB-A51241B

20th November 2023

Dear Dr. Rubinsztein,

Thank you for submitting your revised manuscript "p300 cytoplasm-nucleus shuttling underlies mTORC1 hyperactivation in Hutchinson-Gilford Progeria Syndrome" (NCB-A51241B) again. Upon the advice of Reviewer #1, we'll be happy in principle to publish the manuscript in Nature Cell Biology, pending minor revisions to comply with our editorial and formatting guidelines.

We are beginning to perform detailed checks on your paper and will send you a checklist detailing our editorial and formatting requirements in about 1-2 weeks. Please do not upload the final materials and make any revisions until you receive this additional information from us.

Thank you again for your interest in Nature Cell Biology. Please do not hesitate to contact me if you have any questions.

Sincerely,

Melina

Melina Casadio, PhD
Senior Editor, Nature Cell Biology
ORCID ID: <https://orcid.org/0000-0003-2389-2243>

Reviewer #1 (Remarks to the Author):

The authors have fully answered all of my comments - it is a great paper that should now be published in NCB.

Kind regards

Tim Sargeant

Decision Letter, final checks:

Our ref: NCB-A51241B

28th November 2023

Dear Dr. Rubinsztein,

Thank you for your patience as we've prepared the guidelines for final submission of your Nature Cell Biology manuscript, "p300 cytoplasm-nucleus shuttling underlies mTORC1 hyperactivation in Hutchinson-Gilford Progeria Syndrome" (NCB-A51241B). Please carefully follow the step-by-step instructions provided in the attached file, and add a response in each row of the table to indicate the changes that you have made. Please also check and comment on any additional marked-up edits we have proposed within the text. Ensuring that each point is addressed will help to ensure that your revised manuscript can be swiftly handed over to our production team.

In recognition of the time and expertise our reviewers provide to Nature Cell Biology's editorial process, we would like to formally acknowledge their contribution to the external peer review of your manuscript entitled "p300 cytoplasm-nucleus shuttling underlies mTORC1 hyperactivation in Hutchinson-Gilford Progeria Syndrome". For those reviewers who give their assent, we will be publishing their names alongside the published article.

Nature Cell Biology offers a Transparent Peer Review option for new original research manuscripts submitted after December 1st, 2019. As part of this initiative, we encourage our authors to support increased transparency into the peer review process by agreeing to have the reviewer comments, author rebuttal letters, and editorial decision letters published as a Supplementary item. When you submit your final files please clearly state in your cover letter whether or not you would like to participate in this initiative. Please note that failure to state your preference will result in delays in accepting your manuscript for publication.

Cover suggestions

COVER ARTWORK: We welcome submissions of artwork for consideration for our cover. For more information, please see our guide for cover artwork.

Nature Cell Biology has now transitioned to a unified Rights Collection system which will allow our Author Services team to quickly and easily collect the rights and permissions required to publish your work. Approximately 10 days after your paper is formally accepted, you will receive an email in providing you with a link to complete the grant of rights. If your paper is eligible for Open Access, our Author Services team will also be in touch regarding any additional information that may be required to arrange payment for your article.

Please note that *Nature Cell Biology* is a Transformative Journal (TJ). Authors may publish their

research with us through the traditional subscription access route or make their paper immediately open access through payment of an article-processing charge (APC). Authors will not be required to make a final decision about access to their article until it has been accepted. Find out more about Transformative Journals

Please use the following link for uploading these materials:
[Redacted]

Best regards,

Kendra Donahue
Staff
Nature Cell Biology

On behalf of

Melina Casadio, PhD
Senior Editor, Nature Cell Biology
ORCID ID: <https://orcid.org/0000-0003-2389-2243>

Reviewer #1:

Remarks to the Author:

The authors have fully answered all of my comments - it is a great paper that should now be published in NCB.

Kind regards

Tim Sargeant

Final Decision Letter:

Dear Dr Rubinsztein,

I am pleased to inform you that your manuscript, "p300 nucleocytoplasmic shuttling underlies mTORC1 hyperactivation in Hutchinson-Gilford Progeria Syndrome", has now been accepted for publication in Nature Cell Biology.

You may wish to make your media relations office aware of your accepted publication, in case they consider it appropriate to organize some internal or external publicity. Once your paper has been scheduled you will receive an email confirming the publication details. This is normally 3-4 working days in advance of publication. If you need additional notice of the date and time of publication,

please let the production team know when you receive the proof of your article to ensure there is sufficient time to coordinate. Further information on our embargo policies can be found here: <https://www.nature.com/authors/policies/embargo.html>

Please note that *Nature Cell Biology* is a Transformative Journal (TJ). Authors may publish their research with us through the traditional subscription access route or make their paper immediately open access through payment of an article-processing charge (APC). Authors will not be required to make a final decision about access to their article until it has been accepted. Find out more about Transformative Journals

If you have not already done so, we strongly recommend that you upload the step-by-step protocols used in this manuscript to the Protocol Exchange (www.nature.com/protocolexchange), an open online resource established by Nature Protocols that allows researchers to share their detailed experimental know-how. All uploaded protocols are made freely available, assigned DOIs for ease of citation and are fully searchable through nature.com. Protocols and Nature Portfolio journal papers in which they are used can be linked to one another, and this link is clearly and prominently visible in the online versions of both papers. Authors who performed the specific experiments can act as primary authors for the Protocol as they will be best placed to share the methodology details, but the Corresponding Author of the present research paper should be included as one of the authors. By uploading your Protocols to Protocol Exchange, you are enabling researchers to more readily reproduce or adapt the methodology you use, as well as increasing the visibility of your protocols and papers. You can also establish a dedicated page to collect your lab Protocols. Further information can be found at www.nature.com/protocolexchange/about

With kind regards,

Melina Casadio, PhD
Senior Editor, Nature Cell Biology
ORCID ID: <https://orcid.org/0000-0003-2389-2243>
